## OPEN

# Polycomb repressive complex 2 shields naïve human pluripotent cells from trophectoderm differentiation

Banushree Kumar [1,2,7], Carmen Navarro[1,2,7], Nerges Winblad[2,3,4,7], John P. Schell[2,3,4], Cheng Zhao[3,4], Jere Weltner[2,3,4], Laura Baqué-Vidal[2,3,4], Angelo Salazar Mantero[1,2], Sophie Petropoulos [3,4,5,6], Fredrik Lanner [2,3,4 ✉] and Simon J. Elsässer [1,2 ✉]

The first lineage choice in human embryo development separates trophectoderm from the inner cell mass. Naïve human embryonic stem cells are derived from the inner cell mass and offer possibilities to explore how lineage integrity is maintained. Here, we discover that polycomb repressive complex 2 (PRC2) maintains naïve pluripotency and restricts differentiation to trophectoderm and mesoderm lineages. Through quantitative epigenome profiling, we found that a broad gain of histone H3 lysine 27 trimethylation (H3K27me3) is a distinct feature of naïve pluripotency. We define shared and naïve-specific bivalent promoters featuring PRC2-mediated H3K27me3 concomitant with H3K4me3. Naïve bivalency maintains key trophectoderm and mesoderm transcription factors in a transcriptionally poised state. Inhibition of PRC2 forces naïve human embryonic stem cells into an 'activated' state, characterized by co-expression of pluripotency and lineage-specific transcription factors, followed by differentiation into either trophectoderm or mesoderm lineages. In summary, PRC2-mediated repression provides a highly adaptive mechanism to restrict lineage potential during early human development.

Human embryonic stem cells (hESCs) derived from the inner cell mass (ICM) of the developing blastocysts[1] have the capacity to self-renew and differentiate into any of the three (ectoderm, mesoderm and endoderm) germ lineages, a feature defined as pluripotency. Intriguing data[2–9] also suggest that hESCs, in contrast to mouse embryonic stem cells (mESCs), can be induced to differentiate into extra-embryonic lineages such as trophectoderm cells, a capacity normally restricted to totipotent stem cells. The mechanisms restricting epiblast cells from entering trophectoderm lineage in the developing blastocyst remain elusive.

Conventional hESCs resemble the post-implantation epiblast and their features are similar to murine epiblast stem cells[10,11], emanating the name: primed hESCs[12]. Pre-implantation-like hESCs, referred to as 'naïve', recapitulate the gene expression, X chromosome re-activation and DNA methylation patterns observed in the pre-implantation ICM in vivo[13–18]. Genomic, transcriptomic, proteomic and epigenomic analyses have been conducted to understand the molecular mechanisms governing distinct pluripotent states[19–27].

Promoter bivalency, the presence of both a 'repressive' histone post-translational modification H3K27me3 and 'active' mark H3K4me3, has been attributed an important role in regulating developmental genes[28–30]. However, not much attention has been directed towards exploring a possible role of this epigenetic signature in naïve pluripotency, owing partly to reports that polycomb repressive complex 2 (PRC2), responsible for H3K27 methylation, is required for primed but dispensable for naïve pluripotency[31–33].

Polycomb repressive complex 1 (PRC1), responsible for H2AK119 mono-ubiquitination (hereafter H2Aub), has been shown to be essential for self-renewal, pluripotency and ordered differentiation of mESCs and hESCs[34–40], but the H2Aub landscape of naïve hESCs is unexplored.

In this Article, we use quantitative epigenome and transcriptome profiling of naïve and primed hESCs, and functionally characterize distinct roles for PRC2 in the two states through pharmacologic inhibition and genetic targeting. To understand the functional role of PRC2 in setting up the naïve epigenomic and transcriptional landscape, we depleted H3K27me3 by inhibiting PRC2 in naïve and primed state. We discover a key role of PRC2 in shielding naïve human pluripotency from trophectoderm and mesoderm differentiation.

## Results

**Diffuse gain and distinct H3K27me3 patterns in naïve cells.** To elucidate the function of promoter bivalency in naïve and primed hESCs, we performed quantitative chromatin immunoprecipitation followed by sequencing (ChIP–seq) on H9 female hESCs, maintained in naïve (t2iLGö) or primed (E8) culture conditions, with and without EZH1/2 inhibitor (EZH2i, EPZ-6438[41]) for 7 days (Fig. 1a). We quantitatively profiled three histone modifications associated with bivalent genes, H3K4me3, H3K27me3 and H2Aub with MINUTE-ChIP[42] (Extended Data Fig. 1a)[42]. Phenotypically, EZH2i-treated naïve and primed hESCs maintained their expected morphology over 7 days (Extended Data Fig. 2a).

[1]Science for Life Laboratory, Department of Medical Biochemistry and Biophysics, Karolinska Institutet, Stockholm, Sweden. [2]Ming Wai Lau Centre for Reparative Medicine, Stockholm node, Karolinska Institutet, Stockholm, Sweden. [3]Department of Clinical Sciences, Intervention and Technology, Karolinska Institutet, Stockholm, Sweden. [4]Division of Obstetrics and Gynecology, Karolinska Universitetssjukhuset, Stockholm, Sweden. [5]Département de Médecine, Université de Montréal, Montreal, Canada. [6]Centre de Recherche du Centre Hospitalier de l'Université de Montréal, Axe Immunopathologie, Montreal, Canada. [7]These authors contributed equally: Banushree Kumar, Carmen Navarro, Nerges Winblad. ✉e-mail: fredrik.lanner@ki.se; simon.elsasser@scilifelab.se

H3K27me3 levels were substantially (~3.3-fold) higher in naïve than in primed hESCs (Fig. 1b), an unexpected observation as H3K27me3 has been reported to be lower in the naïve state[17,43,44]. However, our findings are consistent with quantitative mass spectrometry data comparing naïve and primed hESCs[26], further confirmed by immunofluorescence and immunoblotting (Extended Data Fig. 2b–d). H2Aub signal was also significantly (~2.1-fold) higher in naïve cultures (Extended Data Fig. 1b), suggesting a concerted regulation of PRC1/PRC2 activity. Naïve and primed H3K27me3 levels were depleted by 97% and 92%, after 7-day treatment with EZH2i (Fig. 1b and Extended Data Fig. 2c,d). Immunofluorescence analysis established that H3K27me3 was largely lost between days 2 and 4 and further depleted at day 7 (Extended Data Fig. 2e,f).

We next performed genome-wide enrichment analysis on functionally annotated chromatin states (Fig. 1c and Extended Data Fig. 1c), as well as in 10 kb windows (Fig. 1d and Extended Data Fig. 3a). Despite no substantial change in global levels or at bivalent promoters, as previously defined in the primed state[45] (Fig. 1b,c and Extended Data Figs. 1b and 3a–c), H3K4me3 was increased marginally at active promoters in the naïve state (Fig. 1c and Extended Data Fig. 3b).

Naïve cells had a higher basal level of H3K27me3 across non-polycomb chromatin states and most of the 10 kb bins (Fig. 1c,d and Extended Data Fig. 3a), a phenomenon previously described in naïve mESCs[42,46]. While more than 80% of promoters did not significantly change H3K27me3 status (Fig. 1c and Extended Data Fig. 3b,c), approximately 5% of promoters, including a number of *HOX* genes and other classical bivalent genes, showed reduced H3K27me3 in naïve hESCs (Fig. 1e and Extended Data Fig. 3b, c). Among these, naïve-specific genes such as *KLF4* entirely lost H3K27me3 in the naïve state (Fig. 1e). On the other hand, more than 10% of promoters, for example *CDH2*, *DUSP6* and *OTX2*, strongly gained H3K27me3 (Fig. 1d,e and Extended Data Fig. 3b,c).

**Diffuse gain in H2A ubiquitination in naïve pluripotency.**
PRC1 is thought to act through H3K27me3-dependent and H3K27me3-independent mechanisms in mESCs[47–50]. While genome-wide H2Aub and H3K27me3 levels were reasonably correlated in primed and naïve hESCs, H2Aub did not mirror many of the gains and losses of H3K27me3 (Extended Data Fig. 3b–d). For example, levels of H2Aub at *DUSP6* and *OTX2* genes were similar in naïve and primed states (Fig. 1e).

Intriguingly, the H2Aub landscape was quantitatively maintained in the absence of H3K27me3 (Fig. 1c and Extended Data Fig. 1b). Less than 0.15% of 10 kb bins genome wide showed a significant reduction in H2Aub, including ~0.25% of annotated bivalent promoters[45] (Extended Data Fig. 3e). For example, H2Aub

levels were broadly reduced by 10–30% at *HOX* gene clusters (Fig. 1e), with maximal loss of ~50% at *HOXC5*, *HOXB8* and *HOXB9* promoters. Together, these results suggest that PRC1 recruitment and activity was largely independent of H3K27me3.

**H3K27me3 accumulation on X chromosomes in naïve cells.**
Strikingly, H3K27me3 levels of the naïve X chromosomes were approximately twofold higher than naïve autosomes and approximately fivefold higher than the average signal on chromosomes in the primed state of our naïve female H9 hESCs (Fig. 1f,g and Extended Data Fig. 4a,b). Published data from naïve female hESCs showed a similar enrichment (Extended Data Fig. 4b). At the megabase scale, regions of very high H3K27me3 enrichment alternated with less enriched regions of active transcription along the X chromosomes (Fig. 1g and Extended Data Fig. 4c). Integrating the H3K27me3 signal across chromosomes, we estimated that the X chromosomes carry ~10% of all H3K27me3 marks, more than any autosome (Fig. 1h).

Despite the strong increase in H3K27me3, RNA-seq showed that overall transcriptional output from the X chromosomes relative to autosomes did not change between naïve and primed states, and the distribution of differentially expressed genes was similar to autosomes (Fig. 1i and Extended Data Fig. 4d). A phenomenon termed X chromosome dampening is thought to reduce dosage of the X chromosomes in the naïve state in a mechanism that is distinct from X inactivation and not well understood[51]. Hence, we wondered if H3K27me3 contributed to dosage compensation in naïve hESCs. However, depletion of H3K27me3 did not lead to a global upregulation of transcription from the X chromosomes in naïve hESCs (Fig. 1i and Extended Data Fig. 4e). Even though some genes, such as *VGLL1*, were marked with H3K27me3 in the naïve state and strongly derepressed upon EZH2i treatment, up- and downregulated genes were similar in number and showed a similar fold-change range as those of autosomes (Fig. 1i and Extended Data Fig. 4e,f). Thus, we conclude that H3K27me3 hypermethylation on the naïve X chromosome does not confer chromosome-scale dosage compensation.

**Quantitative ChIP defines naïve-specific bivalent genes.** Since prior definitions of bivalent gene sets were based on H3K27me3/H3K4me3 enrichment in primed hESCs only[17,45], we defined bivalent promoters de novo from our datasets in five categories based on their H3K27me3/H3K4me3 enrichment in primed and/or naïve hESCs (for criteria, see Extended Data Fig. 5a). We called 533 H3K4me3-positive promoters with higher H3K27me3 in the primed state (primed bivalent), 1,551 H3K4me3-positive promoters with higher H3K27me3 in the naïve state (naïve bivalent), 3,403 promoters bivalent in both states (common bivalent) and 12,391

**Fig. 1 | Diffuse H3K27me3 and H2Aub cover the naïve pluripotent genome. a**, Experimental design of MINUTE-ChIP experiment comparing hESCs in naïve and primed, untreated and treated (10 µM EZH2i) culture conditions. Three biological replicates of each condition were barcoded and combined into a single MINUTE-ChIP pool. See Extended Data Fig. 1a for a scheme of the MINUTE-ChIP workflow. **b**, Global levels of H3K27me3 as determined by MINUTE-ChIP input-normalized read counts (INRCs) in naïve or primed hESCs, cultured with or without EZH2 inhibitor. *P* values of pairwise comparisons (two-sided unpaired Student's *t*-test) are given. See Extended Data Fig. 1b for corresponding analysis of H2Aub and H3K4me3. **c**, Histone H3K27me3, H3K4me3 and H2Aub levels by chromatin state (reads per genome coverage, RPGC). See Extended Data Fig. 1c for individual replicates. **d**, Genome-wide analysis of H3K27me3, H2Aub and H3K4me3 levels by 10 kb bins, comparing naïve and primed hESCs. **e**, Genome browser examples of genomic regions with differential occupancy of H3K27me3, H3K4me3 and/or H2Aub. Group-scaled histone modification signals and gain/loss tracks comparing naïve and primed signals are shown. **f**, Chromosome average enrichment of H3K4me3, H3K27me3 and H2Aub in naïve and primed hESCs. Box plot boxes show the 25th and 75th percentile with the median, and whiskers indicate 1.5 times the interquartile range. All individual data points are shown. **g**, Chromosome density plot comparing the X chromosome signals in naïve and primed hESCs to an autosome with similar size (chr7). See Extended Data Fig. 4c for a genome browser example of X chromosome region. **h**, Treemap showing a proportional representation of the total (integrated) amount of H3K27me3 by chromosome (area) and average density (colour intensity). **i**, Density plots of log₂FC in promoter H3K27me3 levels (left) and RNA-seq output (middle) of genes grouped by chromosome, comparing naïve and primed hESCs. Density plots of log₂FC in RNA-seq output of genes grouped by chromosome, comparing untreated and EZH2i-treated naïve hESCs (right). Median values by chromosome are given and indicated as vertical lines in the density plot. In **c–g**, three combined biological replicates are shown.

H3K4me3-positive promoters without H3K27me3 (H3K4me3 only) (Fig. 2a, Extended Data Fig. 5a and Supplementary Table 2).

H3K4me3 was co-enriched at bivalent promoters in both naïve and primed condition but tended to be higher in naïve state (Fig. 2a). Intersecting significant changes in H3K4me3 and H3K27me3 between naïve and primed hESCs, three interesting scenarios

occurred commonly (Fig. 2b): the first set of genes, including *SOX11* and *SIX3*, gained H3K4me3 and lost H3K27me3 in the primed state; a second set of genes, including *KLF4*, *KLF5*, *BMI1*, *TBX3* and *DNMT3L*, gained H3K4me3 and lost H3K27me3 in the naïve state; a third set, including *STS*, *CGB8* and *XAGE2*, had higher levels of both H3K27me3 and H3K4me3 in the naïve state (Fig. 2b).

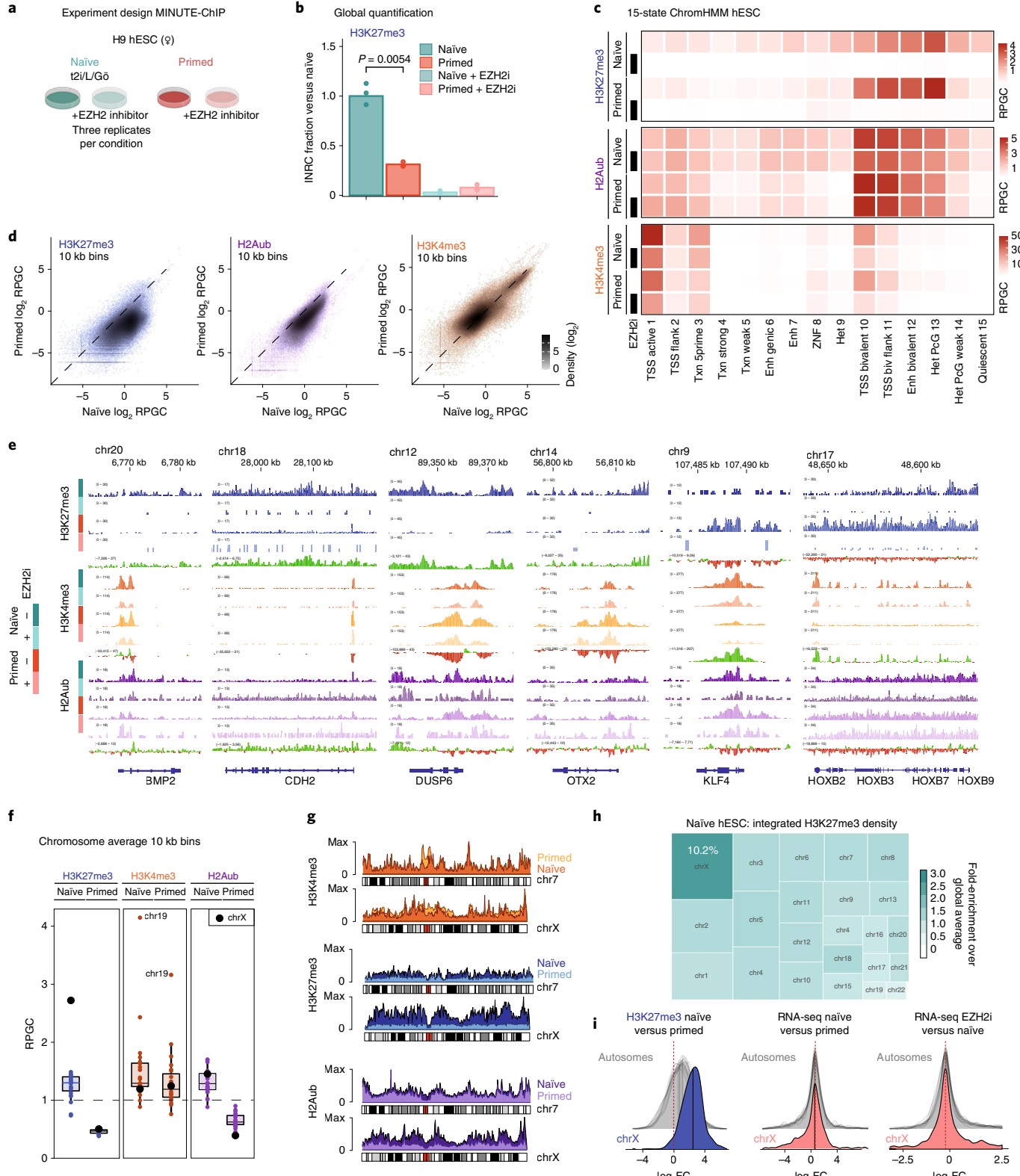

**H3K27me3 repression of bivalent genes in naïve pluripotency.** We next interrogated our RNA-seq datasets to elucidate if H3K27me3 played a functional role in regulating bivalent promoters. Naïve-bivalent genes were indeed predominantly lower expressed ($P = 5 \times 10^{-103}$, Cohen effect size medium) in naïve state (Fig. 2c, left), and primed-bivalent genes were predominantly lower expressed ($P = 3 \times 10^{-49}$, Cohen effect size large) in primed state (Fig. 2c, middle).

As expected, treating naïve hESCs with EZH2i derepressed naïve-bivalent genes ($P = 1 \times 10^{-77}$, Cohen effect size medium), with no significant effect on the expression of primed-bivalent genes (Fig. 2c). Treating primed hESCs with EZH2i derepressed primed-specific genes ($P = 4 \times 10^{-75}$, Cohen effect size large) while naïve-bivalent genes showed a variable response without a significant trend towards derepression (Fig. 2c). EZH2i treatment significantly increased expression of shared bivalent genes in both states ($P \leq 1.36 \times 10^{-149}$, Cohen effect size medium) (Fig. 2c, right).

**H2Aub in hESCs is largely independent of H3K27me3.** The canonical pathway of polycomb repression involves recruitment of canonical PRC1 complex via H3K27me3, following H2A ubiquitination, chromatin compaction and silencing. H2Aub followed similar class-specific trends between naïve and primed states as H3K27me3, albeit within a smaller dynamic range (Fig. 2a), and changed little following H3K27me3 depletion (Fig. 2a). We observed only a weak trend to lose H2Aub (median loss 15%) at derepressed genes (Fig. 2d and Extended Data Fig. 5b). Hence, our data suggest a weak anti-correlation of transcriptional activity and H2Aub at bivalent promoters but does not resolve if the modest reduction in H2Aub is a prerequisite or a consequence of gene activation.

**H3K27me3 acts as a barrier between naïve and primed hESC.** Among the genes switching bivalency status between naïve and primed state (Fig. 2a) were many known marker genes of naïve and primed pluripotency (see also examples above, *KLF4*, *OTX2* and *DUSP6* in Fig. 1e). Another key transcription factor uniquely expressed in naïve hESCs and recently implicated in the establishment of the naïve transcriptional landscape is *TFAP2C*[52]. The promoters of *TFAP2C* gene, as well as the related *TFAP2A* gene, were indeed highly enriched in H3K4me3 and devoid of H3K27me3 in the naïve state, but acquired a bivalent state with intermediate H3K4me3 levels and high H3K27me3 enrichment in the primed state (Extended Data Fig. 5c). Intriguingly, depleting H3K27me3 in the primed state basally activated *TFAP2A/C* transcription, albeit not to the same high level observed in the naïve state. Thus, depletion of H3K27me3 in primed cells removed PRC2-mediated repression but was not able to revert the *TFAP2A/C* genes to a full 'on'

state, presumably because additional activating factors were not expressed in the primed state.

We observed a similar switch from H3K4me3-only promoter status in naïve to bivalent in primed for three additional transcription factors implicated in setting up the naïve pluripotency (*KLF4*, *KLF5* and *TBX3*), and these also increased basal transcription in response to H3K27me3 depletion in the primed state (Fig. 2e). However, the majority of naïve markers, such as *DNMT3L*, *TRIM60* and *ZNF729*, did not accumulate H3K27me3 in the primed state and were also not responsive to EZH2i treatment (Fig. 2e). Most markers of primed pluripotency[53] had H3K4me3-only promoters in the primed state, while assuming a H3K27me3-high/H3K4me3-low bivalent promoter status in the naïve state (Fig. 2b,e). H3K27me3 depletion in the naïve state resulted in increased transcription of most of these genes (including *OTX2*, *CD24* and *DUSP6*), albeit in no case reaching the levels of the primed state (Figs. 1e and 2e). In conclusion, PRC2 appears to establish an epigenetic barrier between the two states through adaptive repression of different gene sets in naïve and primed pluripotency.

**PRC2 represses trophectoderm lineage in naïve cultures.** We next wanted to examine whether bivalent promoters in general may be repressed by H3K27me3 in the naïve state. Differential expression analysis identified 1,894 upregulated and 766 downregulated genes upon EZH2i treatment (Fig. 3a–c). About half of the upregulated genes showed H3K27me3 enrichment in their promoter region (Fig. 3c). A strikingly different transcriptional response of primed cells to EZH2i inhibition (518 up- and 34 downregulated; Extended Data Fig. 5d) highlights the exquisitely context-specific function of PRC2 in repressing largely non-overlapping gene sets in closely related pluripotent states (Fig. 3c). As a number of known developmental genes, including *IGF2*, *FRZB*, *EPAS1* and *GATA2*, were among the upregulated genes in EZH2i-treated naïve cells (Fig. 3a), we evaluated a comprehensive list of lineage marker genes derived from single-cell RNA (scRNA)-seq of the developing human embryo[54] (Fig. 3d). EZH2i treatment of naïve cells reduced marker genes of ICM and epiblast while trophectoderm, amnion and mesoderm markers were upregulated (Fig. 3d). Primed hESCs treated with EZH2i did not show major perturbations for any of these lineages (Extended Data Fig. 5e).

Given the strong upregulation of extra-embryonic lineage markers, we hypothesized that PRC2 shields naïve hESCs from trophectoderm differentiation. Indeed, among the most highly upregulated marker genes were *ENPEP*, the gene for a surface protein (APA) that marks trophectoderm progenitor cells capable of forming syncytiotrophoblast;[9,55] *ABCG2*, a plasma membrane transporter highly expressed in trophectoderm and human placenta;[5,56] *TP63*, a p53 family member defining the cytotrophoblast stem cell

**Fig. 2 | H3K27me3 is adaptive to gene expression changes between naïve and primed pluripotent states and contributes to repression of non-state-specific genes. a**, De novo annotation of bivalent promoters based on DESeq2 analysis. Five promoter classes were defined (for criteria, see Extended Data Fig. 5a): primed bivalent (Pr » Ni), naïve bivalent (Ni » Pr), common bivalent, H3K4me3 only and H3K4me3 negative (not shown). Average H3K27me3, H2Aub and H3K4me3 profile plots (fragments per genome coverage, FPGC) in naïve and primed hESCs for each class and corresponding heat maps for the first three states are shown. Profiles for EZH2i-treated naïve and primed conditions are shown as dashed lines. **b**, Alluvial plot showing H3K27me3 and H3K4me3 gains and losses (DESeq2 adjusted $P < 0.05$, fold change >1.5 from three replicates) at bivalent promoters between naïve and primed hESCs. Select connections are annotated. **c**, Context-specific transcriptional response to global H3K27me3 depletion in different classes of bivalent genes. RNA-seq changes (DESeq2 $\log_2$FC from three replicates each) are plotted, comparing naïve and primed conditions as well as EZH2i treatment and the respective control (left, naïve only; middle, primed only; right, shared bivalent promoters). The distribution of fold changes of all genes is shown as violin plots, and class-specific genes are shown as jitter points. The class-specific group was compared with all genes using a two-sided unpaired Wilcoxon test and Cohen's $d$. **d**, Density plot of fold changes of H2Aub levels following H3K27me3 depletion in hESCs. Only genes that were derepressed upon EZH2i treatment (DESeq2 adjusted $P < 0.05$, fold change >1.5 based on three replicates) were included in the analysis. Bivalent promoters (hence including promoters of the naïve-bivalent and shared class) are compared with H3K27me3-devoid promoters. For an analysis of individual classes, see Extended Data Fig. 5b. **e**, Heat map showing RNA-seq expression ($\log_2$-transformed TPM) of previously defined marker genes for naïve and, primed pluripotency[53] in naïve and primed cultures (± EZH2i treatment), as well as the H3K4me3 and H3K27me3 levels (RPGC) at their respective promoter. RPGC from combined replicates are shown for H3K4me3 and H3K27me3, whereas the individual replicate TPM values are plotted for RNA-seq data. n.s., not significant.

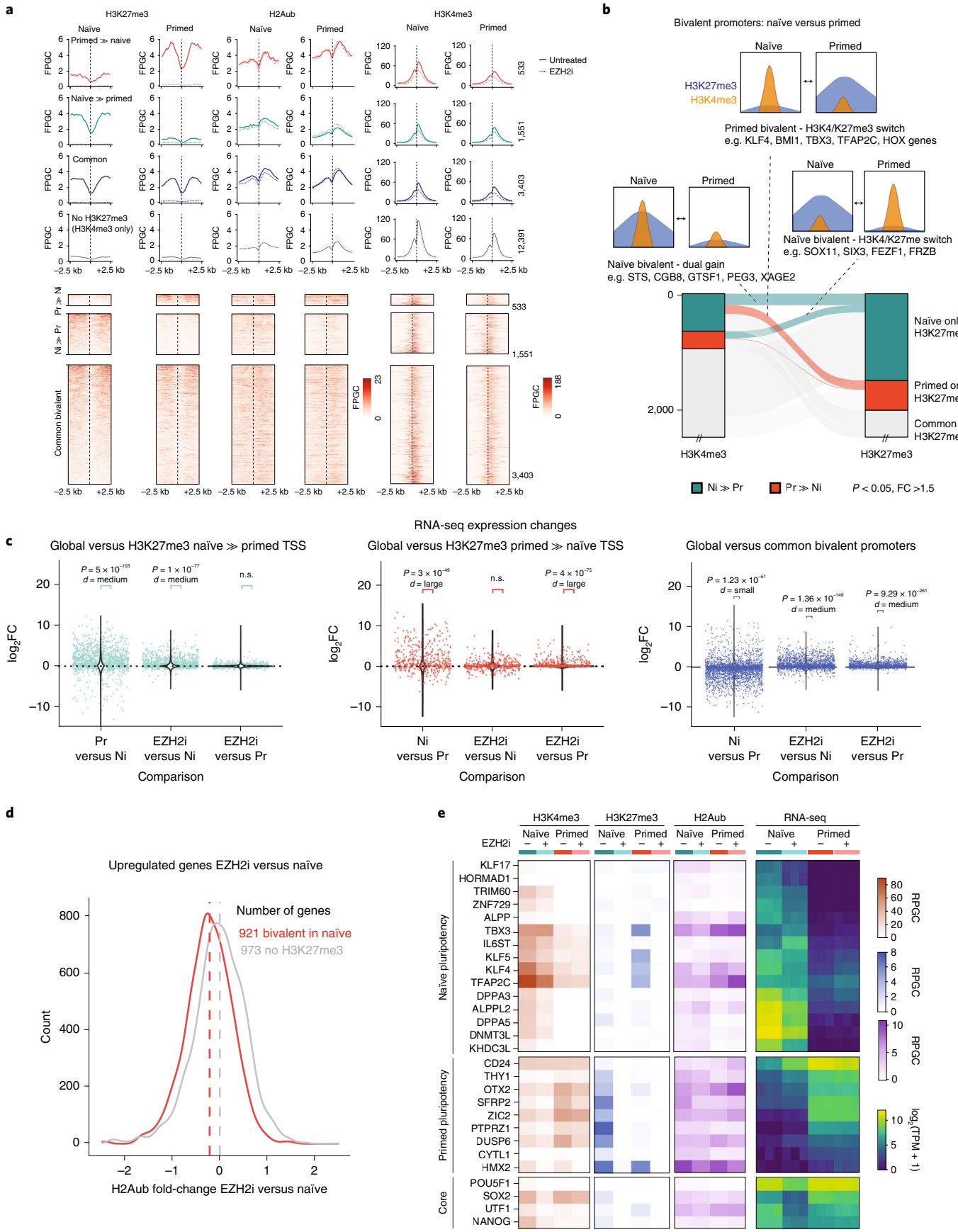

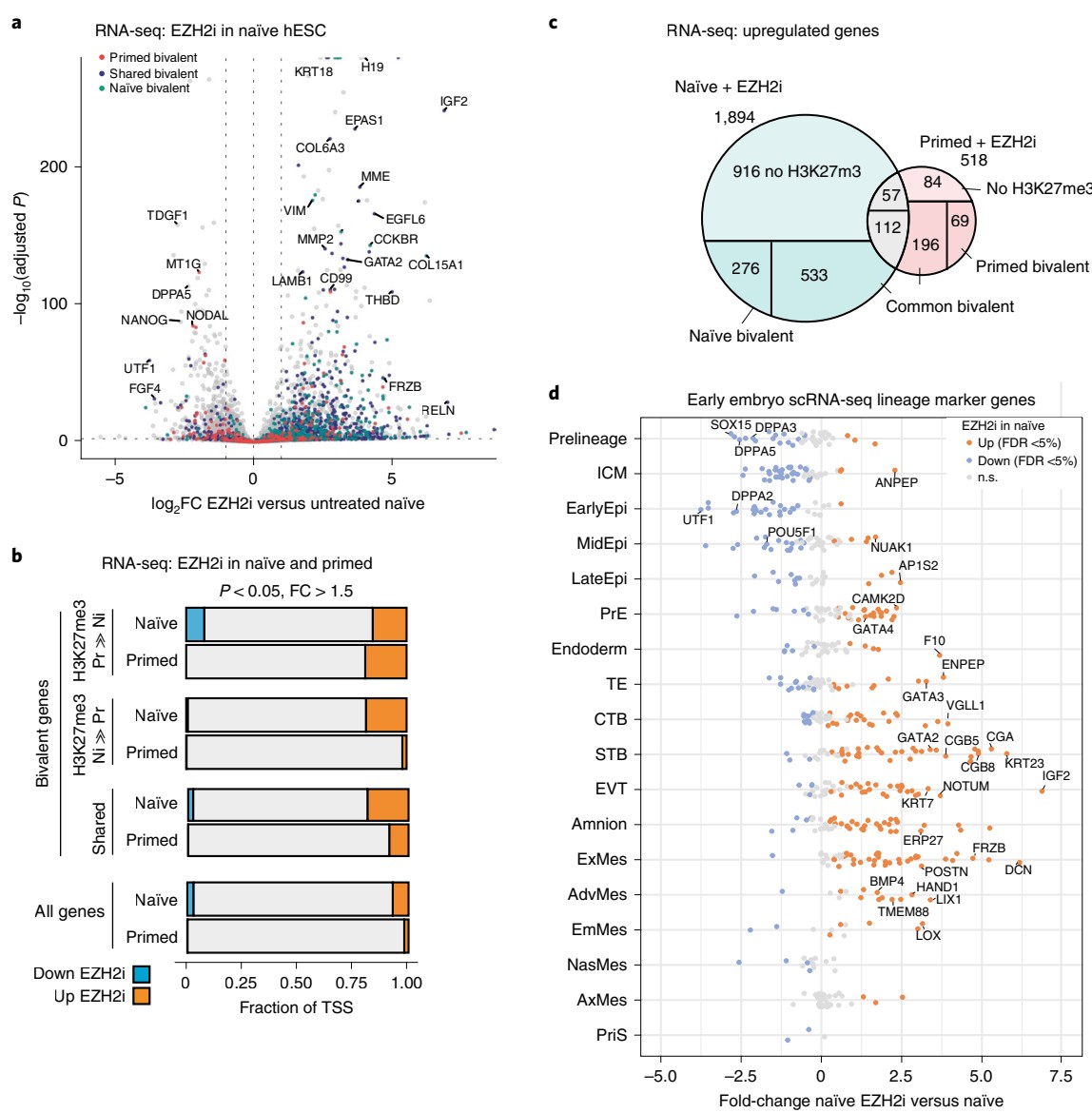

**Fig. 3 | PRC2 inhibition derepresses a naïve-specific subset of bivalent genes. a**, Volcano plot showing differentially expressed genes (DESeq2 FDR <5%, |log₂FC| >1) between 7-day EZH2i-treated naïve hESCs and untreated hESCs. Bivalent gene classes as shown in Fig. 2a are indicated in red (primed bivalent), blue (shared bivalent) and teal (naïve bivalent). See Extended Data Fig. 5d for corresponding analysis in primed hESCs. **b**, Fractions of significant (FDR <5%) transcriptional changes in response to EZH2 inhibitor treatment of naïve and primed hESCs, among all or bivalent promoter classes. **c**, Intersection of genes derepressed after EZH2i treatment in naïve and primed hESCs. Each Venn intersection is further annotated by the H3K27me3 promoter class. **d**, Strip chart showing expression of a comprehensive set of marker genes defined from human embryo single-cell data comparing naïve hESCs and naïve hESCs treated with EZH2i. Markers are grouped into pre-lineage, ICM, epiblast, primitive endoderm (PrE), trophectoderm (TE), cytotrophoblast (CTB), syncytiotrophoblast (STB), extravillous trophoblast (EVT), amnion, extra-embryonic mesoderm (exMes), advanced mesoderm (AdvMes), emerging mesoderm (EmMes), nascent mesoderm (NasMes), axial mesoderm (AxMes) and primitive streak (PriS). Significant differences (DESeq2 FDR <5% from triplicates) are highlighted in blue (downregulated) or orange (upregulated).

compartment[5,57]. Well-known transcription factors specifying trophectoderm lineage, *CDX2*[58,59], *GATA3*[9,59,60], *MSX2*[61] and *NR2F2*[62], were also specifically upregulated in EZH2i-treated naïve hESCs (Fig. 4a and Extended Data Figs. 6a–c). Investigating their promoter status, we noticed that transcription factors *GATA3*, *CDX2*, *MSX2*, *NR2F2*, *GATA2* and *Wnt* agonist *FRZB* and *AKT* agonist *IGF2* all shared bivalent promoters (Fig. 4a and Extended Data Fig. 6a). ChIP–seq in trophoblast progenitors has shown that *GATA3* binds to the promoters of many marker genes of trophectoderm and placental development, including *KRT23*, *VGLL1*, *CGA* and *TP63*[9]. Intersecting with the published binding data[9], we found

20% of upregulated genes in EZH2i-treated naïve cells to be bound by GATA3 (Extended Data Fig. 6b). Genes like *ENPEP*, *KRT23* and *ERP27* lacked H3K4me3 in the naïve state but featured *GATA3* binding sites (Fig. 4a and Extended Data Fig. 6c), suggesting they may be activated downstream of *GATA3*. We sought to confirm *GATA3* induction on the protein level, also including a second PRC2 inhibitor, EED226 (EEDi), which targets the EED subunit[63] in addition to EZH2i. Both inhibitors effectively reduced H3K27me3 at 7 days of treatment (Fig. 4b,c). PRC2 inhibition induced heterogeneity within colonies, with a fraction of cells acquiring GATA3 protein expression (11.9% for EZH2i and 2.6% for EEDi), concomitant with loss

of NANOG (Fig. 4d). Similar induction of GATA3+ cells (6.3%) following EZH2 inhibition was confirmed in an additional naïve hESC line, HS975 (Extended Data Fig. 7a). In contrast, primed H9 hESCs did not show a loss of pluripotency markers or induced *GATA3* expression with EZH2 inhibition (Extended Data Fig. 5f).

As an orthogonal approach to pharmacologic inhibition, we genetically targeted PRC2 using clustered regularly interspaced short palindromic repeats (CRISPR)/Cas9 (Fig. 4e). We electroporated naïve hESCs with synthetic Cas9/guide RNA (gRNA) complex and plated for clonal growth over 7 days (Fig. 4e,f). Targeting with EED gRNAs, we achieved complete loss of EED protein in a large fraction of colonies (Fig. 4g and Extended Data Fig. 7b). While we observed few GATA3-expressing cells within untreated (1.4% GATA3+) and NT-gRNA colonies (1.4% GATA3+), most of the EED-depleted colonies showed GATA3+ cells (total 16.7% GATA3+) (Extended Data Fig. 7b). On the single-cell level, GATA3+ cells were, with few exceptions, low or negative for EED (Fig. 4g).

To better resolve the cellular heterogeneity of the treated naïve cultures, we performed scRNA-seq during EZH2i treatment in naïve and primed cultures. To capture subpopulations and assign their cellular identity, we integrated a comprehensive cellular reference (Fig. 5a and Extended Data Fig. 8a,b), which included data from in vitro cultured blastocysts[23,64], a human stem cell-based post-implantation amniotic sac embryoid (PASE) model[65] and an in vivo gastrulation-stage human embryo specimen[66]. While over 97% of untreated naïve stem cells clustered with the expected pre-implantation epiblast cells of the reference, a small fraction (1.8%) clustered within the trophectoderm reference, suggesting a low but detectable spontaneous differentiation towards trophectoderm lineage in naïve cultures (Fig. 5b). A smaller fraction (1%) clustered within the mesoderm reference (Fig. 5b). This is in agreement with another single-cell study identifying a rare 'intermediate population' in naïve cultures[53], which we confirmed to express trophectoderm and mesoderm markers (Extended Data Fig. 8d). With EZH2 inhibition, the fraction of these differentiated cells progressively reached 9.1% trophectoderm-like cells (TLCs) and 11.7% mesoderm-like cells (MeLCs) after 7 days (Fig. 5b). The MeLCs clustered mainly with the definitive mesoderm but to a lesser degree also to extra-embryonic mesoderm reference of the CS7 gastrula cells (annotated as yolk sac mesoderm). EZH2i treatment of primed cells, on the other hand, produced very few differentiated cells and 99.96% of the cells remained within the epiblast-like cell (ELC) cluster (Fig. 5c). MeLCs generated by our EZH2i treatment strongly expressed markers that also specified the mesoderm lineage within the reference datasets, namely *TMEM88*, *LIX1* and *PMP22* (Fig. 5d). TLCs expressed well-known trophectoderm factors *MSX2*, *GATA2* and *CLDN4* (Fig. 5d). TLCs and MeLCs also shared

some upregulated genes, including *KRT19*, transcription factor *HAND1* and *CD24*, a known surface marker of primed hESCs[25,67] (Fig. 5d). Mapping selected marker genes on our UMAP reference, *TMEM88* and *GATA2* showed unique expression in cells of the MeLC and TLC population, respectively, and were essentially absent in ELCs (Fig. 5e). The well-established transcription factor for primitive streak and mesodermal lineage *TBXT* was induced after 7-day EZH2i treatment in a subset of cells of the ELCs and MeLCs (Fig. 5e). *GATA3* appeared sporadically in ELCs and gained expression in TLCs (Fig. 5e). Although over 75% of the cells remained in the ELC population, our analysis indicated that the majority of cells display a transcriptional shift in the UMAP space with induction of lineage markers following 4 and 7 days of EZH2i treatment (Fig. 5b). Hence, we performed differential gene expression analysis within the ELC population following EZH2i treatment, identifying 482 up- and 387 downregulated genes (Extended Data Fig. 8e,f and Supplementary Table 4). Among the significantly upregulated genes were transcription factors *HAND1*, *CDX1*, *CDX2*, *TBXT*, *TBX3*, *TFAP2A*, *SOX9*, *MEIS2* and *TWIST1*, (Fig. 5f,g), many of which were classical bivalent in naïve and primed hESCs. They were not expressed in primed hESCs, and gained additional H3K4me3 and low basal expression in naïve hESCs (Extended Data Fig. 9a,b).

**An 'activated' naïve state precedes lineage branching.** To further describe how naïve hESCs exit from pluripotency, we performed a trajectory analysis using 864 differentially expressed genes across the different naïve hESC-derived populations and EZH2i time course. The analysis yielded a bifurcated trajectory with three principal states (Extended Data Fig. 9c), on which naïve hESCs first transition towards a bifurcation point from which two independent cell fates, trophectoderm and mesoderm, are accessed (Fig. 6a). Mapping our reference annotation (Fig. 5a,b) onto the trajectory, we were able to resolve a lineage bifurcation point already within the ELC population (Extended Data Fig. 9d). We subclassified cells within the ELC population by their position on the trajectory and pseudotime axis as follows: we termed the starting population 'ground state' ELC (gELC) and the trajectory leading up to the bifurcation point 'activated' ELC (aELC). We termed the ELC branch towards MeLC mesoderm-activated ELC (MaELC), and the other trophectoderm-activated ELC (TaELC) (Fig. 6c). In total, 91.4% of untreated epiblast-like naïve hESCs resided in the ground state (Fig. 6c,d). Intriguingly, 2.5% and 6.1% of single cells within the untreated naïve population also populated the activated and lineage states, respectively (Fig. 6c,d). This differentiation accelerated over 2 and 4 days of EZH2i treatment, and by 7 days 66.2% ELC cells were activated and branched into the two lineages at approximately even proportions (Fig. 6c,d). Interestingly, the fraction of aELCs

**Fig. 4 | Loss of H3K27me3 in naïve hESCs activates trophectoderm gene expression programmes. a**, Heat map showing expression (TPM) in naïve and primed hESCs (± EZH2i treatment), as well as H3K4me3, H3K27me3 and H2Aub promoter status of selected trophectoderm and placenta-specific genes. RPGC from combined replicates are used for H3K4me3 and H3K27me3, whereas the three individual replicate TPM values are plotted for RNA-seq data. GATA3 binding as determined by ChIP–seq peaks during trophectoderm differentiation[9] is indicated. For genome browser examples, see Extended Data Fig. 6a,c. **b**, Immunofluorescence confocal microscopy images of naïve H9 hESC colonies without treatment or with EZH2i (EPZ-6438) or EEDi (EED226) treatment for 7 days, assessing H3K27me3 levels, expression of pluripotency marker NANOG and trophectoderm transcription factor GATA3. Merged images show NANOG in purple, H3K27me3 in red, GATA3 in green, and Hoechst in blue. Scale bars, 10 μm. Data shown represent three independent experiments for EZH2i and two independent experiments for EEDi. For analogous experiment in naïve HS975 hESCs, see Extended Data Fig. 7a. **c,d**, CellProfiler image analysis of the experiment described in **b**. Per-nucleus H3K27me3 and GATA3 immunofluorescence intensities of a total of 670 nuclei derived from one experiment (165 untreated, 235 EZH2i and 270 EEDi). Box plot boxes show the 25th and 75th percentile with the median, and whiskers indicate 1.5 times the interquartile range. P values are determined by two-sided unpaired t-test (H3K27me3 $P < 2.2 \times 10^{-16}$ (top), GATA3 $P = 2.4 \times 10^{-5}$ for EZH2i and $P = 0.046$ for EEDi treatment). The threshold for GATA3+ cells (dashed line) is defined as 1.5× mean of untreated cells. Scatter plots contrasting per-nucleus GATA3 with H3K27me3 or NANOG immunofluorescence intensities. **e**, Scheme of experiment for CRISPR/Cas9-mediated acute deletion of EED. **f**, Representative image of experiment described in **e**. Example shown represents two independent experiments. Scale bars, 20 μm. **g**, CellProfiler image quantification of the experiment described in **e**. Scatter plots contrasting per-nucleus GATA3 and H3K27me3 immunofluorescence intensities of untransfected cells, NT gRNA or EED-targeting gRNA (EED gRNA) transfected cells. In total, 4,312 nuclei from two independent experiments were analysed. For statistics by individual colonies, see Extended Data Fig. 7b.

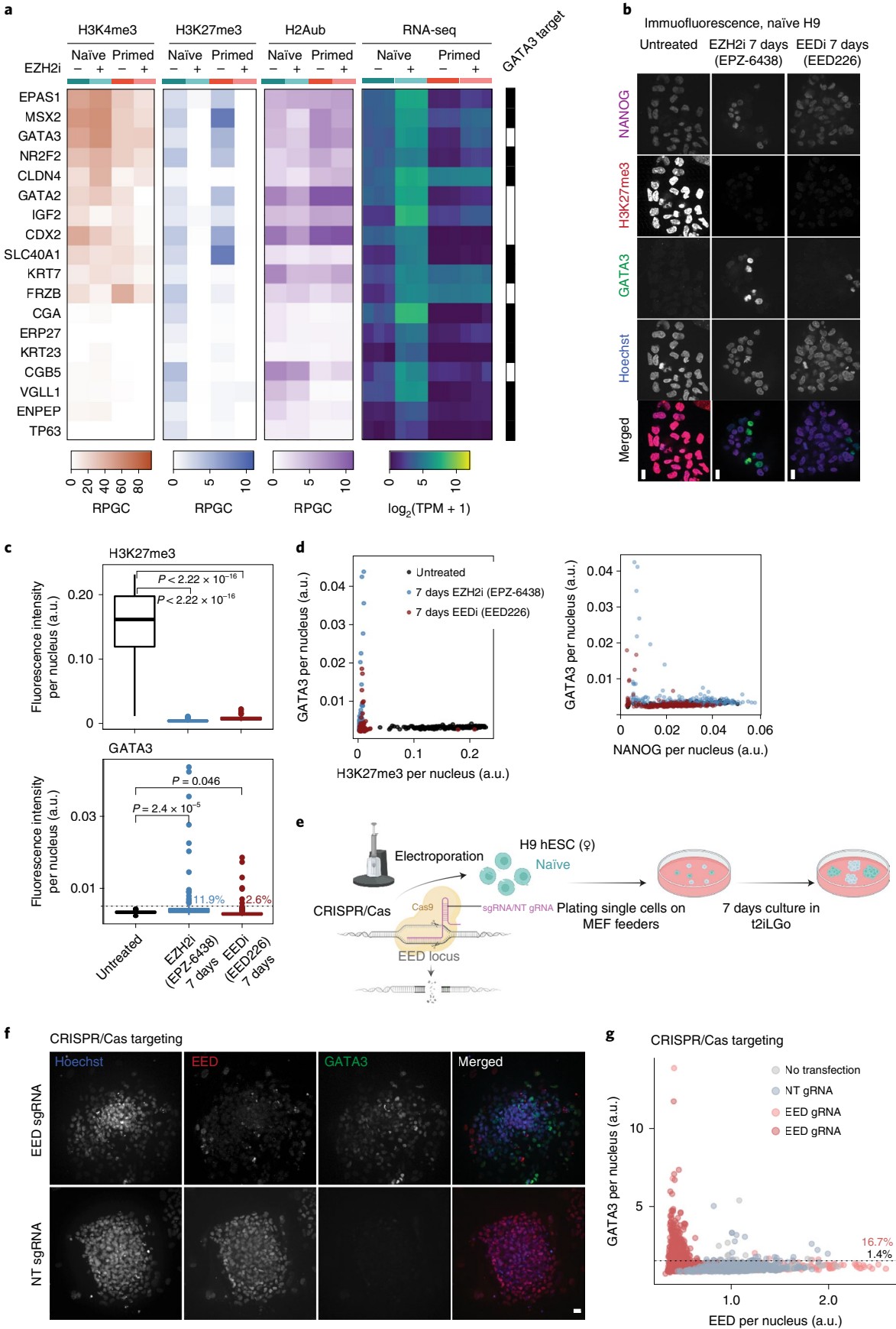

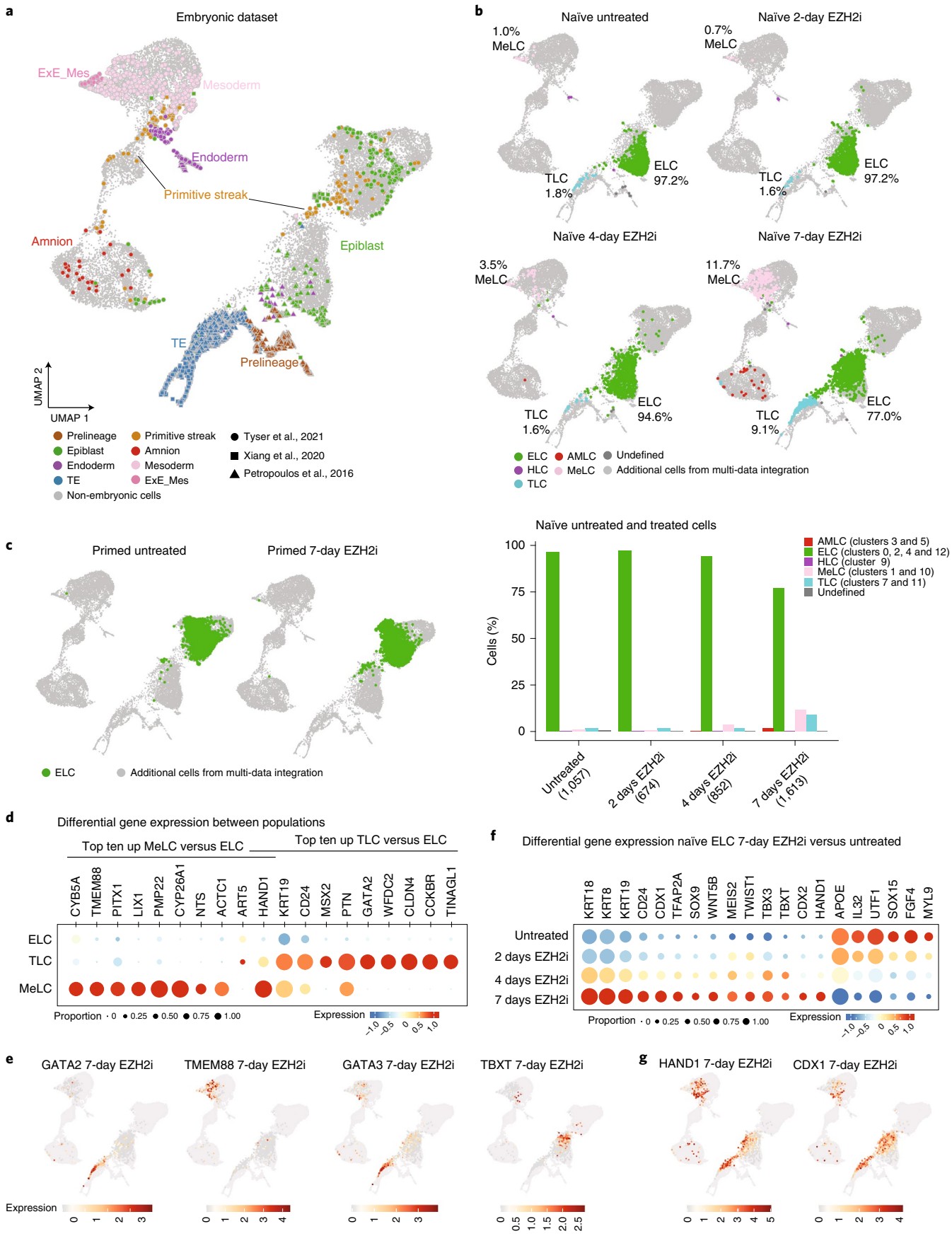

**a** Embryonic dataset

ExE_Mes / Mesoderm / Endoderm / Primitive streak / Amnion / Epiblast / TE / Prelineage

UMAP 2 / UMAP 1

Prelineage / Primitive streak / Epiblast / Amnion / Endoderm / Mesoderm / TE / ExE_Mes / Non-embryonic cells
Tyser et al., 2021 / Xiang et al., 2020 / Petropoulos et al., 2016

**b** Naïve untreated / Naïve 2-day EZH2i / Naïve 4-day EZH2i / Naïve 7-day EZH2i

1.0% MeLC / 0.7% MeLC / 3.5% MeLC / 11.7% MeLC
TLC 1.8% / ELC 97.2% / TLC 1.6% / ELC 97.2% / TLC 1.6% / ELC 94.6% / TLC 9.1% / ELC 77.0%

ELC / HLC / TLC / AMLC / MeLC / Undefined / Additional cells from multi-data integration

**c** Primed untreated / Primed 7-day EZH2i

ELC / Additional cells from multi-data integration

Naïve untreated and treated cells

Cells (%)

AMLC (clusters 3 and 5) / ELC (clusters 0, 2, 4 and 12) / HLC (cluster 9) / MeLC (clusters 1 and 10) / TLC (clusters 7 and 11) / Undefined

Untreated (1,057) / 2 days EZH2i (674) / 4 days EZH2i (852) / 7 days EZH2i (1,613)

**d** Differential gene expression between populations

Top ten up MeLC versus ELC / Top ten up TLC versus ELC

CYB5A / TMEM88 / PITX1 / LIX1 / PMP22 / CYP26A1 / NTS / ACTC1 / ART5 / HAND1 / KRT19 / CD24 / MSX2 / PTN / GATA2 / WFDC2 / CLDN4 / CCKBR / TINAGL1

ELC / TLC / MeLC

Proportion 0 0.25 0.50 0.75 1.00 / Expression -1.0 -0.5 0 0.5 1.0

**f** Differential gene expression naïve ELC 7-day EZH2i versus untreated

KRT18 / KRT8 / KRT19 / CD24 / CDX1 / TFAP2A / SOX9 / WNT5B / MEIS2 / TWIST1 / TBX3 / TBXT / CDX2 / HAND1 / APOE / IL32 / UTF1 / SOX15 / FGF4 / MYL9

Untreated / 2 days EZH2i / 4 days EZH2i / 7 days EZH2i

Proportion 0 0.25 0.50 0.75 1.00 / Expression -1.0 -0.5 0 0.5 1.0

**e** GATA2 7-day EZH2i / TMEM88 7-day EZH2i / GATA3 7-day EZH2i / TBXT 7-day EZH2i

Expression 0 1 2 3 / 0 1 2 3 4 / 0 1 2 3 / 0.5 1.0 1.5 2.0 2.5

**g** HAND1 7-day EZH2i / CDX1 7-day EZH2i

0 1 2 3 4 5 / 0 1 2 3 4

**Fig. 5 | Single-cell transcriptomic profiling of EZH2i-treated naïve and primed cells. a**, UMAP projection reference annotation (TE: trophectoderm; Exe_Mes: extra-embryonic mesoderm). The colour of each data point represents the cell annotations retrieved for each publication. Light-grey-coloured data points represent cells in the multi-data integration that were not used to infer cell annotations. The shape of data points for the embryonic cells indicates the data source. **b** UMAP projection and cell identity annotations of single-cell transcriptome datasets obtained for EZH2i-treated naïve and primed cells at indicated timepoint, showing ELCs, TLCs, MeLCs, AMLCs and HLCs. Percentage of cells within major clusters are indicated in the maps. Bar chart (bottom) shows the percentage of cells per cluster at each time point. **c**, UMAP projection and cell identity annotations as in **b** for primed hESCs treated with EZH2i. In **a–c**, UMAP was generated from an extensive reference dataset, including data from human embryonic and PASE model (Extended Data Fig. 8a,b and Methods). **d**, Top ten significantly upregulated genes in MeLCs or TLCs, compared with ELC population. Bubble plot shows proportion of expressing cells and average expression per gene and condition. **e**, Single-cell expression of select TLC and MeLC markers mapped onto UMAP. log₂ expression of selected genes overlaid on the UMAP. **f**, Selected genes differentially expressed within ELC population comparing 7-day-EZH2i-treated samples and untreated samples. Bubble plot shows proportion of expressing cells and average expression per gene and condition. **g**, Single-cell log₂ expression of select factors mapped onto UMAP.

plateaued at 16–18% after 4 and 7 days of EZH2i treatment, indicating that single cells shift through this state only transiently towards a stable, lineage-committed end state.

As the trajectory analysis allowed us to separate early (generation of aELC) and late (bifurcation into TaELC and MaELC) steps in the differentiation process, we wanted to further dissect the trajectory underlying activation and lineage choice. First, we performed differential expression analysis comparing aELC and gELC to understand the process of activation. We identified 24 upregulated and 27 downregulated genes in the aELC, including *CDX1*, *HAND1*, *TBXT* and *WNT5B*, all key genes involved in streak and mesoderm induction (Fig. 6e and Supplementary Table 4).

Differential gene expression analysis comparing TaELC and MaELC identified 67 mesoderm-enriched and 27 trophectoderm-enriched genes (Fig. 6f, false discovery rate (FDR) <5%; Supplementary Table 4). For bifurcation towards mesoderm, we identified secreted factors *LEFTY1*, *LEFTY2*, *NODAL* and *WNT5B*, implying involvement of several signalling pathways, together with transcription factors *TBXT*, *ID1*, *MIXL1* and *UTF1*. Towards trophectoderm, we confirmed core trophectoderm transcription factor *TFAP2C*, and additionally found surface proteins *CD24* and *CDH1*, as well as tight junction protein *CLDN4*, among the top differentially expressed genes (Fig. 6f,g and Supplementary Table 4).

Finally, we wondered if we could delineate the definitive exit from pluripotency within our trajectory analysis. In agreement with population-wide downregulation of pluripotency factors already observed in bulk RNA-seq (Fig. 3a), we observed a continuous decline of *NANOG* and *SOX2*, while *POU5F1* continued to be highly expressed throughout aELC, steeply dropping only after the bifurcation point (Fig. 6g). Together, this characterizes the activated ELC population as a state of opposing pluripotency and differentiation cues, from which more than one lineage trajectory remains accessed.

## Discussion
Capturing quantitative differences between epigenomic profiles, as achieved with MINUTE-ChIP, has allowed us to uncover a more complex pattern of gains and losses of H3K27me3 between naïve and primed states than previously appreciated[14,16,17,44]. Importantly, we found that bivalency was largely maintained in

the naïve state and identified hundreds of promoters that gained bivalency uniquely in the naïve state. Surprisingly, despite the extensive overlap of target promoters between naïve and primed states, inhibiting PRC2 catalytic activity activated independent gene sets, characteristic for each state. Hence, H3K27me3 per se cannot be attributed a constitutive repressive function at bivalent promoters; instead, we view bivalency as a default state of many developmental genes that can optionally acquire a transcriptionally poised state (that is, a state of low transcription in which solely H3K27me3 impedes activation) in the presence of cell-type- or lineage-specific activators.

We identified a network of transcription factors that may act as such activators and appear to cooperatively mediate activation and subsequent lineage commitment. These include factors already expressed in naïve pluripotency (TFAP2C, LEFTY2 and NODAL), and transcription factors, including TBXT, CDX1, HAND1 and GATA3, that drive the transition towards an activated state and lineage commitment. We observed the activated state to be populated with low frequency in naïve hESC cultures, and the transition may be reversible. However, a committing step may be reached at the bifurcation of mesoderm and trophectoderm lineages. Given the widespread promoter bivalency in naïve pluripotent cells, it is plausible that PRC2 basally represses early transcription factors and their targets, hence effectively intercepting extrinsic stimulatory signals (Wnt and Smad signalling) and positive feedback between the transcription factors itself. Recent studies highlight how multi-lineage transcription factors, including GATA3, TFAP2A/C, HAND1, TBXT and CDX1, cooperate in driving exit from primed pluripotency[9,60].

By inhibiting or genetically ablating PRC2, we demonstrated an important function for H3K27me3 in counteracting the action of potent lineage-specifying factors in the pluripotent state. We identified trophectoderm and mesoderm as the endpoints of a differentiation invoked in the absence of H3K27me3. We traced the trajectory towards these lineages, including a bifurcation point at which a choice is made towards one of the two possible fates. Our data demonstrate an intrinsic propensity of naïve hESC cultures to produce bona fide trophectoderm and mesoderm lineages through defined differentiation trajectories and assign a previously undescribed role to PRC2 in shielding naïve pluripotency.

**Fig. 6 | Trajectory inference and gene expression dynamics for EZH2i-treated naïve hESCs. a,b**, Trajectory inference colour-coded by inferred pseudotime and cell types. **c**, Further subclassification of ELC cells based on cell state and pseudotime distribution: gELC, aELC, TaELC and MaELC. Ridge plot (bottom) shows distribution of single cells along the pseudotime axis for naïve hESCs untreated or treated for 2, 4 and 7 days with EZH2i. Two dashed lines on ridge plot represent the pseudotime for transition from gELC to aELC, and bifurcation into MaELC and TaELC, respectively. **d**, Cells from untreated and EZH2i-treated (2, 4 and 7 days) naïve hESCs mapped onto trajectory. **e**, Volcano plot showing differentially expressed genes (24 up and 27 down) comparing aELC and gELC population (combined data from 4-day and 7-day EZH2i treatment of naïve hESCs). **f**, Volcano plot showing differentially expressed genes comparing mesoderm (MaELC, 67 genes) and trophectoderm (TaELC, 27 genes) branching at bifurcation point in 7-day-EZH2i treated naïve hESCs. **g**, Expression dynamics (pseudotime) of selected genes during EZH2i treatment on naïve cells. The confidence interval (95%) is indicated by bandwidth. MeLC branch and TLC branch are indicated by pink and blue lines, respectively.

Our findings contrast in several aspects with earlier reports that PRC2 is dispensable for maintenance of naïve pluripotency in 5iLAF and t2iLGö media[31,32]. Shan et al.[31] have reported that *NANOG*, *OCT4* and *SOX2* levels are unaffected in naïve EZH2[−/−] hESCs with additional short hairpin RNA-mediated depletion of EZH1. However, the remaining EZH1 or H3K27me3 levels were not

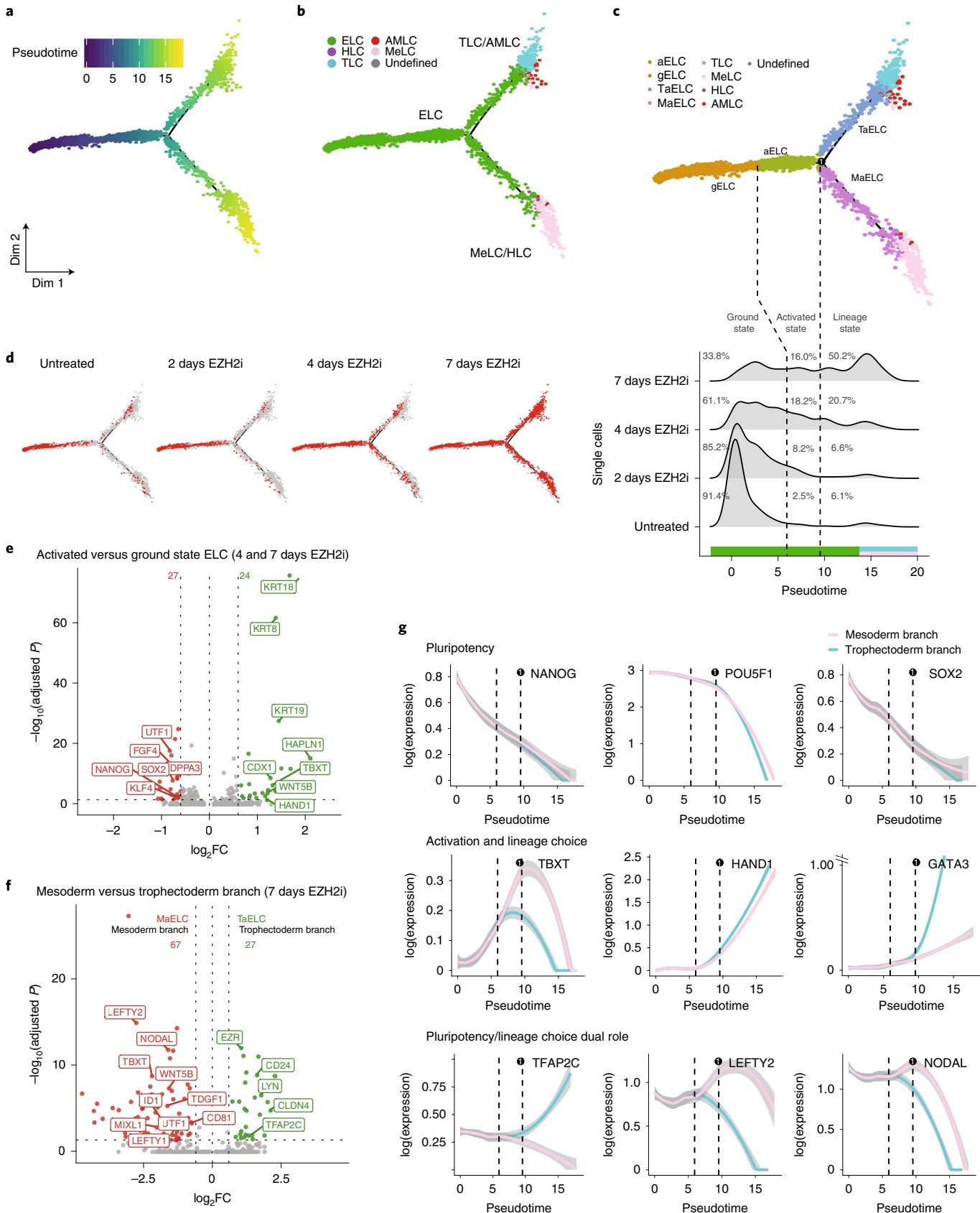

explicitly assessed, leaving the possibility that our EZH2i treatment and genetic targeting resulted in a greater depletion of H3K27me3 and, thus, a more pronounced phenotype. Of note, the EZH2 inhibitor EPZ-6438 also inhibits EZH1 with nanomolar IC50[68], probably achieving complete inhibition of EZH1 and EZH2 at the 10 μM concentration used here (Figs. 1b and 2a), and further experiments with pharmacologic inhibition and genetic ablation of EED, the core subunit of PRC2, corroborate our conclusion that PRC2 is indispensable to prevent differentiation of naïve hESC cultures in t2iLGö. In another study published in this issue, Zijlmans et. al.[69] recapitulate key findings regarding the rewiring of bivalent promoters in naïve hESCs, but did not observe a differentiation response to PRC2 inhibition in naïve hESCs cultured in PXGL medium. This suggests that culture in PXGL medium alleviates the requirement for PRC2 to maintain naïve pluripotency. To test this hypothesis, we performed a direct comparison of PXGL and t2iLGö cultures and observed 0.2% GATA3+ cells after EZH2i treatment in PXGL versus 8.3% in t2iLGö (Extended Data Fig. 7c). Providing further rationale for this differences in response, the Tankyrase/*WNT* pathway inhibitor XAV939 contained in PXGL medium has been shown to reduce basal expression of GATA3 and TBXT in naïve hESCs[70].

Genetic ablation of PRC2 in primed hESCs results in meso-endoderm development rather than trophectoderm development[31,33]. It is important to note that the process of exiting primed pluripotency and meso-endoderm speciation was not a rapid response to depletion of PRC2 in primed hESCs, but emerged over 3–4 weeks of culture[31,33]. Therefore, the transcriptional rewiring and appearance of TLCs and MeLCs within 7 days of inhibiting PRC2 in naïve hESCs (Fig. 5a) is in striking contrast to the slow exit from pluripotency of primed hESCs. This is confirmed by our control experiments treating primed hESCs with EZH2i (Fig. 5c), which did not detect any differentiation in the 7-day window. Together with the observations that inhibition of PRC2 enhances directed differentiation towards trophectoderm[69], our data suggest that the naïve state is uniquely poised towards trophectoderm and mesoderm lineages through a unique bivalent gene signature of key developmental regulators.

In summary, our quantitative comparison of epigenetic landscapes in naïve and primed hESCs reveals an extensive rewiring of promoter bivalency in which PRC2-mediated H3K27me3 constitutes an important epigenetic barrier that dynamically follows and reinforces lineage choices and developmental progression. It will be an exciting focus of future work to elucidate if and how PRC2 is involved in guiding the totipotency-to-pluripotency transition in the early human embryo.

## Online content

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

## Methods

**Culture of hESCs.** Primed female H9 hESCs (Wicell, WA09) p28(11) were thawed using Nutristem hESC XF medium (Biological Industries, 05–100–1 A) and cultured on tissue culture plates (Sarstedt, 83.3922) pre-coated with 10% LN521 (Biolamina, LN521-02). Passaging was done when confluency reached 70–80%. Naïve H9 p28(8) naïve p7 that had been previously converted to the naïve stem cell state using NaïveCult induction medium (Stemcell Technologies, 05580) were thawed onto plates with high-density (strain ICR) inactivated mouse embryonic fibroblasts (MEFs; Gibco, A24903) using NaïveCult expansion medium (t2iLGö; Stemcell Technologies, 05590) supplemented with 10 μM ROCKi (Merck, Y-27632). Twenty-four hours after thawing, medium was changed to fresh NaïveCult expansion medium. Passaging was done every 4–5 days with medium supplemented with 10 μM ROCKi. Naïve HS975[71] p35(17) cells were cultured in 5iLAF medium[17] on inactivated MEFs (Gibco, A24903) under ethical permit 2011/745:31/3 issued by the Swedish Ethical Review Authority. All hESCs were cultured at 37 °C, 5% $O_2$ and 5% $CO_2$.

*EZH2i treatment of H9 naïve and primed hESCs.* For EZH2i treatment, primed and naïve cells were grown in respective media, supplemented with 10 μM EZSolution EPZ-6438 (BioVision, 2428-5) for 7 days, with daily medium changes. All cells were passage on d3-4 counting from start of treatment. H9 naïve cells were also maintained in PXGL[70] consisting of N2B27 basal medium supplemented with 1 μM PD0325901, 2 μM XAV939, 2 μM Gö6983 and 10 ng ml[−1] human leukaemia inhibitory factor, with or without EZH2i for 7 days before being fixed with 4% paraformaldehyde (FF-Chemicals, FFCHFF22023000) and used for immunofluorescence staining. For sample collection, primed and naïve H9 were dissociated using TrypLE Select (Gibco, 12563011). Primed cells were washed once with phosphate-buffered saline (PBS; Sigma, D8537) and counted using Moxi Z mini automated cell counter (Orflo, MXZ001). Collected naïve cells were centrifuged at 300g for 4 min, resuspended in fresh NaïveCult expansion medium and kept on ice. Mouse feeder removal microbeads (Milteny Biotech, 130-095-531) were used according to the manufacturer's protocol to reduce the amount of MEFs in the naïve cell samples. In brief, naïve cells were mixed with the microbeads and incubated at 4 °C for 15 min. Meanwhile, the columns were equilibrated with NaïveCult medium. The cell–microbead suspension was added to the column mounted onto a magnet, allowing unbound naïve cells to pass through into a collection tube. Columns were rinsed using NaïveCult medium, and the flow-through was collected into the same collection tube.

*EEDi treatment of H9 naïve hESC.* Naïve H9 hESCs were maintained in NaïveCult expansion medium supplemented with 10 μM EED226 (Cayman Chemical, CAYM22031-10) or 2.5 μM UNC1999 (Sigma, SML0778). The latter was also applied to naïve H9 hESCs maintained in PXGL medium. Treatments lasted for 7 days, and the medium was changed daily. All cells were passaged on d3-4 counting from the start of treatment. On day 7, cells were fixed using 4% paraformaldehyde (FF-Chemicals, FFCHFF22023000) for 10 min. Samples were left in DPBS[−/−] (Gibco, 14190144) at 4 °C until proceeding with the immunofluorescence protocol.

*EED gRNA and NT gRNA targeting in H9 naïve cells.* Naïve cells were cultured as detailed above. MEF cells were plated 24 h before electroporation. On the day of electroporation, MEF wells were rinsed with DPBS[−/−] (Gibco, 14190144) before 2 ml fresh NaïveCult expansion medium supplemented with 10 μM ROCKi was added to the wells. Naïve colonies were dissociated into a single-cell suspension using TrypLE select and counted using the small particle setting on Moxi Z mini automated cell counter. The cell solution was passed through a 40 μm cell strainer (VWR, 732-2757) before it was centrifuged at 300g for 4 min. Meanwhile, 0.5 μl TrueCut Cas9 Protein v2 (Invitrogen, A36496) was complexed with 1 μl 100 μM EED or non-targeting (NT) gRNA (both from Synthego; GeneKnockout kit V2, GKO-HS1-000-0-1.5n-0-0), respectively, for 10–15 min. Supernatant was removed and the cell pellet was washed in PBS[−/−] and centrifuged at 300g for 4 min. The cell pellet was resuspended in the appropriate Buffer R volume as described in the manufacturer's protocol. Cells were distributed into the tubes containing the ribonucleoprotein mixtures and briefly vortexed. Neon electroporator (Invitrogen, MPK5000) settings were: 1,400 V, 20 ms and 1 pulse. Medium was changed 24 h post-transfection, and daily medium changes followed until 7 days post-transfection when cells were fixed using 4% paraformaldehyde (FF-Chemicals, FFCHFF22023000). In the first electroporation experiment untransfected cells were used as control, and in the second electroporation experiment NT gRNA electroporated cells were used as control.

**ChIP–seq.** Triplicate pellets of $1 \times 10^6$ cells were collected for all conditions, flash frozen and stored at −80 °C before use. Mouse feeder removal microbeads (Milteny Biotech, 130-095-531) were used according to the manufacturer's protocol to reduce the amount of MEFs in the naïve cell samples. Samples were prepared for ChIP–seq following the MINUTE-ChIP protocol (https://doi.org/10.17504/protocols.io.8nkhvcw)[42]. Briefly, native cell pellets were lysed, MNase digested to mono- to tri-nucleosome fragments and ligated with double-stranded DNA adaptors (containing T7 promoter, 8 bp sample barcode and a 6 bp unique molecular identifier (UMI)) in a one-pot reaction. Barcoded samples were then pooled and aliquoted into individual ChIP reactions with Protein A/G magnetic beads (Bio-Rad, 161- 4013/23) coupled with the desired antibodies (5 μg each of H3K27me3 (Millipore, 07-449), H3K4me3 (Millipore, 04-745) and H2AUb (Cell Signaling, 8240 S)). Upon incubation for 4 h at 4 °C with rotation and washing steps, ChIP DNA was isolated and set up in sequential reactions of in vitro transcription, RNA 3′ adapter ligation, reverse transcription and PCR amplification to generate final libraries for each ChIP (Extended Data Fig. 1a). After quality assessment and concentration estimation, libraries were diluted to 4 nM, combined and sequenced on the Illumina NextSeq500 platform with paired-end settings.

**Bulk RNA-seq.** Per condition, $1 \times 10^6$ cells were collected, resuspended in Buffer RLT (Qiagen, 74106) and spiked in with $5 \times 10^4$ *Drosophila* cells per sample. Then, total RNA was extracted using the RNeasy Plus Mini Kit (Qiagen, 74136) according to the manufacturer's protocol. Mouse feeder removal microbeads (Milteny Biotech, 130-095-531) were used according to the manufacturer's protocol to reduce the amount of MEFs in the naïve cell samples. Purified RNA quantities were estimated using the Qubit RNA HS assay kit (Life Technologies, Q32852), and the samples were subsequently flash frozen. RNA-seq libraries were generated and sequenced through BGI service (www.bgi.com) for strand-specific RNA-seq with poly(A) selection (DNBseq Eukaryotic Transcriptome De novo Sequencing).

**scRNA-seq.** *Sample preparation.* H9 naïve cells were treated with EZH2i for 2, 4 or 7 days and collected along with untreated control cells. Primed H9 cells were treated with the same inhibitor for 7 days and collected along with untreated control cells. Mouse feeder removal microbeads (Milteny Biotech, 130-095-531) were used according to the manufacturer's protocol to reduce the amount of MEFs in the naïve cell samples.

*Cell Multiplexing Oligo labelling (Cellplex).* Samples EZH2i d2, d4 and untreated control naïve as well as EZH2i d7 and untreated control primed cells were multiplexed following the manufacturer's protocol. In brief, cells were dissociated, treated with mouse feeder removal beads (naïve cells), counted and resuspended in Cell Multiplexing Oligo. After oligo incubation, cells were washed three times using DPBS[−/−] containing 1% BSA (Sigma Aldrich, A7284) and counted using the NucleoCounter NC-200 automated cell counter. At this point, cells were pooled accordingly: EZH2i d2, d4 and untreated naïve cells, and EZH2i d7 and untreated control primed cells, respectively, and each pool was counted once more.

*Chromium Next GEM Single Cell 3′ Reagent Kits v3.1 (dual index).* Complementary DNA preparation and library construction was done according to the manufacturer's detailed protocol. Briefly, the appropriate number of cells, as defined by the protocol, were complexed with gel beads in emulsion and barcoded. After cleanup, cDNA was amplified and purified. For the samples that were multiplexed, the supernatant as well as the pellet were purified, whereas for the single samples only the pellet was purified. Samples were analysed using TapeStation High Sensitivity 5000 at the Bioinformatics and Expression Analysis core facility at Karolinska Institutet. Following the quality control results, the samples went through fragmentation, end repair and A-tailing followed by adaptor ligation and index PCR. The transferred supernatant from the multiplexed samples was indexed through a sample index PCR. Once samples were purified and eluted in Buffer EB (Qiagen, 19086), they were analysed using TapeStation High Sensitivity 1000 at the Bioinformatics and Expression Analysis facility. Finally, cDNA libraries were sequenced using Illumina Nextseq 2000 Platform 100 cycles P2 and Illumina Nextseq 550 High Output Kit v2.5 (150 cycles).

**Immunofluorescence.** Samples fixed with 4% paraformaldehyde as described above and permeabilized using 0.3% Triton X-100 (Sigma Aldrich, T9284-100ML) in PBS (Gibco, 14190144) for 10 min, after which three washes with PBS were carried out. The samples were blocked for 2 h using 0.1% Tween-20 (Sigma Aldrich, P9416-100ML) and 4% FBS (Thermo Fisher, 10082147) in PBS. Primary antibodies (GATA3 clone L50-823 (1:200; BD, 558686), H3K27me3 C36B11 (1:500; Cell Signaling Technologies, 9733 S), OCT4 (1:200; SantaCruz, sc-5279), SOX2 clone EP103 (1:3; Biogenex, AN833), NANOG (1:200; RnD, AF1997-SP) and EED (E4L6E) XP (1:200; Cell Signaling Technology, 85322)) were diluted in blocking solution and added to the samples, which were incubated at 4 °C overnight. Excess antibodies were washed away using blocking buffer. Secondary antibodies, donkey anti-mouse IgG (H + L) Alexa Fluor 555, donkey anti-rabbit IgG (H + L) Alexa Fluor 647, donkey anti-goat IgG (H + L) Alexa Fluor 647, donkey anti-mouse IgG (H + L) Alexa Fluor 488, donkey anti-rabbit IgG (H + L) Alexa Fluor 555 (all from Thermo Fisher; A-31570, A-31573, A-21447, A-21202 and A-31572, respectively), were diluted in blocking solution, added to the samples and then incubated for 2 h at room temperature. Again, excess antibodies were washed away and samples were incubated with Hoechst 33342 (Thermo Fisher, H3570), which was followed by another set of washes. Samples were mounted using DAKO fluorescent mounting medium (DAKO, S3023). Images were acquired using a Nikon Eclipse Ti spinning disk confocal microscope with a 20× air and 60× oil immersion objective, respectively, and Z-stacks were analysed using ImageJ.

**Image analysis.** Image analysis was performed using CellProfiler 4 software[72]. Briefly, cell nuclei were segmented on the basis of Hoechst staining (three-class global thresholding strategy with Otsu method, threshold smoothing scale of 1.3488) and the intensity of these objects/nuclei across all fluorescence channels was quantified. The image analysis pipeline and raw data are provided as supplementary information. The resulting raw data (mean immunofluorescence intensities) were analysed and figures generated using R. Statistics were plotted with ggpubr (v0.4.0) package. Cells were counted as GATA3 positive if their mean intensity was above 1.5 times the mean of the GATA3 intensity distribution in the matching control sample across all experiments.

**Immunoblotting.** For each growth condition, $1 \times 10^6$ cells were lysed in 100 µl of ice-cold radioimmunoprecipitation buffer (0.1% sodium deoxycholate, 0.1% SDS, 1% Triton X-100, 10 mM HEPES (pH 7.6), 1 mM EDTA, 5% glycerol and 140 mM NaCl) supplemented with Protease Inhibitor Cocktail (PIC, Roche) on ice for 10 min. Lysates were homogenized by sonication for eight to ten cycles at high power, 30 s on/off in a Bioruptor sonicator (Cosmo Bio). Samples were boiled at 95 °C for 5 min with 6× SDS sample buffer before loading onto 4–20% Tris–glycine gels (Bio-Rad). Resolved proteins were transferred to nitrocellulose membranes using the Trans-Blot Turbo system (Bio-Rad) according to the manufacturer's instructions. Membranes were then blocked for 1 h in 1% casein prepared in Tris-buffered saline and 0.1% Tween-20 (TBS-T) before blotting with respective primary antibodies diluted in TBS-T, overnight at 4 °C. Blots were washed three times with TBS-T and incubated with secondary antibodies in the same buffer for 1 h at room temperature (protect from light). After three TBS-T washes, the membranes were imaged on a LI-COR Odyssey FC system. Quantitation of signal and analysis was performed using the LI-COR Image studio software. Primary antibodies included total H3 1:10,000 (Active Motif, 39763), H3K4me3 1:5,000 (Millipore, 04-745), H3K27me3 1:5,000 (Millipore, 07-449) and H2Aub (Cell Signaling, 8240 S). The secondary antibodies were IRDye 680RD anti-rabbit and IRDye 800CW anti-mouse (LI-COR) at 1:5,000 dilution.

**MINUTE-ChIP analysis.** *Preparation of FASTQ files.* Sequencing was performed using 50:8:34 cycles (read 1:index 1:read 2). Illumina bcl2fastq was used to demultiplex paired-end sequencing reads by 8 nt index 1 read (PCR barcode). NextSeq lanes were merged into single fastq files, creating the primary fastq files. Read 1 starts with 6 nt UMI and 8 nt barcode in the format NNNNNNABCDEFGH.

*Primary analysis.* MINUTE-ChIP multiplexed FASTQ files were processed using minute[73], a data processing pipeline implemented in Snakemake[74]. To ensure reproducibility, a conda environment was set. Source code and configuration are available on GitHub (https://github.com/NBISweden/minute). Main steps performed are described below.

*Adaptor removal.* Read pairs matching parts of the adaptor sequence (SBS3 or T7 promoter) in either read 1 or read 2 were removed using cutadapt v3.2[75].

*Demultiplexing and deduplication.* Reads were demultiplexed using cutadapt (v3.2)[75] allowing only one mismatch per barcode. Demultiplexed reads were written into sample-specific FASTQ files used for subsequent mapping and GEO submission.

*Mapping.* Sample-specific paired FASTQ files were mapped to the human genome (hg38) using bowtie2 (v2.3.5.1)[76] with –fast parameter. Alignments were processed into sorted BAM files with samtools (v1.10)[77]. Pooled BAM files were generated from replicates using samtools.

*Deduplication.* Duplicate reads are marked using UMI-sensitive deduplication tool je-suite (v2.0.RC)[78]. Read pairs are marked as duplicates if their read1 (first-in-pair) sequences have the same UMI (allowing for one mismatch) and map to the same location in the genome. Blacklisted regions were then removed from BAM files using BEDTools (v2.29.2)[79].

*Generation of coverage tracks and quantitative scaling.* Input coverage tracks with 1 bp resolution in BigWig format were generated from BAM files using deepTools (v3.5.0)[80] bamCoverage and scaled to a reads per genome coverage (RPGC) of one (1× RPGC, also referred to as '1× normalization') using hg38 genome size 3095978588. ChIP coverage tracks were generated from BAM files using deepTools (v3.5.0) bamCoverage. Quantitative scaling of the ChIP–seq tracks among conditions within each pool was based on their input-normalized mapped read count (INRC). INRC was calculated by dividing the number of unique hg38-mapped reads by the respective number of input reads: #mapped[ChIP]/#mapped[Input]. This essentially corrected for an uneven representation of barcodes in the input, and we previously demonstrated that the INRC is proportional to the amount of epitope present in each condition[42]. Untreated naïve hESC (pooled replicates) was chosen as the reference condition, which was scaled to 1× coverage (also termed RPGC). All other conditions were scaled relative to the reference using the ratio of INRCs

multiplied by the scaling factor determined for 1× normalization of the reference: (#mapped[ChIP]/#mapped[Input])/(#mapped[ChIP_Reference]/#mapped[Input_Reference]) × scaling factor.

*Quality control.* FastQC was run on all FASTQ files to assess general sequencing quality.

Picard (v2.24.1) was used to determine insert size distribution, duplication rate and estimated library size. Mapping statistics were generated from BAM files using samtools (v1.10)[77] idxstats and flagstat commands. Final reports with all the statistics generated throughout the pipeline execution are gathered with MultiQC[81].

*Downstream analysis and visualization.* Total mapped read counts from BAM files were used to calculate relative global levels of histone modifications. Summary values for fixed sized bins or custom intervals were calculated from scaled BigWig files using wigglescout (v0.13.5).

(https://github.com/cnluzon/wigglescout), which aggregates data calculated using rtracklayer (v1.54.0)[82]. Differentially H3K27me3-enriched bins and gene transcription start site (TSS) were calculated using DESeq2 (v1.32.0)[83] using fixed size factors since BigWig files were already scaled by the minute pipeline. Data handling was done using tidyverse (v1.3.1)[84] suite. Figures were created using R (v4.1.0) ggplot2 (v3.3.5)[85]. Additionally, combined heat maps were created using heatmaply (v1.2.1)[86] and extra statistics were plotted with ggpubr (v0.4.0) package when required. Figure 1i was made with ggridges (v0.5.3). Figure 1g was made with karyoploteR package (v1.18.0)[87]. Figure 2b was done with ggalluvial (v0.12.3)[88]. Genome track figures were made using Integrative Genomics Viewer[89] image export function. Corresponding source code for data analysis and figures can be found in the GitHub repository companion to this publication (https://github.com/elsasserlab/hesc-epigenomics). The repository was built using workflowr (v1.6.2)[90]. Data analyses were rendered from R markdown notebooks, and results can be navigated at the corresponding website, generated using workflowr package (v1.6.2)[90].

**RNA-seq analysis.** *Primary analysis.* Bulk RNA-seq data were analysed by RNA-seq pipeline (v2.0) available on nf-core[91] (https://nf-co.re/rnaseq/2.0) with hg38 as reference and RefSeq gene annotation, using STAR[92] as read aligner and RSEM[93] to quantify read counts.

*Downstream analysis and visualization.* Read counts produced by RSEM were used as input for DESeq2[83] differential expression analysis with default parameters. $\log_2$ fold-change ($\log_2$FC) shrinkage apegm[94] was used to filter low-read-count genes. Significance adjusted *P* value cut-off was set to $P < 0.05$ and a fold change of 2. In heat map RNA-seq figures, transcripts per million (TPM) values are shown as ($\log_2$(TPM) + 1) instead of raw counts. Figures were created using R (v4.1.0) ggplot2 (v3.3.5). Additionally, combined heat maps were created using heatmaply (v1.2.1) and extra statistics were plotted with ggpubr (v0.4.0) package when required.

*Pre-processing scRNA-seq data.* Raw scRNA-seq FASTQ files for EZH2i d2, d4 and untreated naïve cells, and EZH2i d7 and untreated primed cells, were aligned to the human GRCh38 reference genome (v.3.0.0, GRCh38, downloaded from the 10X Genomics website) using Cell Ranger version 6.1.1 with default settings for the 'cellranger multi' pipeline (10X Genomics). Reads from EZH2i d7 naïve were aligned on the same reference with the default setting for the 'cellranger count' pipeline (10X Genomics). To minimize differences associated with the sequencing platform used and data processing, published 10X scRNA-seq reads from the PASE model[65] were reprocessed as the same as for EZH2i d7. Published Smart-Seq2 datasets were also mapped on the same reference using the same aligner with default settings, and only uniquely mapped reads were kept for gene expression quantification. Raw read counts were further estimated using rsem-calculate-expression from RSEM tool[93] with the option of '–single-cell-prior'. To filter low-quality cells, a cut-off based on the number of expressed genes (nGene) and percentage of mitochondrial genes (percent. mito) was used. High-quality cells from the EZH2i-related dataset were required to have 1,000 < nGene < 6,000 and percent.mito less than 0.15. Cells from published Smart-Seq2 datasets were required to have at least 2,000 nGene and percent.mito less than 0.125. Cells from PASE model were filtered using the same cut-off reported in their paper[65]. Cells belonging to haemogenic endothelial progenitors and erythroblasts from the CS7 gastrula[66] were excluded in the following analysis. After quality control and exclusion of mitochondrial genes, we focused on genes with one or more counts in at least five cells (assessed for each dataset separately). We then calculated the log-normalized counts using the deconvolution strategy implemented by the computeSumFactors function in R scran package (v1.14.6)[95] followed by rescaled normalization using the multiBatchNorm function in the R batchelor package (v1.2.4)[96] so that the size factors were comparable across batches. log-Normalized expression after rescaling was then used for the human dataset integration and differentially expressed gene detection.

*Data integration, dimensionality reduction and clustering.* As discussed in ref. [54], to avoid over-integration of the amnion with the trophectoderm cells, mutual nearest neighbours were utilized to integrate all datasets together (cells from naïve

cells and primed cells were treated as separate datasets). In detail, this was done by performing a principal component analysis using the top 2,500 most variable genes selected by RunFastMNN function and then correcting the principal components according to their mutual nearest neighbours. We then selected the corrected top 30 principal components for downstream UMAP dimensional reduction and clustering analysis using the RunUMAP function and FindClusters function in the R Seurat (v3.1.4) package[97]. The clustering resolution was set to 0.4 when using the FindClusters function. Cluster identities for 14 individual clusters were assigned manually using co-localization of cell identities from all embryonic datasets as reference, as well as key marker gene and gene signature expression. In detail, cells from EZH2i treated and untreated naïve cells were reclassified as TLCs (C7 and C11), ELCs (C0, C2, C4, and C12), hypoblast-like cells (HLCs) (C9), amnion-like cells (AMLCs) (C3 and C5) and MeLC (C1 and C10), respectively (Extended Data Fig. 8b). The remaining EZH2i-treated and untreated naïve cells (16 cells) were assigned as undefined cells. Fifty and 41 cells annotated as 'ICM' and 'PSA-EPI' from Xiang et al. were excluded for UMAP visualization as they are misclassified[54,98]. Cells from pre-implantation TE, cytotrophoblast, syncytiotrophoblast and extravillous trophoblast were all labelled as 'TE', and cells from nascent mesoderm, emergent mesoderm, advanced mesoderm and axial mesoderm were merged as 'mesoderm' on UMAP visualization. A shiny app generated by shinyCell[99], which includes the above integration, can be browsed at https://petropoulos-lanner-labs.clintec.ki.se.

*Trajectory inference and pseudotime analysis.* Single-cell trajectory analysis of EZH2i-treated and untreated naïve cells was utilized by R Monocle2 package (v.2.14.0) using DDRTree method with default parameters[100]. In detail, raw UMI count was fed as the input for Monocle2. After default normalization, 864 top differentially expressed genes between clusters with $q$ value less than $1 \times 10^{-20}$ were identified by differentialGeneTest function, followed by reconstructing the single-cell trajectory, estimating functional states of cells and calculating pseudotime by reduceDimension and orderCells function. On the basis of the assumption that 95% untreated naïve cells belong to 'ground state' for differentiating, cells with a pseudotime less than 5.949 were identified as the ground ELC. Activated ELC was defined as the ELC cells with pseudotime higher than 5.949, which were further divided into activated ELC, trophectoderm-activated ELC and mesoderm-activated ELC on the basis of their functional states.

*Detection of significant differentially expressed genes using scRNA-seq data.* Detection of differentially expressed genes from scRNA-seq data was performed using 'MAST' test[101] implemented in FindMarkers function from Seurat package. Genes with an FDR less than 0.05, with a fold change more than 1.5 and expressed in more than 10% of the cells were considered differentially expressed. Pseudocount 0.1 was added to avoid dividing by 0.

*Lineage marker-gene detection.* Gene expression from E-MTAB-3929[23], GSE136447[64] and E-MTAB-9388[66] were used to select lineage marker genes. According to published annotations (epiblast cells from E5-E7, E8-E14 and Carnegie stage 7 were grouped as Early Epi, Middle Epi and Late Epi, respectively), pairwise differential expression analysis between lineages was performed using the 'roc' test of FindMarkers function from R package Seurat[97]. The top 50 upregulated marker genes with average power of more than 0.3 conserved in all comparisons were selected.

**Statistics and reproducibility.** For each ChIP, data from three biological replicates were analysed, fulfilling the ENCODE Consortium recommendations of two biological replicates for ChIP–seq data analysis[102]. No statistical method was used to pre-determine sample size. No data were excluded from the analyses. The experiments were not randomized. The investigators were not blinded to allocation during experiments and outcome assessment.

**Reporting summary.** Further information on research design is available in the Nature Research Reporting Summary linked to this article.

## Data availability

The high-throughput data reported in this study have been deposited in GEO under the accession number GSE181244, which includes demultiplexed and deduplicated reads and a quantitatively scaled bigwig track for each sample for the MINUTE-ChIP, bulk RNA-seq and scRNA-seq data. Previously published datasets that were re-analysed here, including scRNA-seq data from three human embryonic datasets and one dataset of PASE (E-MTAB-3929[103], GSE136447 ([64], E-MTAB-9388[66] and GSE134571[65]) are listed in Supplementary Table 1. Source data are provided with this paper. All other data supporting the findings of this study are available from the corresponding author on reasonable request.

## Code availability

Additional code, supplementary data and HTML summaries are available on GitHub: https://github.com/elsasserlab/hesc-epigenomics. HTML summaries

can be browsed at https://elsasserlab.github.io/hesc-epigenomics. The integrated UMAP can be easily browsed at https://petropoulos-lanner-labs.clintec.ki.se.

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

## Acknowledgements

We thank the Live Cell imaging Core facility/Nikon Center of Excellence, at the Karolinska Institute, supported by grants from the Swedish Research Council, KI Infrastructure, and Center for Innovative Medicine, where images were acquired. We acknowledge the Swedish National Infrastructure for Computing (SNIC) at Uppmax server (projects SNIC 2020/15-9 and SNIC 2020/6-3) for providing resources on which the bioinformatics analyses were performed. We are grateful to the Bioinformatics and Expression Analysis facility, supported by the board of research at the Karolinska Institute and the research committee at the Karolinska Hospital for the scRNA-seq infrastructure. We would also like to thank Philip Yuk Kwong Yung for helpful discussions and input. Some schemes were created with BioRender.com. S.J.E. acknowledges funding from Karolinska Institutet SFO Molecular Biosciences, Vetenskapsrådet (2015-04815 and 2020-04313), H2020 ERC-2016-StG (715024 RAPID), the Ming Wai Lau Center for Reparative Medicine, Ragnar Söderbergs Stiftelse, Knut and Alice Wallenberg Foundation (2017.0276, 2019.0215) and Cancerfonden (2015/430). F.L. acknowledges funding from the Ming Wai Lau Center for Reparative Medicine, Ragnar Söderbergs Stiftelse, Wallenberg Academy Fellow (4-148/2017 and 2021.0188), Center for Innovative Medicine and Karolinska Institutet SFO Stem Cells and Regenerative Medicine. S.P. acknowledges funding from the Swedish Research Council (2016-01919), Swedish Society for Medical Research (S16-0039), Natural Sciences and Engineering Research Council of Canada Discovery Grant (RGPIN-2019-05423) and The Canadian Institutes of Health Research (PJT-178082). S.P. holds the Canada Research Chair in Functional Genomics of Reproduction and Development

(950-233204). J.W. acknowledges funding from Päivikki and Sakari Sohlberg Foundation and Sigrid Jusélius Foundation.

## Author contributions

Scientific hypothesis/question: S.J.E. and F.L. Study design: S.J.E., F.L., B.K. and J.P.S. J.P.S. and N.W. cultured the hESCs. B.K. performed the ChIP–seq and bulk RNA-seq experiments. C.N. analysed the associated MINUTE-ChIP and bulk RNA-seq data generated and prepared the corresponding figures. N.W. performed the microscopy experiments and prepared the scRNA-seq libraries with help from L.B.-V. B.K. analysed the microscopy data with input from A.S.M. C.Z. analysed scRNA-seq data and prepared the corresponding figures. J.W. helped with revision experiments. S.J.E, F.L and B.K. wrote the manuscript, with valuable discussions and input from N.W., C.N. and S.P. All others contributed to manuscript proofreading and discussion.

## Funding

## Competing interests

The authors declare no competing interests.

## Additional information

**Extended data** is available for this paper at https://doi.org/10.1038/s41556-022-00916-w.

**Correspondence and requests for materials** should be addressed to Fredrik Lanner or Simon J. Elsässer.

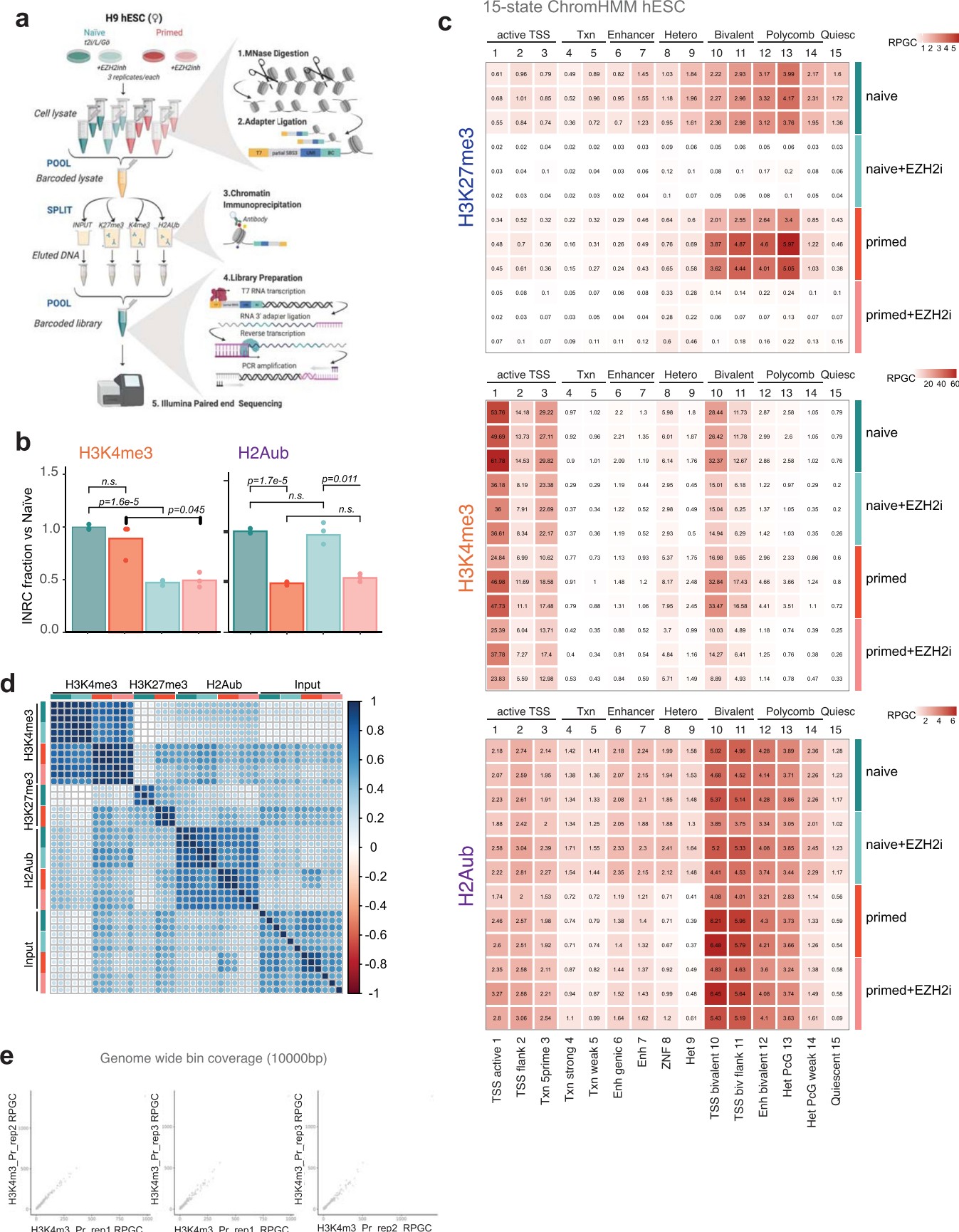

**Extended Data Fig. 1 | See next page for caption.**

**Extended Data Fig. 1 | Scheme of MINUTE-ChIP workflow. a** Scheme of the MINUTE-ChIP workflow. Triplicate cell pellets for each condition were lysed and the chromatin was enzymatically fragmented to mono- and di- nucleosomes and barcoded by ligating on a dsDNA adapter. Samples were then pooled and the barcoded lysate was aliquoted to individual ChIP reactions (5% of the ChIP volume was reserved as input material and carried through protocol in a manner similar to the IPs) with magnetic beads pre-coupled with the respective antibodies. Subsequently, beads were washed and IPed DNA was eluted, proteinase K digested and purified. For constructing the final libraries, DNA from each IP was *in vitro* transcribed using the T7 promoter in the adapter that was ligated on in the initial step. The resulting RNA was appended with an RNA 3′ adapter (RA3) allowing for specific paired end sequencing. The RA3 in turn was used to prime the reverse transcription reaction, generating cDNA that was used as a template for the final low-cycle library PCR. At this stage, in addition to the Illumina-compatible sequences, PCR primers also carried a second barcode sequence to serve as an identifier for the IP performed. Finally, libraries are pooled and sequenced on the Illumina platform[42,104]. **b** Global genome-wide levels of H3K4me3 and H2Aub as determined by input-normalized total read counts (INRC) in naïve or primed hESC, cultured with or without EZH2 inhibitor. *P* values of pairwise comparisons (two-sided unpaired Student's t test) are given. n = 3 biologically independent samples. Tracks for subsequent analysis are scaled according to the INRC values as reads per genome coverage (RPGC), with naïve serving as a reference scaled to 1x genome coverage (global average equals to 1 RPGC) **c** Histone H3K27me3, H3K4me3 and H2Aub levels by chromatin state. RPGC of individual replicates are shown. **d** Genome-wide correlation (10 kb bins, Persson-correlation coefficient) of MINUTE-ChIP replicates, including corresponding inputs. **e** Exemplary genome-wide comparison of 10 kb bins as scatter plot.

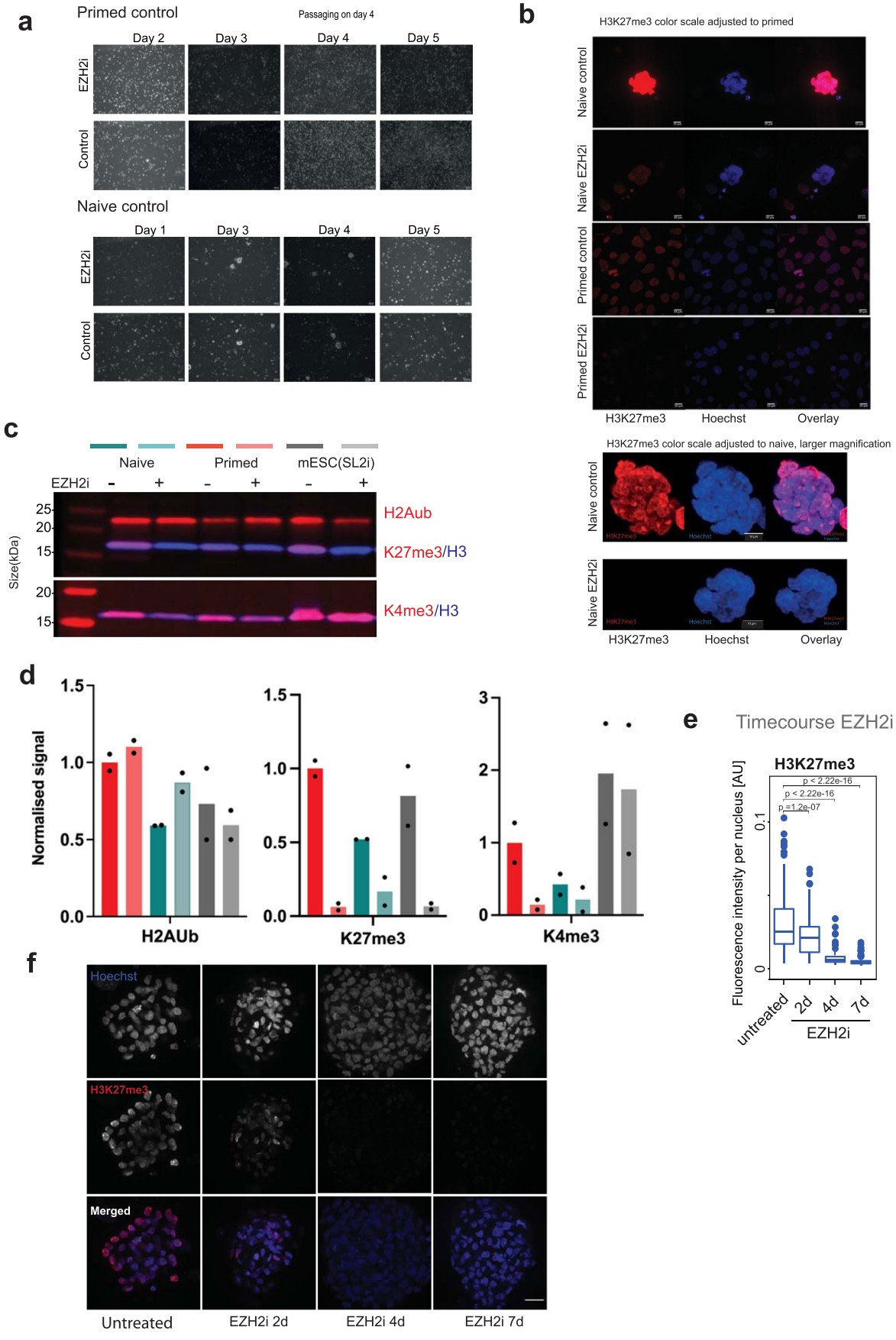

**Extended Data Fig. 2 | See next page for caption.**

**Extended Data Fig. 2 | H3K27me3 levels in naïve and primed hESC. a** Phase contrast microscopy images of primed and naïve hESC, untreated or treated with EZH2i over 5 days showing similar growth and morphology. Data shown represent 2 independent experiments. Scale bars 100 μm. **b** Immunofluorescence microscopy showing H3K27me3 staining of naïve and primed hESC +/- EZH2i treatment. Stainings were performed in parallel in the same antibody dilutions and images were acquired with the same gain settings to allow a quantitative comparison. Data shown represent 2 independent experiments. Scale bars 10 μm. **c** Representative western blot for quantification of H3K4me3, H2Aub and H3K27me3 assayed using two-color IR western blot, in hESC and WT(J1) mouse ESC (grown in 2i/Serum) with and without EZH2i treatment. **d** Quantification of western blots, represented as mean of two biological replicates. Representative image shown. We note that the discrepancy in fold-change comparing naïve and primed hESC with different quantitative methodologies MINUTE-ChIP, IF, western blots may arise from method-specific threshold sensitivity, dose-response curves and signal saturation levels. MINUTE-ChIP measures H3K27me3-density on the level of nucleosomes, whereas western blot yields a per-histone level quantification. We have previously confirmed that MINUTE-ChIP signal is linearly proportional under the dynamic range relevant to the H3K27me3 levels assayed[42]. **e** Time course of H3K27me3-depletion using immunofluorescence microscopy. Boxplot showing quantification performed using CellProfiler image analysis of 1172 nuclei (237 untreated, 140 day 2, 314 day 4, 451 day 7) derived from 1 experiment. Box plot boxes show the 25th and 75th percentile with the median, and whiskers indicate 1.5x the interquartile range. P values determined by two-sided unpaired t test (respectively $p = 1.2e-7$, $p < 2.22e-16$, $p < 2.22e-16$). **f** Representative immunofluorescence microscopy image for e), scale bar 10 μm.

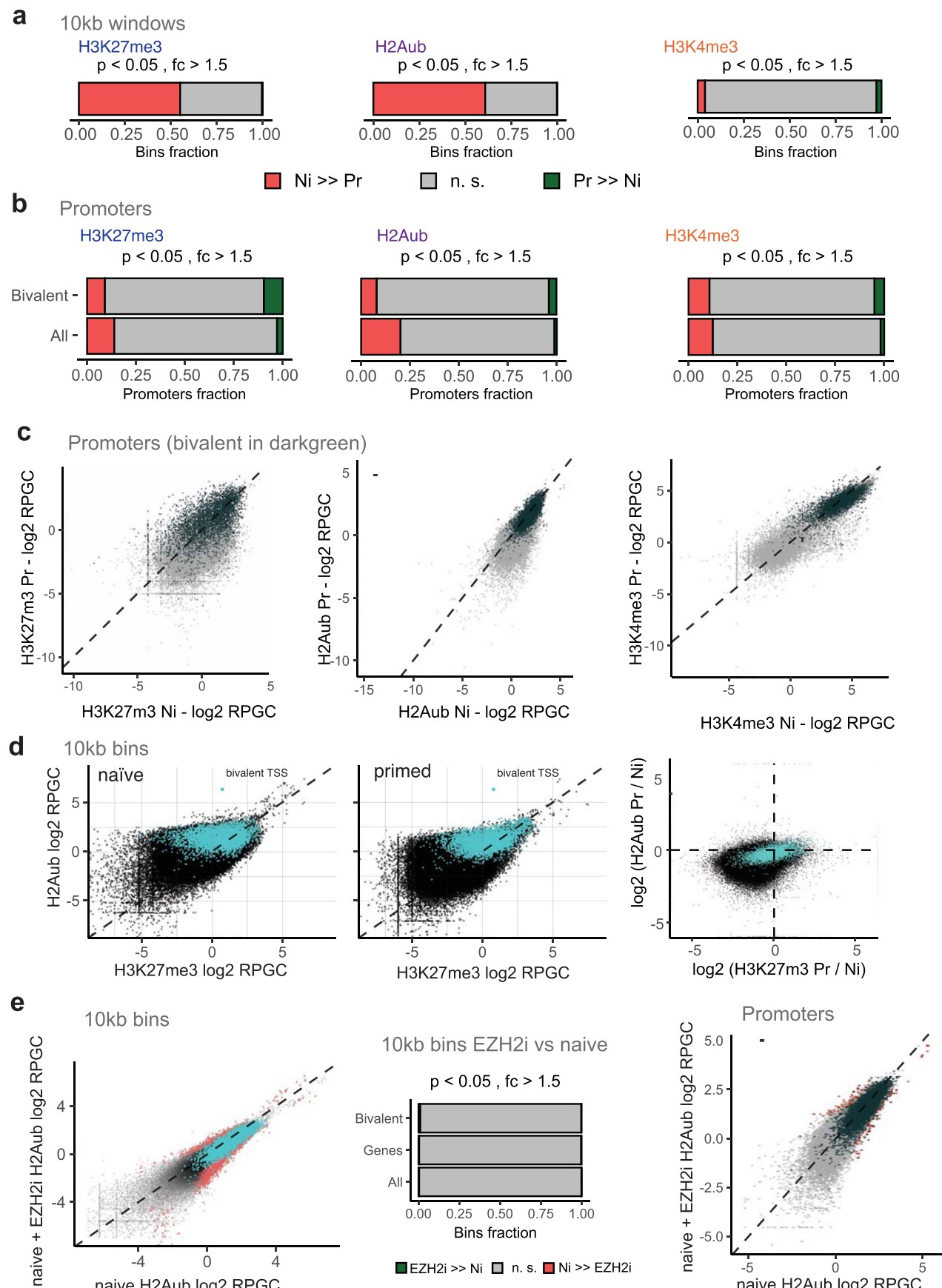

**Extended Data Fig. 3 | See next page for caption.**

**Extended Data Fig. 3 | Genome-wide histone posttranslational modification changes between naïve and primed hESC. a** Fraction of genome-wide 10 kb windows that significantly (DESeq2 p.adj < 0.05 and fold-change > 1.5 from three replicates) gain or lose H3K27me3, H2Aub or H3K4me3 levels between naïve and primed state. **b** Fraction of promoters (all or bivalent[45]) that significantly (DESeq2 p.adj < 0.05 and fold-change > 1.5 from three replicates) gain or lose H3K27me3, H2Aub or H3K4me3 levels between naïve and primed state. **c** Scatter plots showing H3K27me3, H2Aub or H3K4me3 levels (log2-transformed RPGC) at promoters in naïve versus primed hESC. Bivalent genes are highlighted in black. **d** Scatter plots showing the relation of H3K27me3 and H2Aub across genome-wide 10 kb bins in naïve and primed hESC, as well as the correlation of changes. 10 kb bins overlapping with bivalent TSS are highlighted in turquoise **e** Scatter plot showing the relation of H2Aub across genome-wide 10 kb bins in naïve vs. EZH2i-treated naïve hESC. Significant bins are highlighted in red. 10 kb bins overlapping with bivalent promoters are highlighted in turquoise. Fraction of 10 kb bins significantly changing (DESeq2 p.adj < 0.05 and fold-change > 1.5 from three replicates) are given for all 10 kb bins, 10 kb bins overlapping with promoters, and 10 kb bins overlapping with bivalent promoters. Scatter plot showing the relation of H2Aub across promoters in naïve vs. EZH2i-treated naïve hESC. Significant bins are highlighted in red. Bivalent promoters are highlighted in black.

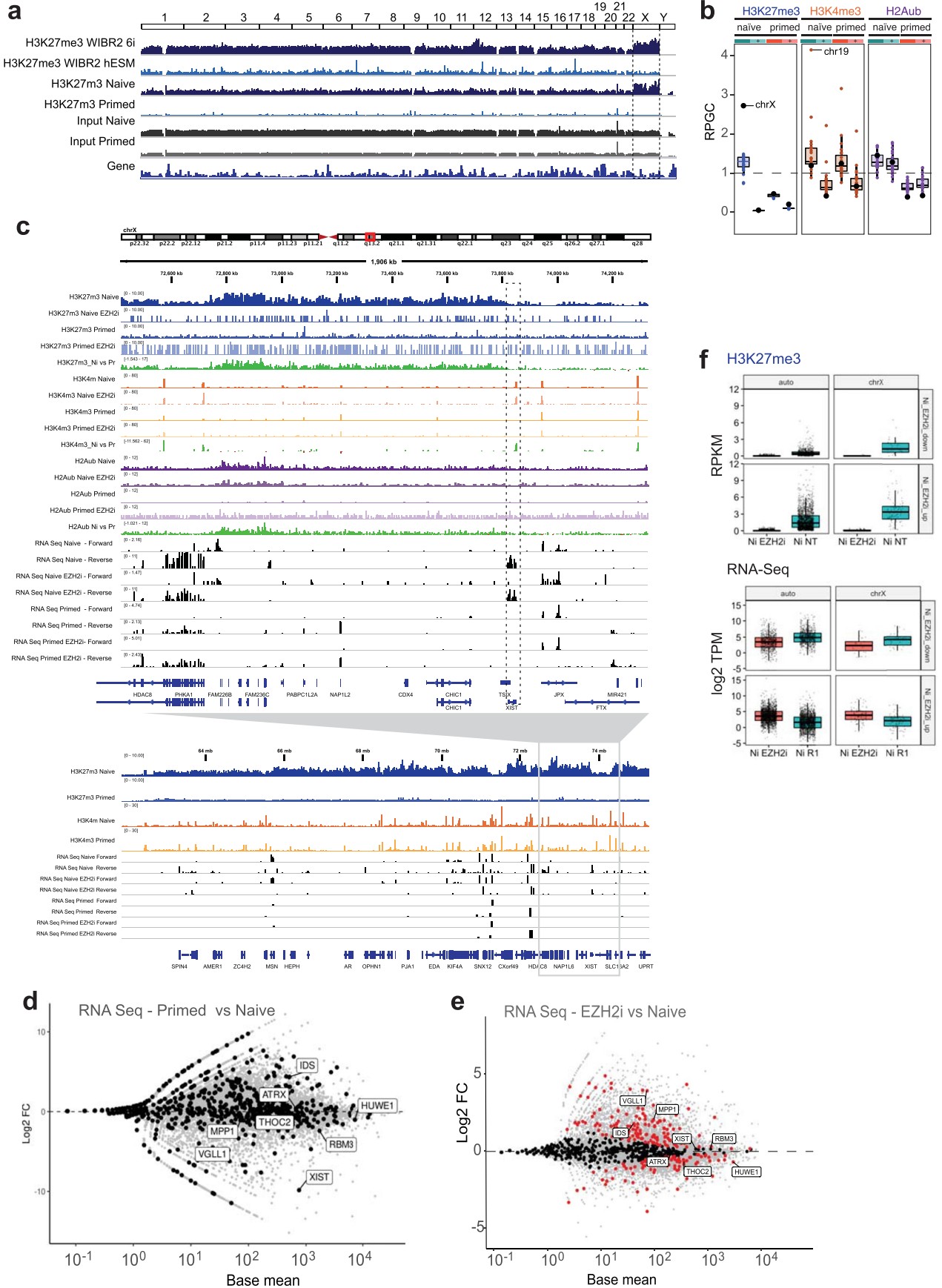

Extended Data Fig. 4 | See next page for caption.

**Extended Data Fig. 4 | High H3K27me3 density on naïve X chromosomes. a** Representative Genome browser overview showing H3K27me3 density across all chromosomes. Published H3K27me3 ChIP[17] and MINUTE-ChIP H3K27me3 tracks (combined replicates), as well as MINUTE-ChIP input tracks are shown. **b** chromosome average enrichment of H3K4me3, H3K27me3 and H2Aub in naïve and primed hESC, cultured with or without EZH2i for 7 days. RPGC of combined replicates is shown. Box plot boxes show the 25th and 75th percentile with the median, and whiskers indicate 1.5x the interquartile range. **c** Genome browser view of the XIST neighborhood on the X chromosome at megabase resolution. Shown is the combined signal of three replicates for H3K27me3, H3K4me3, H2Aub as well as gain/loss tracks comparing naïve and primed signal, and stranded RNA-Seq signal (one of three replicates). Tracks from the same histone modification are shown on the same RPGC scale. RNA-Seq expression is shown on the same scale. Strong H3K27me3 accumulation in large blocks on naïve X chromosome is evident, with the exception of some islands that typically harbor genes active in naïve state, such as the XIST locus, which do not accumulate H3K27me3. **d** MA-plot of RNA-Seq data (Base mean and log2 fold-change as calculated with DESeq2 from triplicates) comparing naïve and primed hESC. Genes on the X chromosome are highlighted in black with selected annotations. **e** MA-plot of RNA-Seq data (Base mean and log2 fold-change as calculated with DESeq2 from triplicates) comparing untreated and EZH2i-treated naïve hESC. Genes on the X chromosome are highlighted in black (padj < 0.05 highlighted in red) with selected annotations. **f** H3K27me3 levels at promoters on the X chromosome or autosomes in naïve hESC +/- EZH2i treatment (top). Two groups of promoters are plotted: those of genes upregulated with EZH2i treatment in naïve hESC and those downregulated. Expression levels (log2-transformed TPM) at genes on the X chromosome or autosomes in naïve and primed hESC +/- EZH2i treatment (bottom). Box plot boxes show the 25th and 75th percentile with the median, and whiskers indicate 1.5x the interquartile range.

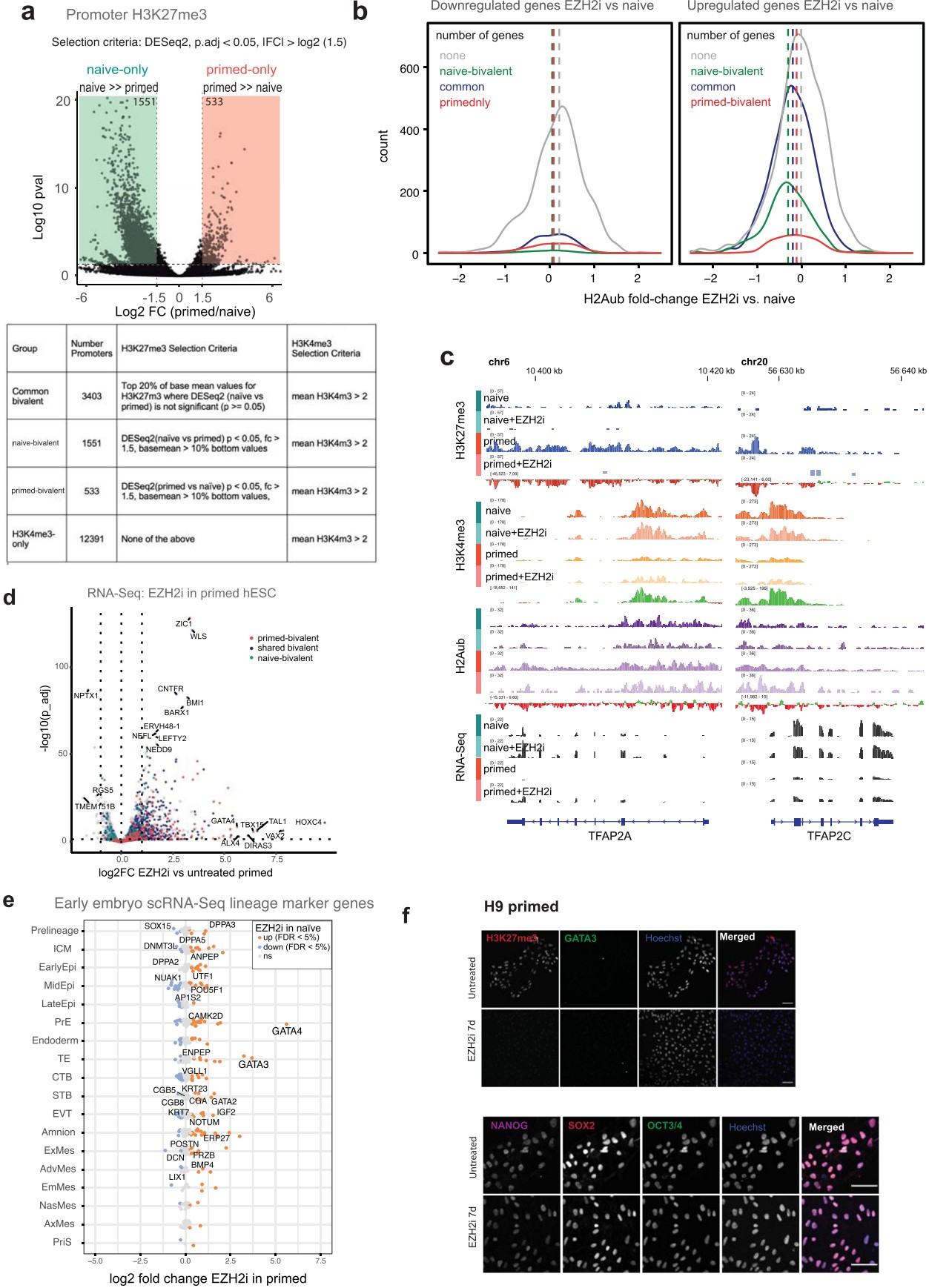

**Extended Data Fig. 5 | See next page for caption.**

**Extended Data Fig. 5 | Defining shared and state-specific bivalent promoters in naïve and primed hESC. a** Differential occupancy H3K27me3 at promoters using DESeq2. Volcano plot (DESeq2 based on three replicates) comparing promoter H3K27me3 levels between naïve and primed hESC. Explanation of criteria for defining naïve-bivalent, primed-bivalent and common bivalent gene classes. **b** Density plot of fold-changes of H2Aub levels following H3K27me3 depletion in hESC. Only genes that were derepressed upon EZH2i-treatment (DESeq2 p.adj < 0.05, fold-change > 2 based on three replicates) were included in the analysis. The different classes of bivalent promoters as defined above are compared to H3K27me3-devoid promoters. **c** Genome browser view of *TFAP2A* and *TFAP2C* transcription factors in naïve and primed hESC +/− 7d EZH2i treatment. Shown is the combined signal of three replicates for H3K27me3, H3K4me3, H2Aub as well as gain/loss tracks comparing naïve and primed signal, and stranded RNA-Seq signal (one of three replicates shown). Tracks from the same histone modification are shown on the same RPGC scale. RNA-Seq expression is shown on the same TPM scale. **d** Volcano plot with selected annotations showing differentially expressed genes (DESeq2 FDR < 5%, |log2FC| > 1) between 7d EZH2i-treated and untreated primed hESC. Bivalent gene classes are shown as colored dots: red (primed-bivalent), blue (shared-bivalent), teal (naïve bivalent).**e** Stripchart showing expression of a comprehensive set of marker genes defined from human embryo single cell data comparing primed hESC +/− 7d EZH2i. Markers are grouped into pre-lineage, inner cell mass (ICM), epiblast, primitive endoderm, (PrE), trophectoderm (TE) cytotrophoblast (CTB), syncytiotrophoblast (STB), extravillous trophoblast (EVT), amnion, extraembryonic mesoderm (exMes), advanced mesoderm (AdvMes), emerging mesoderm (EmMes), nascent mesoderm (NasMes), axial mesoderm (AxMes) and primitive streak (PriS). Significant differences (DESeq2 p.adj < 0.05 from triplicates) are highlighted in color. **f** Immunofluorescence confocal microscopy images of primed H9 hESC colonies with or without EZH2i (EPZ-6438) treatment for 7 days, stained for pluripotency markers NANOG, SOX2 and OCT3/4 (*right panel*) *and* H3K27me3 and trophectoderm transcription factor GATA3 (*left panel*). Data shown represent 2 independent experiments. Scale bars 50 μm.

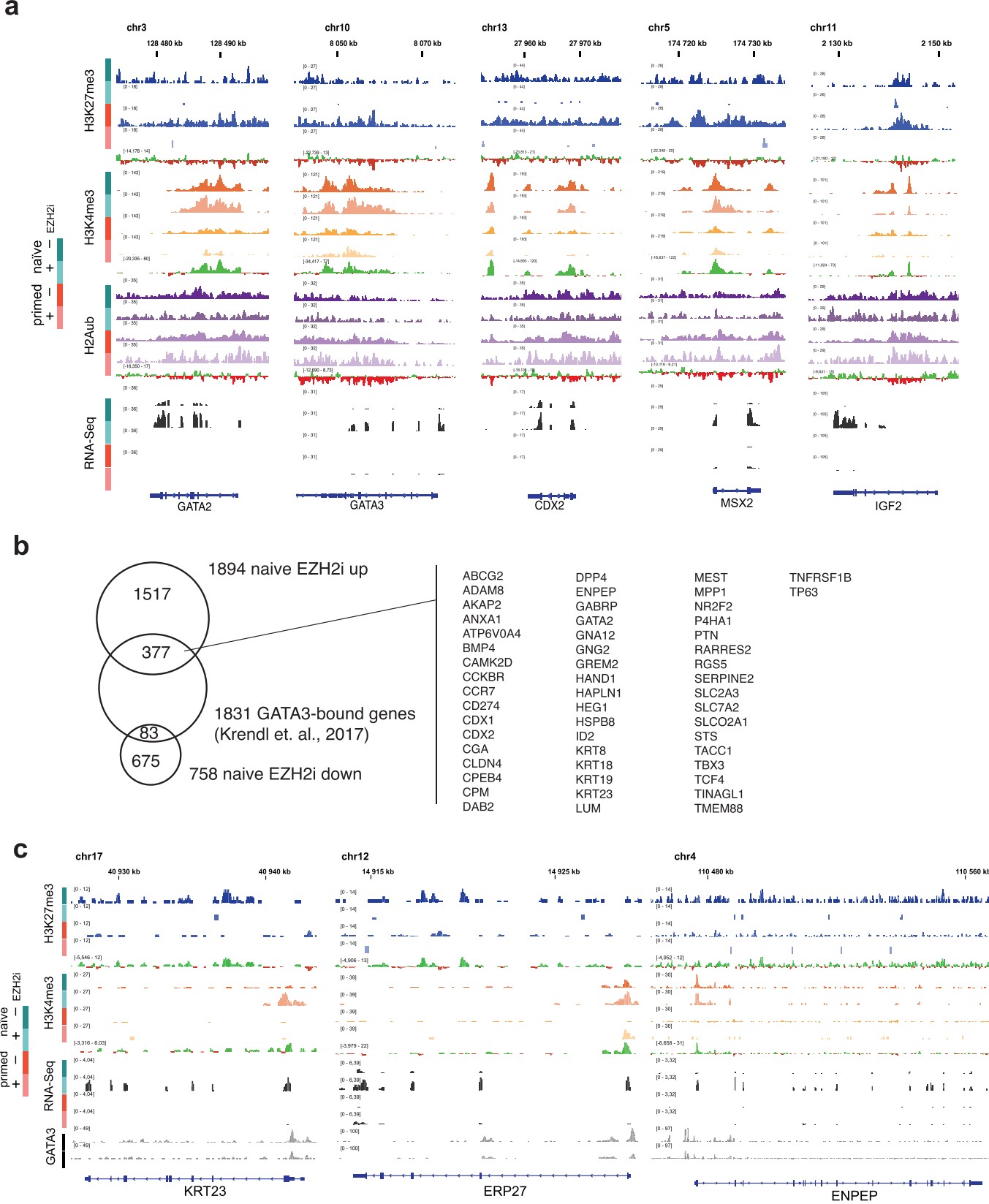

Extended Data Fig. 6 | See next page for caption.

**Extended Data Fig. 6 | Loss of H3K27me3 in naïve hESC activates trophectoderm gene expression programs. a** Genome browser examples of selected bivalent trophectoderm lineage markers. Shown is the combined signal of three replicates for H3K27me3, H3K4me3, H2Aub, RNA-Seq, as well as gain/loss tracks comparing naïve and primed signals. Tracks from the same histone modification are shown on the same RPGC scale. RNA-Seq expression from one replicate is shown on the same TPM scale. **b** Venn diagram intersecting gene sets: 1894 significantly upregulated genes in naïve hESC treated with EZH2i, 758 significantly downregulated genes in naïve hESC treated with EZH2i, 1831 genes with promoter-proximal GATA3 binding sites[9]. **c** Genome browser examples of selected trophectoderm genes with GATA3 binding sites. Shown is the combined signal of three replicates for H3K27me3, H3K4me3, H2Aub, RNA-Seq, as well as gain/loss tracks comparing naïve and primed signals. Tracks from the same histone modification are shown on the same RPGC scale. RNA-Seq expression from one replicate is shown on the same TPM scale.

**a**   HS975 naive in 5iLAF medium

HS975 naive

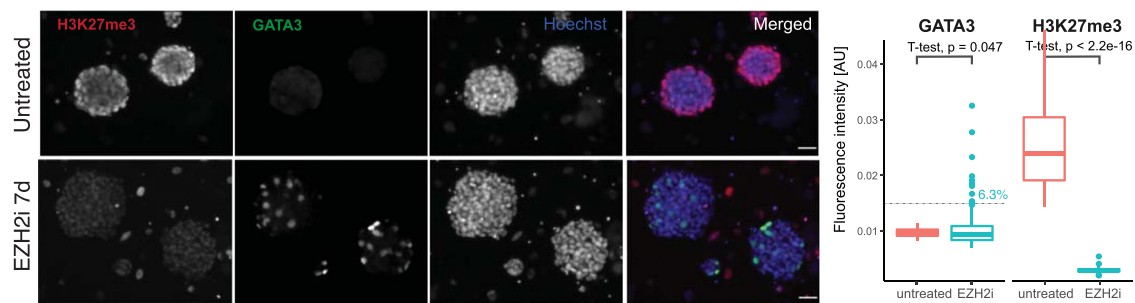

**b**   Immunofluorescence imaging EED and GATA3

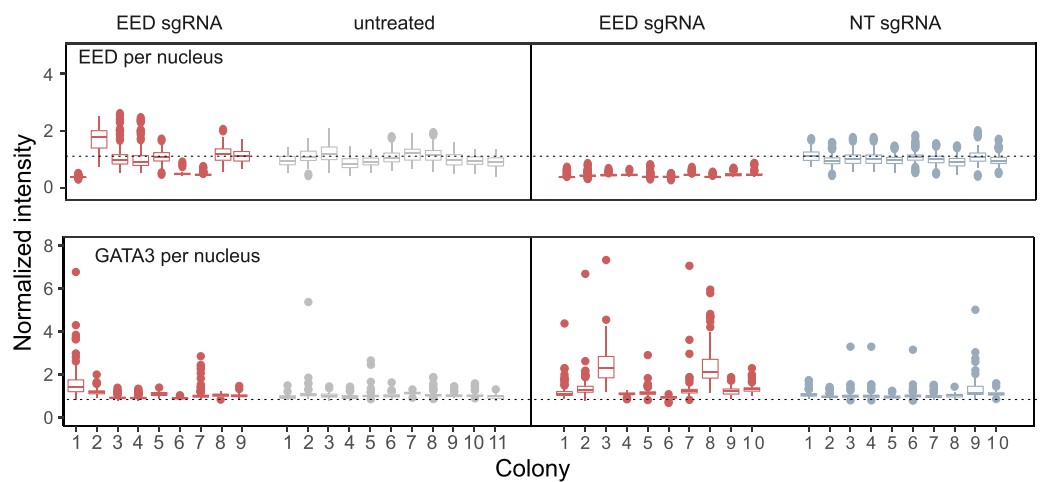

**c**   H9 naive in t2iLGo vs PXGL

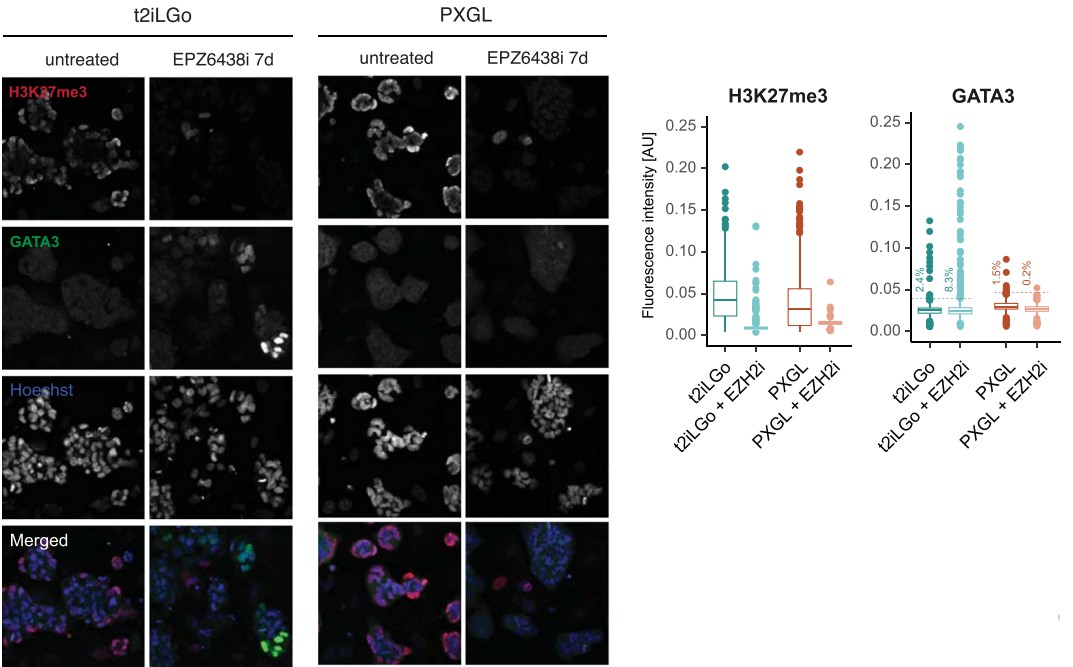

**Extended Data Fig. 7 | See next page for caption.**

**Extended Data Fig. 7 | GATA3 induction upon pharmacological and genetic targeting of PRC2. a** Immunofluorescence microscopy (on Olympus IX81, 20x magnification) showing H3K27me3 and GATA3 staining of naïve HS975 hESC cultured in 5iLAF medium with or without EZH2i (EPZ-6438) treatment for 7 days. Scale bars are 50 μm. Tukey boxplot generated from corresponding CellProfiler image analysis shows per-nucleus H3K27me3 and GATA3 intensities over n = 195 nuclei (69 untreated, 127 EZH2i) derived from 1 experiment. EZH2i treatment resulted in 6.3% GATA3 + nuclei for this cell type. The threshold for GATA3 + cells (dashed line) is defined as 1.5x mean of untreated cells. Box plot boxes show the 25th and 75th percentile with the median, and whiskers indicate 1.5x the interquartile range. P values estimated by two-sided unpaired t test (p = 0.047; p < 2.2e-16, respectively).
**b** Boxplots show per-nucleus EED (top panel) and GATA3 (bottom panel) intensities summarized by the imaged colonies for CRISPR/Cas9 based acute targeting of EED. corresponding to Fig. 4f. Box plot boxes show the 25th and 75th percentile with the median, and whiskers indicate 1.5x the interquartile range. **c** Immunofluorescence microscopy showing H3K27me3 and GATA3 staining of naïve H9 hESC cultured in t2iLGö (NaiveCult) or PXGL medium with or without EZH2i (EPZ-6438) treatment for 7 days. Scale bar is 20 μm. Tukey boxplot generated from corresponding CellProfiler image analysis shows per-nucleus H3K27me3 and GATA3 intensities over 3045 nuclei in total (500 t2iLGö, 846 t2iLGö +EZH2i, 1037 PXGL, 662 PXGL + EZH2i) derived from 1 experiment. The threshold for GATA3 + cells (dashed line) is defined as 1.5x mean of untreated cells. Box plot boxes show the 25th and 75th percentile with the median, and whiskers indicate 1.5x the interquartile range.

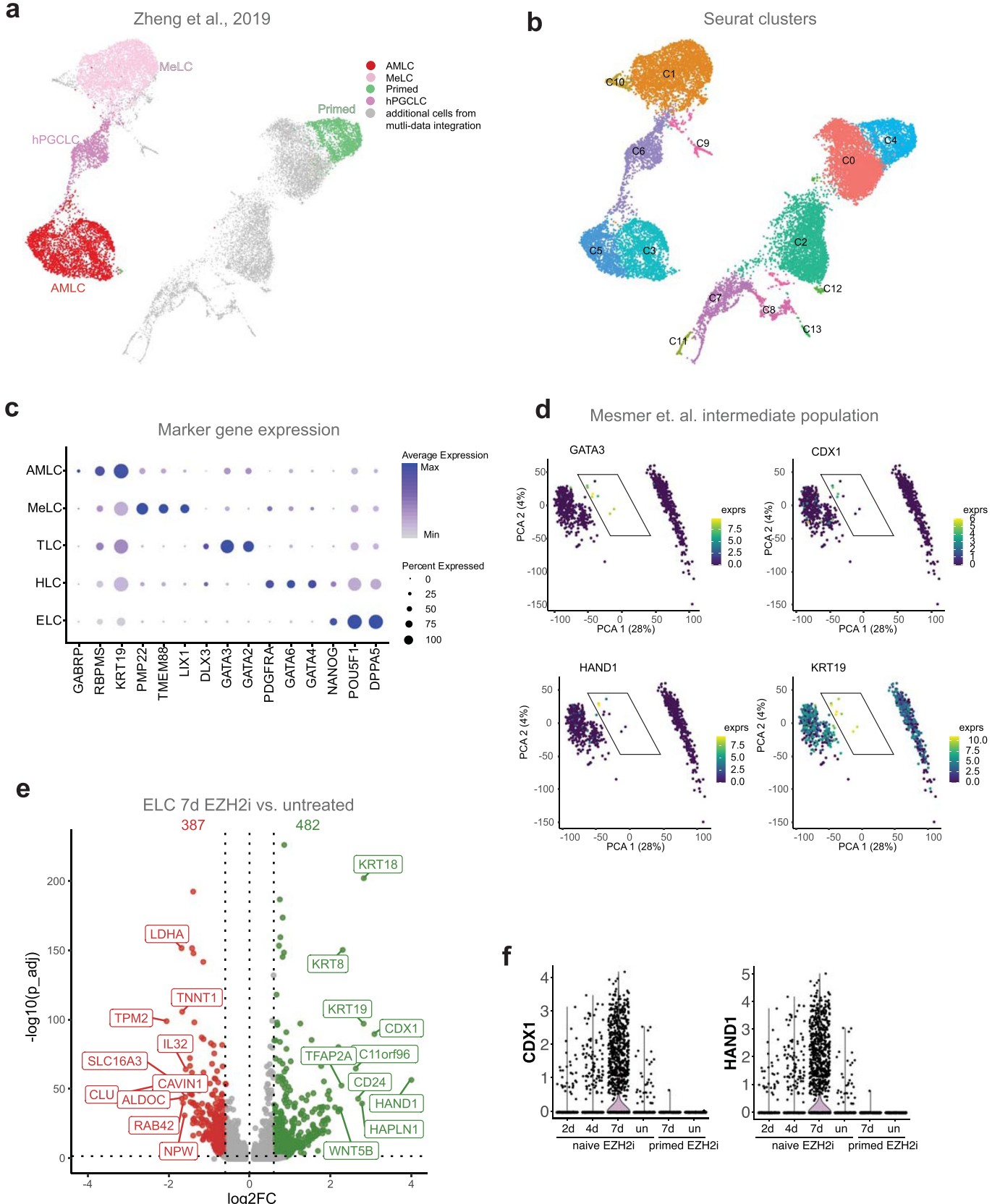

**Extended Data Fig. 8 | See next page for caption.**

**Extended Data Fig. 8 | scRNA-seq UMAP cluster assignment and characterization. a** UMAP projection of integrated datasets as in Fig. 5a, highlighting cells from Zheng et al[65]. **b** Unassigned Seurat cluster distribution of UMAP used in Fig. 5a. **c** Bubble plot showing expression of key marker genes across the cell clusters as indicated. The sizes and colors of dots indicate the proportion of cells expressing the corresponding genes and their averaged scaled values of log-transformed expression, respectively. **d** Reanalysis of published scRNA-seq in naïve hESC from Messmer et. al[53]. Expression of *GATA3, CDX1, HAND1, KRT19* is color coded in PCA map as defined in Messmer et. al. **e** Volcano plot showing differentially expressed genes (482 up, 387 down) comparing ELC population at 7d EZH2i with ELC population in untreated naïve hESC. **f** Violin plot showing single-cell log-transformed expression of HAND1 and CDX1 transcription factors.

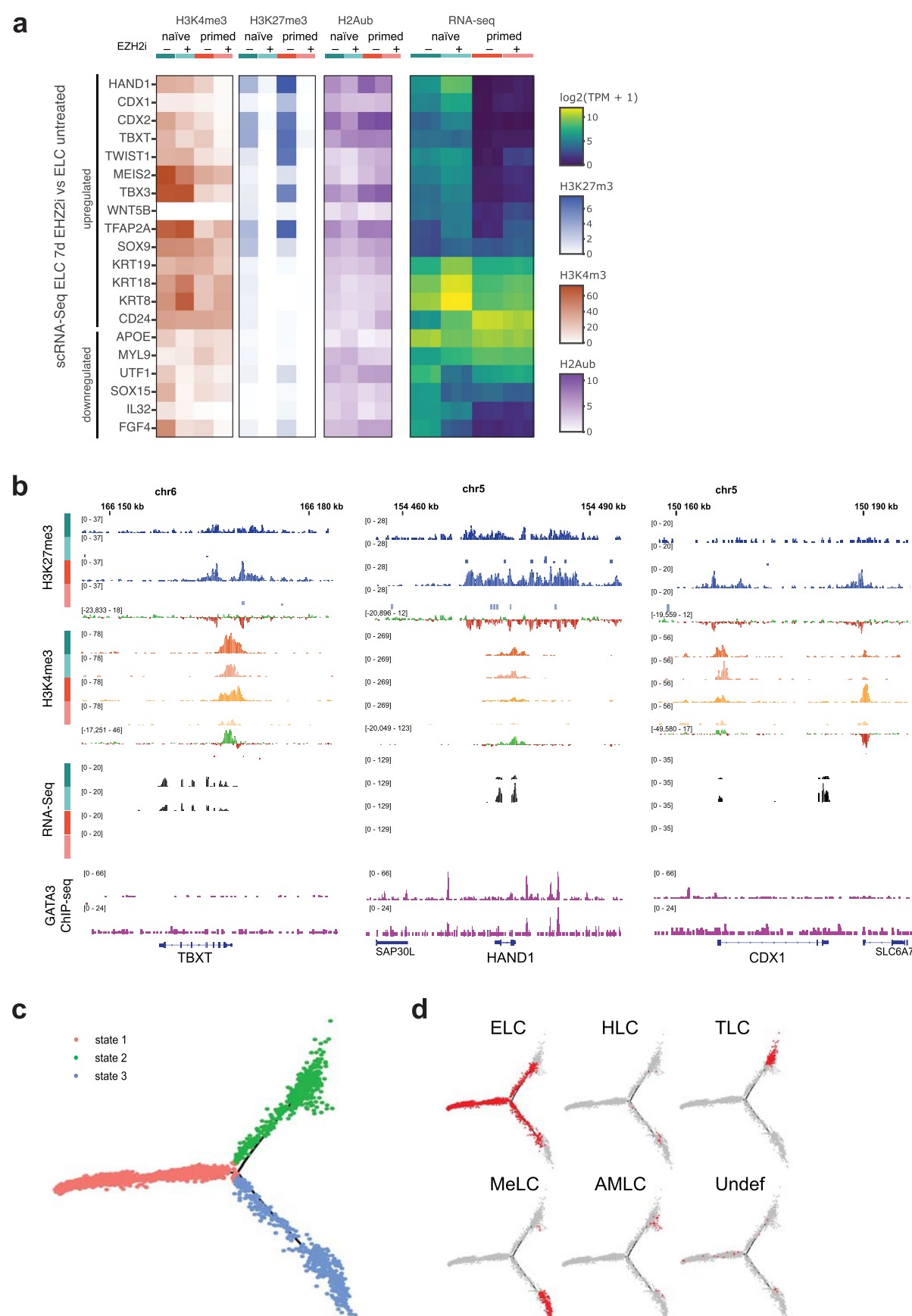

**Extended Data Fig. 9 | See next page for caption.**

**Extended Data Fig. 9 | Epigenomic landscape of differentially expressed genes in ELC population after 7d EZH2i treatment. a** Heatmap showing RNA-Seq expression levels (log2-transformed TPM) of core pluripotency markers in naïve and primed hESC +/− 7d EZH2i treatment, as well as the H3K4me3 and H3K27me3 levels (RPGC) at their respective promoter. RPGC from combined replicates are used for H3K4me3 and H3K27me3, whereas the three individual replicate TPM values are plotted for RNA-Seq data. **b** Genome browser view of *TBXT*, *HAND1* and *CDX1* transcription factors in naïve and primed hESC +/− 7d EZH2i treatment. Shown is the combined signal of three replicates for H3K27me3, H3K4me3, H2Aub as well as gain/loss tracks comparing naïve and primed signal, and stranded RNA-Seq signal. Tracks from the same histone modification are shown on the same RPGC scale. RNA-Seq expression is shown on the same TPM scale. Published GATA3 ChIP-seq tracks are shown[9]. **c** Trajectory inference colored-coded by inferred cell state calculated by monocle. **d** Indicated cell populations as defined in Fig. 5b are mapped onto the trajectory (in red).

Fredrik Lanner

# Reporting Summary

## Statistics

For all statistical analyses, confirm that the following items are present in the figure legend, table legend, main text, or Methods section.

| n/a | Confirmed | |
|---|---|---|
| ☐ | ☒ | The exact sample size (*n*) for each experimental group/condition, given as a discrete number and unit of measurement |
| ☐ | ☒ | A statement on whether measurements were taken from distinct samples or whether the same sample was measured repeatedly |
| ☐ | ☒ | The statistical test(s) used AND whether they are one- or two-sided *Only common tests should be described solely by name; describe more complex techniques in the Methods section.* |
| ☒ | ☐ | A description of all covariates tested |
| ☐ | ☒ | A description of any assumptions or corrections, such as tests of normality and adjustment for multiple comparisons |
| ☐ | ☒ | A full description of the statistical parameters including central tendency (e.g. means) or other basic estimates (e.g. regression coefficient) AND variation (e.g. standard deviation) or associated estimates of uncertainty (e.g. confidence intervals) |
| ☐ | ☒ | For null hypothesis testing, the test statistic (e.g. *F*, *t*, *r*) with confidence intervals, effect sizes, degrees of freedom and *P* value noted *Give P values as exact values whenever suitable.* |
| ☒ | ☐ | For Bayesian analysis, information on the choice of priors and Markov chain Monte Carlo settings |
| ☒ | ☐ | For hierarchical and complex designs, identification of the appropriate level for tests and full reporting of outcomes |
| ☐ | ☒ | Estimates of effect sizes (e.g. Cohen's *d*, Pearson's *r*), indicating how they were calculated |

*Our web collection on statistics for biologists contains articles on many of the points above.*

## Software and code

Policy information about availability of computer code

**Data collection**
For ChIP-seq data collection, Illumina NextSeq500 platform with paired-end settings was used. FASTQ files were obtained from Illumina BaseSpace.RNA-seq data collection through bgi service https://www.bgi.com/global/index.scRNA-seq: cDNA libraries were sequenced using Illumina Nextseq 2000 Platform 100 cycles P2 and Illumina Nextseq 550 High Output kit v2.5 (150 Cycles). More information can be found in the methods section of the manuscript.

**Data analysis**
ChIP-seq data primary analysis was done using minute, a workflow for multiplexed ChIP analysis: https://github.com/NBISweden/minute.RNA-seq data primary analysis was done using standard rnaseq analysis pipeline (v2.0) from nf-core: https://nf-co.re/rnaseq . Differential expression analysis was done using DESeq2. scRNA-seq raw reads were aligned to the human GRCh38 reference genome (v.3.0.0, GRCh38, from the 10X Genomics website) using Cell Ranger v6.1.1 with default settings for the 'cellranger multi' pipeline (10X Genomics).Details in the workflow steps on the methods section of the manuscript.
Downstream analysis performed with a broad set of open-source publicly available R v4.1.2 libraries: ggplot2 v3.3.5, ggpubr v0.4.0, DESeq2 v1.34.0, tidyverse v1.3.1, rtracklayer v1.54.0, workflowr v1.6.2, wigglescout v0.13.5, seurat v3.1.4, scran v1.14.6, batchelor v1.2.4. Supplementary code for the downstream analysis and figure generation is available at https://github.com/elsasserlab/hesc-epigenomics. scRNA-seq data can be browsed at https://petropoulos-lanner-labs.clintec.ki.se/app/shinyEZH2i. Trajectory analysis was performed using monocle2 v2.14.0.

For manuscripts utilizing custom algorithms or software that are central to the research but not yet described in published literature, software must be made available to editors and reviewers. We strongly encourage code deposition in a community repository (e.g. GitHub). See the Nature Portfolio guidelines for submitting code & software for further information.

## Data

Policy information about availability of data

All manuscripts must include a data availability statement. This statement should provide the following information, where applicable:

- Accession codes, unique identifiers, or web links for publicly available datasets
- A description of any restrictions on data availability
- For clinical datasets or third party data, please ensure that the statement adheres to our policy

The high-throughput data reported in this study have been deposited in GEO under the accession number GSE181244, which includes demultiplexed and deduplicated reads and a quantitatively scaled bigwig track for each sample. Previously published datasets that were re-analysed here, including scRNA-seq data from 3 human embryonic datasets and one dataset of post-implantation amniotic sac embryoid (PASE) (E-MTAB-3929 86, GSE136447 68, E-MTAB-9388 70, and GSE134571 69) are listed in Supplementary Table 1.

# Field-specific reporting

Please select the one below that is the best fit for your research. If you are not sure, read the appropriate sections before making your selection.

☒ Life sciences  ☐ Behavioural & social sciences  ☐ Ecological, evolutionary & environmental sciences

For a reference copy of the document with all sections, see nature.com/documents/nr-reporting-summary-flat.pdf

# Life sciences study design

All studies must disclose on these points even when the disclosure is negative.

| | |
|---|---|
| Sample size | No statistical test was performed to determine sample size. Sample size for omics experiments ( 3 biological replicates for ChIP seq, bulk and sc RNA seq) was determined based on typical ranges used in the field (https://genome.cshlp.org/content/genome/22/9/1813.full.html.) |
| Data exclusions | No data were excluded for the analyses. |
| Replication | The ChIP-seq, Bulk and sc-RNA seq were performed once with 3 biological replicates within each of the experiments. All other experiments in the study were repeated in separate batches and are reported in the respective figure legends. |
| Randomization | For cell culture experiments, treatment groups were attributed randomly between wells. Within the MINUTE-ChIP and sc RNA seq experiment barcodes were assigned randomly between samples. IF images were also acquired with no particular bias. For other experiments in the study randomization was not applicable due to the nature of the protocol. |
| Blinding | No blinding was done |

# Reporting for specific materials, systems and methods

We require information from authors about some types of materials, experimental systems and methods used in many studies. Here, indicate whether each material, system or method listed is relevant to your study. If you are not sure if a list item applies to your research, read the appropriate section before selecting a response.

## Materials & experimental systems

| n/a | Involved in the study |
|---|---|
| ☐ | ☒ Antibodies |
| ☐ | ☒ Eukaryotic cell lines |
| ☒ | ☐ Palaeontology and archaeology |
| ☒ | ☐ Animals and other organisms |
| ☒ | ☐ Human research participants |
| ☒ | ☐ Clinical data |
| ☒ | ☐ Dual use research of concern |

## Methods

| n/a | Involved in the study |
|---|---|
| ☐ | ☒ ChIP-seq |
| ☒ | ☐ Flow cytometry |
| ☒ | ☐ MRI-based neuroimaging |

## Antibodies

| | |
|---|---|
| Antibodies used | Primary antibodies :<br>Immunoflourescence<br>GATA3 clone L50-823 (1:200, BD; 558686)<br>H3K27me3 C36B11 (1:500, Cell Signaling Technologies; 9733S)<br>OCT4 (1:200, SantaCruz; sc-5279)<br>SOX2 clone EP103 (1:3, Biogenex; AN833) |

NANOG (1:200, RnD; AF1997-SP)
EED (E4L6E) XP® (1:200, Cell Signaling Technology; 85322)
Immunoblotting
H3 ( 1:10,000,Active motif 39763)
H3K4me3 (1: 5000, Millipore 04-745)
H3K27me3 (1: 5000, Millipore 07-449)
H2Aub (1:5000, Cell Signaling 8240S)

Secondary antibodies:
Immunofluorescence
donkey a-mouse IgG (H+L) Alexa fluor 555, donkey a-rabbit IgG (H+L) Alexa fluor 647, donkey a-goat IgG (H+L) Alexa fluor 647,
donkey a-mouse IgG (H+L) Alexa fluor 488, donkey a-rabbit IgG (H+L) Alexa fluor 555
(all from Thermofisher; A-31570, A-31573, A-21447, A-21202 and A-31572, respectively)
Immunoblotting
IRDye® 680RD anti-rabbit and IRDye® 800CW anti-mouse (LI-COR) at 1:5000 dilution

Validation

All antibodies were previously validated by vendors and/or published work.
GATA3 clone L50-823 (BD; 558686): https://www.bdbiosciences.com/en-au/products/reagents/microscopy-imaging-reagents/
immunofluorescence-reagents/purified-mouse-anti-gata3.558686; cited in 7 publications
H3K27me3 C36B11 (Cell Signaling Technologies; 9733S): https://www.cellsignal.com/products/primary-antibodies/tri-methyl-
histone-h3-lys27-c36b11-rabbit-mab/9733; cited in 759 publications
OCT4 (SantaCruz; sc-5279): https://www.scbt.com/sv/p/oct-3-4-antibody-c-10; cited in 2201 publications
SOX2 clone EP103 (Biogenex; AN833): https://biogenex.com/wp-content/uploads/2019/11/932-833N.pdf; cited in 4 publications
NANOG (RnD; AF1997-SP): https://www.rndsystems.com/products/human-nanog-antibody_af1997; cited in 166 publications
EED (E4L6E) XP® (Cell Signaling Technology; 85322): https://www.cellsignal.com/products/primary-antibodies/eed-e4l6e-xp-rabbit-
mab/85322; cited in 4 publications
H3 (Active motif 39763): https://www.activemotif.com/catalog/details/39763; cited in 23 publications
H3K4me3 (Millipore 04-745) : https://www.merckmillipore.com/SE/en/product/Anti-trimethyl-Histone-H3-Lys4-Antibody-clone-
MC315-rabbit-monoclonal,MM_NF-04-745; cited in 98 publications
H3K27me3 (Millipore 07-449): https://www.merckmillipore.com/SE/en/product/Anti-trimethyl-Histone-H3-Lys27-
Antibody,MM_NF-07-449?ReferrerURL=https%3A%2F%2Fwww.google.com%2F; cited in > 200 publications
H2Aub (Cell Signaling 8240S): https://www.cellsignal.com/products/primary-antibodies/ubiquityl-histone-h2a-lys119-d27c4-xp-
rabbit-mab/8240; cited in 230 publications.

# Eukaryotic cell lines

Policy information about cell lines

Cell line source(s)

H9 (Wicell; WA09), HS975 (inhouse; DOI: 10.1038/ncomms4195, PMID: 24463987) , mouse inactivated embryonic
fibroblasts ( Gibco; A24903)

Authentication

Cell lines were not further authenticated.

Mycoplasma contamination

All the cell lines are tested negative for mycoplasma contamination.

Commonly misidentified lines
(See ICLAC register)

No cell line used in this paper is listed in the ICLAC database.

# ChIP-seq

## Data deposition

☒ Confirm that both raw and final processed data have been deposited in a public database such as GEO.

☒ Confirm that you have deposited or provided access to graph files (e.g. BED files) for the called peaks.

Data access links
*May remain private before publication.*

GSE181244: https://www.ncbi.nlm.nih.gov/geo/query/acc.cgi?acc=GSE181244

Files in database submission

MINUTE-ChIP: Demultiplexed FASTQ files, scaled bigWig files.
RNA-seq: FASTQ files, normalized bigWig files.

Genome browser session
(e.g. UCSC)

n/a

## Methodology

Replicates

Three biological replicates

Sequencing depth

ChIP seq replicates were sequenced at a average depth of 6 million paired-end reads.

Antibodies

5 ug each of H3K27me3 {Millipore 07-449}, H3K4me3 {Millipore 04-745} and H2AUb {Cell Signaling 8240S}

| | |
|---|---|
| Peak calling parameters | No peak calling was done for this analysis. |
| Data quality | Multiple QC checks were performed throughout the analysis (FastQC, Picard repeat stats, insert size, estimated library size). |
| Software | All software used is listed in the methods section of the manuscript. minute pipeline is paired with a conda environment for reproducibility. nf-core RNA-seq primary analysis pipeline was pulled from the repository and run on a Singularity container. Downstream analysis is available on GitHub and rendered as a website with workflowr v1.6.2. |

