## [Peer Review File · Nature Cell Biology]

Peer Review Information

Journal: Nature Cell Biology

Manuscript Title: Polycomb Repressive Complex 2 shields naïve human pluripotent cells from trophectoderm differentiation

Corresponding author names: Fredrik Lanner, Simon J Elsässer

Editorial Notes:

Redactions – unpublished data	Parts of this Peer Review File have been redacted as indicated to maintain the confidentiality of unpublished data.
Redactions – confidential patient information	Parts of this Peer Review File have been redacted as indicated to maintain patient confidentiality.
Redactions – published data	Parts of this Peer Review File have been redacted as indicated to remove third-party material.

Reviewer Comments & Decisions:

Decision Letter, initial version:

Subject: Decision on Nature Cell Biology submission NCB-E46376
Message: *Please delete the link to your author homepage if you wish to forward this email to co-authors.

Dear Dr Elsässer,

Your manuscript, "Polycomb Repressive Complex 2 shields naïve human pluripotent cells from trophoblast differentiation", has now been seen by 3 referees, who are experts in polycomb complex and stem cells (referees 1 and 3) and trophoblasts (referee 2). As you will see from their comments (attached below) they find this work of potential interest, but have raised substantial concerns, which in our view would need to be addressed with considerable revisions before we can consider publication in Nature Cell Biology.

Nature Cell Biology editors discuss the referee reports in detail within the editorial team, including the chief editor, to identify key referee points that should be addressed with priority, and requests that are overruled as being beyond the scope of the current study. To guide the scope of the revisions, I have listed these points below. We are committed to providing a fair and constructive peer-review process, so please feel free to contact me if you would like to discuss any of the referee comments further.

In particular, it would be essential to:

a) strengthen the claim that PRC2 shields naïve human pluripotent cells from trophoblast differentiation, as noted by:

Reviewer #1:

Secondly, PRC2 had been shown dispensable in naïve cells, as pluripotency markers did not change. In line with that, Kumar et al. also find that the cells grow the same way in the PRC2 inhibited state, but a fraction of cells lose pluripotency, referred to it as a stochastic event, and a subset of bivalent genes are derepressed. In their bulk assay, they see a loss of pluripotency and gain of new markers such as GATA3, though, in the discussion section, they state the majority of cells maintain pluripotency and do not gain GATA3. The latter point is worth addressing in more detail and possibly adjusting the title. If only a

2fraction turns trophectoderm genes on and transitions that way, then the title and some claims seem a bit misleading (more below).

- The authors put a major focus on the trophectoderm differentiation potential (see, e.g., title, discussion, though Figure 1-4 cover other aspects); moreover, the data shows (Figure 5a) that markers of multiple extraembryonic lineages are upregulated upon EZH2i (pointed out in a paragraph heading by the authors). With the current data presented, it is not possible to tell if lineage restriction by PRC2 in the naïve state is most prominent for the trophectoderm lineage; thus, the title is not perfectly fitting.

Reviewer #3:

(2) authors must provide additional information to strengthen their conclusions regarding the functional role of PRC2 in lineage restriction in hESCs.

(2) As acknowledged by the authors in their discussion, the suggested functionality of PRC2 on restricting the trophectodermal lineage in naïve hESCs is certainly surprising, as two previous studies indicate a dispensable function of PRC2 for the maintenance of the naïve pluripotent state in hESCs (PMID: 28864533; PMID: 28939884). The authors provided two potential explanations for this inconsistency: (1) the stochasticity differentiation towards trophectoderm of a fraction of hESCs, which might be overlooked in the previous studies; or, (2) a more pronounced phenotype observed in this study by using an EZH2/1 inhibitor, in contrast to a previous study (PMID: 28939884) in which EZH1 expression was reduced in an EZH2-KO background.

However, in the first case, the potential stochasticity is not evaluated in the current study. And, in the second case, the current study associates the functional impact of EPZ-6438 treatment to the loss of H3K27me3, although non-specific or secondary effects (e.g. the observed global lost H3K4me3) are not experimentally ruled out. Thus, considering that the role of PRC2 in sustaining naïve hESCs identity is an important conclusion raised by the authors, it needs to be further supported at the experimental level.

a) by single-cell RNA sequencing (naïve hESCs +/- EZH2 inh.) to evaluate the proportion of hESCs in naïve state that spontaneously transient to trophectoderm upon EZH2 inhibitor.

b) Evaluating the functional impact of full KO of PRC2 core components SUZ12 or EED in naïve hESCs (loss of pluripotency markers, gain of trophectoderm lineage).

b) All other referee concerns pertaining to strengthening existing data, providing controls, methodological details, clarifications and textual changes, should also be addressed.

c) Finally please pay close attention to our guidelines on statistical and methodological reporting (listed below) as failure to do so may delay the reconsideration of the revised manuscript. In particular please provide:

We would be happy to consider a revised manuscript that would satisfactorily address these points, unless a similar paper is published elsewhere, or is accepted for publication in Nature Cell Biology in the meantime.

- ensure that it conforms to our format instructions and publication policies (see below and <https://www.nature.com/nature/for-authors>).

- provide a point-by-point rebuttal to the full referee reports verbatim, as provided at the end of this letter.

- provide the completed Reporting Summary (found here <https://www.nature.com/documents/nr-reporting-summary.pdf>). This is essential for reconsideration of the manuscript will be available to editors and referees in the event of peer review. For more information see <http://www.nature.com/authors/policies/availability.html> or contact me.

When submitting the revised version of your manuscript, please pay close attention to our [href="https://www.nature.com/nature-research/editorial-policies/image-integrity">Digital Image Integrity Guidelines](https://www.nature.com/nature-research/editorial-policies/image-integrity). and to the following points below:

Nature Cell Biology is committed to improving transparency in authorship. As part of our efforts in this direction, we are now requesting that all authors identified as 'corresponding author' on published papers create and link their Open Researcher and Contributor Identifier (ORCID) with their account on the Manuscript Tracking System (MTS), prior to acceptance. ORCID helps the scientific community achieve unambiguous attribution of all scholarly contributions. You can create and link your ORCID from the home page of the MTS by clicking on 'Modify my Springer Nature account'. For more information please visit www.springernature.com/orcid.

This journal strongly supports public availability of data. Please place the data used in your paper into a public data repository, or alternatively, present the data as Supplementary Information. If data can only be shared on request, please explain why in your Data Availability Statement, and also in the correspondence with your editor. Please note that for some data types, deposition in a public repository is mandatory - more information on our data deposition policies and available repositories appears below.

[REDACTED]

We would like to receive a revised submission within six months.

We hope that you will find our referees' comments, and editorial guidance helpful. Please do not hesitate to contact me if there is anything you would like to discuss.

Best wishes,

Jie Wang

Jie Wang, PhD
Senior Editor
Nature Cell Biology

Tel: +44 (0) 207 843 4924
email: jie.wang@nature.com

Reviewers' Comments:

Reviewer #1:

Remarks to the Author:

Kumar et al. investigate the role of Polycomb repressive complex 2 (PRC2) in human naïve and primed pluripotent stem cells. By performing MINUTE-ChIP of three histone modifications, the authors compare the epigenetic status between naïve and primed female H9 hESCs, in untreated and EZH2 inhibitor-treated conditions. They show that naïve hESCs have higher levels of H3K27me3 and H2Aub genome-wide compared to primed hESCs, and the majority of the H2Aub deposition is independent of H3K27me3. Unlike primed hESCs, naïve hESCs have elevated H3K27me3 on the X chromosomes, but this does not contribute to dosage compensation. The authors identify bivalent promoters which are specific to the naïve or primed state or present in both. Depletion of H3K27me3 resulted in changes in gene expression for a certain subset of bivalent genes including GATA3. Based on the derepressed genes in the naïve cells upon EZH2 inhibition, the authors suggest similarity to a rare population normally present in the naïve state and that PRC2 plays a role in counteracting gene expression of extraembryonic lineage markers although only a small fraction activates those in the treated naïve cells.

Overall, the data appear convincing and the presentation is robust (see below for a few visual improvements). Interestingly, the results presented in the current manuscript appear to correct and refine previous findings. Others showed lower global levels of H3K27me3 in naïve compared to primed,

5while Kumar et al. uses quantitative MINUTE-ChIP and identifies ~3-fold higher level, in line with recent mass-spectrometry-based data. However, some select loci, e.g., *Dusp6*, shows the same trend both in the current manuscript and Ref. 17. Furthermore, Ref. 14 states that “H3K27me3 peaks undergo genomic redistribution and become preferentially depleted from promoters and gene-body regions, rather than from intergenic regions”. Secondly, PRC2 had been shown dispensable in naïve cells, as pluripotency markers did not change. In line with that, Kumar et al. also find that the cells grow the same way in the PRC2 inhibited state, but a fraction of cells lose pluripotency, referred to it as a stochastic event, and a subset of bivalent genes are derepressed. In their bulk assay, they see a loss of pluripotency and gain of new markers such as *GATA3*, though, in the discussion section, they state the majority of cells maintain pluripotency and do not gain *GATA3*. The latter point is worth addressing in more detail and possibly adjusting the title. If only a fraction turns trophoderm genes on and transitions that way, then the title and some claims seem a bit misleading (more below).

Major comments:

- The authors put a major focus on the trophoderm differentiation potential (see, e.g., title, discussion, though Figure 1-4 cover other aspects); moreover, the data shows (Figure 5a) that markers of multiple extraembryonic lineages are upregulated upon EZH2i (pointed out in a paragraph heading by the authors). With the current data presented, it is not possible to tell if lineage restriction by PRC2 in the naïve state is most prominent for the trophoderm lineage; thus, the title is not perfectly fitting.
- The authors state that the overall morphology of EZH2i naïve cells does not change, and only a fraction expresses *GATA3*. Quantification of that fraction would be important to assess the extent of trophoderm potential (Figure 5c, d).
- Co-staining pluripotency markers and TE markers (*Gata3*, *Cdx2*) can show if it is mutually exclusive or not.
- Figure 5e shows that H2Aub is still present over some of the now expressed loci, which seems to be in line with the heterogeneity in response. This should be quantified better.
- Staining for other extraembryonic lineage markers (e.g., *Sox17* and *Gata4* for PrE) and quantifying the fraction of cells gaining differentiation potential is recommended to determine the gain of differentiation potential of another extraembryonic lineage. Similar suggestion for amnion and yolk sac mesoderm markers.
- The authors compare the EZH2i condition to the intermediate population in naïve cells identified by Messmer et al. There, in single cells, *GATA3* and other markers (Figure 4d, e) are co-expressed. If the similarity of intermediate cells to EZH2i naïve cells is present, as suggested by the authors, the marker genes would be co-expressed in single cells. Staining of these markers (ideally different extraembryonic lineages) or single-cell analysis is recommended to address that.

- The authors identify naïve-specific and primed-specific sets of bivalent promoters. When analyzing the corresponding gene expression changes (Figure 3c), they conclude that EZH2i treatment in naïve cells derepresses naïve-only genes but not primed-only genes. However, they conclude that the same level of change for primed-only genes in EZH2i-treated primed cells is derepression. This conclusion is not supported by the data. A violin plot and heatmap of RNAseq are suggested to complement Figure 2b. Presenting the fraction of genes changing expression upon EZH2i treatment for both the naïve-specific and the primed-specific sets is suggested.
- For TFAP2C, a naïve-specific gene, Kumar et al. show that it is bivalent in the primed state, and depleting H3K27me3 leads to basal activation therein. It would be useful to include this gene in Figure 3e, as the authors base that analysis on this finding. They start by “To generalize this observation,” however, as the data does not support it would work like this in general for naïve-specific pluripotency genes, it is important to highlight that in the text and avoid generalization.
- XIST expression seems lower in Figure 2c, and the H3K4me3 level is also lower in EZH2 treated cells, however in Figure 2g, it is indicated with black color that XIST does not change. Clarification would be useful.
- The EZH2i is a key component of the manuscript and while some analysis is provided (reads and IF), it would be important to show a western blot time course for the treatment. H3K27me3 would be most relevant but other could also be included.
- In Figure 1c the effect on DNA methylation is briefly assessed. Overall the chromHMM shows limited differences per features as the authors note. Some features lose indeed H3K27 methylation and gain DNA methylation, but others seem to maintain or even gain (especially the bivalent promoters) H3K27 and still gain DNA methylation. This needs to be broken down in more detail.
- The ChIPs are quantitative and it may be good to discuss some of the changes in enrichment further in the discussion (which is currently more of a recap). How does one have to think about the few-fold changes (up and down) mechanistically...more/less modified histones per locus or more cells that have the enrichment since it's a bulk assessment.

Minor comments:

- In the text, there are instances where it is not entirely clear which condition is described (naïve versus primed or untreated versus treated).
- The wording H3K27me3 hypermethylation is a bit confusing. The lysine can't be further methylated, so its of course more about the local enrichment.
- Figure 2b: scale missing
- Figure 2c is very small and has many details, while Figure 2d is almost the same size. This can be balanced better between panels.
- Figure 2d: color scale missing

- Figure 3a: data visualization in the alluvial plot and the corresponding main text does not help the reader to easily understand the presented results. The three groups described in the text could be shown in a more effective way. Also the panel is huge given its limited information.
- Figure 3c: instead of a violin plot for all genes, a violin plot for the class-specific genes is suggested.
- Figure 4c: The labels on the Venn diagram are unclear, and most numbers described in the corresponding main text do not fit the numbers on the plot, thus making it difficult to follow.
- Figure 4e: color scale missing
- Figure 5a: indicate Gata3 on the plot (importantly for TE) and provide clearer annotation of gene names to corresponding dots.

Reviewer #2:

Remarks to the Author:

In their manuscript, Kumar et al. report that H3K27me3 is hypermethylated in naïve human PSCs and also a new naïve-specific set of bivalent promoters. By inhibiting EZH1/2, the enzymatic subunits of PRC2, H3K27me3 was depleted without changing the H3K4me3 status. Especially, PRC2 inhibition depleted naïve PSC-specific H3K27me3. Interestingly, trophoctoderm genes were upregulated after the depletion of H3K27me3 in naïve human PSCs but not in primed human PSCs.

The findings provide interesting insight on why naïve human PSCs can differentiate into trophoctoderm. However, I have several concerns before publication.

One issue in that the manuscript confuses naïve-derived trophoctoderm-like cells and primed-derived BMP-induced cells. The authors often refer to previous reports regarding primed PSC-derived BMP-induced cells but then perform experiments using naïve PSCs(p16-17). Naïve and primed PSCs are different cell types. In addition, recent reports suggest that BMP-induced cells are a counterpart of the amnion-like state, while naïve PSC-derived cells are trophoblast lineage cells (Guo et al., Cell Stem Cell 2021; Io et al., Cell Stem Cell 2021; Zhou et al Nature 2019).

-The authors mentioned the similarity of H3K27me3 hypermethylation in naïve human and mouse PSCs. If true, can they provide some explanation for why mouse ES cells are difficult to differentiate into trophoctoderm-like cell type without transgene-expression? If exposed to PRC2 inhibitor, do naïve mouse PSCs differentiate into trophoctoderm?

-Fig. 3: Text and Figs do not match. Please revise.

-The authors observed that EZH2i treatment caused naïve PSCs to express trophoblast marker genes, but in Fig. 5a, amnion marker genes also seem to be upregulated. Since both amnion and trophoblast express many of the same genes, distinguishing the two by gene sets is difficult. The authors should consider PCA or UMAP using naïve (EZH2i +/-) and primed (EZH2i+/-) cells and human embryo data including trophoblast and amnion.

-Fig. 5c, 5d: The GATA3 expression in Fig. 5d is very heterogeneous. The authors should co-stain pluripotency genes and GATA3.

- Other ES/iPS cell lines should be used to confirm their observations.

Minor points

-KLF2 and SOX11 are not shown in Fig. 1e.

-Fig. 5a Please show each gene set and the expression level in a table. Color codes look the opposite of what is described in the legend.

-Extended Data Fig.3 c, d, e, Fig. 4 a, c. What are the color codes?

Reviewer #3:

Remarks to the Author:

In this manuscript, Kumar and collaborators aimed to clarify the molecular mechanisms that limit the conversion of human embryonic stem cells (hESCs) into extra-embryonic cell lineages. For that, they performed a quantitative epigenome profiling (MINUTE-ChIP), on H9 female hESCs, for three specific histone posttranslational modifications (H3K4me3, H3K27me3, and H2AK119ub), together with transcriptome analysis, in four different experimental conditions (naïve/primed-state plus/minus EZH2 inhibitor; EPZ-6438). By performing complex computational analysis, the authors provide an exhaustive characterization of the distribution of these histone modifications on naïve and primed hESCs. They use this information to categorize different genomic loci, according to the abundance of each histone mark, with a potential impact hESCs lineage restriction. Upon EPZ-6438 treatment, the authors analysed (1) the expression profile of lineage-specific genes and, (2) performed immunofluorescent of pluripotency factors (e.g. NANOG, SOX2, and OCT3/4) and GATA3, the master TF for trophoblast development. These analyses enable the authors to postulate a role of the Polycomb Repressive Complex 2 restricting the lineage potential of hESCs.

How the cell types are established and maintained during early human development, while other alternative lineages are restricted, is an open and interesting question for both fundamental and

biomedical research. In this sense, the characterization of the epigenome dynamics in hESCs at different stages could provide relevant insights into the molecular mechanism controlling cell type specification *in vivo*. For that, we consider that this is a very useful resource manuscript, with intensive computational analysis, and with some relevant functional information included.

However, (1) we encourage the authors to provide a revision of the narrative of the text, to increase and facilitate understanding; and, (2) authors must provide additional information to strengthen their conclusions regarding the functional role of PRC2 in lineage restriction in hESCs.

Main points:

(1) The amount of interesting epigenomic and transcriptomic data generating or this manuscript is clear. However, we are afraid that, despite being a topic very much related to the research area in your lab, it has been extremely difficult to follow the narrative of the manuscript, which seemed to be 'drammatically' rewritten!

In addition, using quantitative epigenome profiling, the authors provide a comprehensive categorization of different genomic features. These categories are characterized by gain/loss of histone marks and/or gain/loss of expression in different experimental conditions. To increase the readability of this resource, we would suggest including an informative table with all genomic features categorized in this study (active promoters, bivalent promoters, bivalent promoters *de novo*, primed-only, naïve-only, common bivalent, H3K4me-only promoters, etc), thus including their abundance and the criteria used for their categorization.

(2) As acknowledged by the authors in their discussion, the suggested functionality of PRC2 on restricting the trophectodermal lineage in naïve hESCs is certainly surprising, as two previous studies indicate a dispensable function of PRC2 for the maintenance of the naïve pluripotent state in hESCs (PMID: 28864533; PMID: 28939884). The authors provided two potential explanations for this inconsistency: (1) the stochasticity of differentiation towards trophectoderm of a fraction of hESCs, which might be overlooked in the previous studies; or, (2) a more pronounced phenotype observed in this study by using an EZH2/1 inhibitor, in contrast to a previous study (PMID: 28939884) in which EZH1 expression was reduced in an EZH2-KO background.

However, in the first case, the potential stochasticity is not evaluated in the current study. And, in the second case, the current study associates the functional impact of EPZ-6438 treatment to the loss of H3K27me₃, although non-specific or secondary effects (e.g. the observed global loss of H3K4me₃) are not experimentally ruled out. Thus, considering that the role of PRC2 in sustaining naïve hESCs identity is an important conclusion raised by the authors, it needs to be further supported at the experimental level.

a) by single-cell RNA sequencing (naïve hESCs +/- EZH2 inh.) to evaluate the proportion of hESCs in naïve state that spontaneously transient to trophoctoderm upon EZH2 inhibitor.

b) Evaluating the functional impact of full KO of PRC2 core components SUZ12 or EED in naïve hESCs (loss of pluripotency markers, gain of trophoctoderm lineage).

Check figure references:

- Figure 5a, “up” and “down” in the legend are switched

- Extended Data Figure 10a is not a barchart, as it says in the figure description, and it does not show what the manuscript refers to.

- In the results paragraph: “whereas primed-only bivalent genes were predominantly higher expressed (..) in naive (Fig.3d -> 3c...) ”

- In the discussion paragraph: “on the other hand, we observed an overall reduction of core pluripotency marker expression on the population level (Extended Data Fig. 7d -> 8c) ”

Extra comments on the comparison of the two co-submissions:

At the functional level, although both studies point towards an implication of PRC2 in pluripotent-to-trophoblasts transition, there are relevant discrepancies among both studies. Zijlmans et al. findings, well-supported at multiple experimental levels, indicate a function of PRC2 as a barrier during trophoblast induction. This is because, in the absence of instructive signals, PRC2 inhibition does not cause spontaneous differentiation or loss of pluripotency marks in naïve hPSCs. This dispensable role of PRC2 in naïve culture conditions is an observation that seems in line with previously published studies (PMID: 28864533; PMID: 28939884). Only when cultured under trophoblast differentiation media, hPSCs transient more efficiently towards trophoblast cells in the presence of the EZH2 inhibitor. In contrast, Kumar et al. study shows that the treatment with EPZ-6438 for 7 days results in the spontaneous differentiation of a fraction of naïve hESCs (yet to be quantified) towards trophoblast cells, in the absence of inductive signals. This would suggest that PRC2 functions as the active blocker of trophoblast differentiation. We agree that the discrepancies might result from technical differences between both studies.

Methods should be written concisely, but should contain all elements necessary to allow interpretation and replication of the results. As a guideline, Methods sections typically do not exceed 3,000 words. The Methods should be divided into subsections listing reagents and techniques. When citing previous methods, accurate references should be provided and any alterations should be noted. Information must be provided about: antibody dilutions, company names, catalogue numbers and clone numbers for monoclonal antibodies; sequences of RNAi and cDNA probes/primers or company names and catalogue numbers if reagents are commercial; cell line names, sources and information on cell line identity and authentication. Animal studies and experiments involving human subjects must be reported in detail, identifying the committees approving the protocols. For studies involving human subjects/samples, a statement must be included confirming that informed consent was obtained. Statistical analyses and information on the reproducibility of experimental results should be provided in a section titled “Statistics and Reproducibility”.

All Nature Cell Biology manuscripts submitted on or after March 21 2016 must include a Data availability statement as a separate section after Methods but before references, under the heading "Data Availability". For Springer Nature policies on data availability see <http://www.nature.com/authors/policies/availability.html>; for more information on this particular policy see <http://www.nature.com/authors/policies/data/data-availability-statements-data-citations.pdf>. The Data availability statement should include:

- Accession codes for primary datasets (generated during the study under consideration and designated as "primary accessions") and secondary datasets (published datasets reanalysed during the study under consideration, designated as "referenced accessions"). For primary accessions data should be made public to coincide with publication of the manuscript. A list of data types for which submission to community-endorsed public repositories is mandated (including sequence, structure, microarray, deep sequencing data) can be found here <http://www.nature.com/authors/policies/availability.html#data>.
- Unique identifiers (accession codes, DOIs or other unique persistent identifier) and hyperlinks for datasets deposited in an approved repository, but for which data deposition is not mandated (see here for details <http://www.nature.com/sdata/data-policies/repositories>).
- At a minimum, please include a statement confirming that all relevant data are available from the authors, and/or are included with the manuscript (e.g. as source data or supplementary information), listing which data are included (e.g. by figure panels and data types) and mentioning any restrictions on availability.
- If a dataset has a Digital Object Identifier (DOI) as its unique identifier, we strongly encourage including this in the Reference list and citing the dataset in the Methods.

We recommend that you upload the step-by-step protocols used in this manuscript to the Protocol Exchange. More details can found at www.nature.com/protocolexchange/about.

FIGURES – Colour figure publication costs \$600 for the first, and \$300 for each subsequent colour figure. All panels of a multi-panel figure must be logically connected and arranged as they would appear in the

final version. Unnecessary figures and figure panels should be avoided (e.g. data presented in small tables could be stated briefly in the text instead).

All imaging data should be accompanied by scale bars, which should be defined in the legend. Cropped images of gels/blots are acceptable, but need to be accompanied by size markers, and to retain visible background signal within the linear range (i.e. should not be saturated). The boundaries of panels with low background have to be demarked with black lines. Splicing of panels should only be considered if unavoidable, and must be clearly marked on the figure, and noted in the legend with a statement on whether the samples were obtained and processed simultaneously. Quantitative comparisons between samples on different gels/blots are discouraged; if this is unavoidable, it should only be performed for samples derived from the same experiment with gels/blots were processed in parallel, which needs to be stated in the legend.

- We do not recommend using Adobe Photoshop for designing figures, but we can accept Photoshop generated (.PSD or .TIFF) files only if each element included in the figure (text, labels, pictures, graphs, arrows and scale bars) are on separate layers. All text should be editable in 'type layers' and line-art

15such as graphs and other simple schematics should be preserved and embedded within 'vector smart objects' - not flattened raster/bitmap graphics.

The total number of Supplementary Figures (not including the “unprocessed scans” Supplementary Figure) should not exceed the number of main display items (figures and/or tables (see our Guide to Authors and March 2012 editorial <http://www.nature.com/ncb/authors/submit/index.html#suppinfo>; <http://www.nature.com/ncb/journal/v14/n3/index.html#ed>). No restrictions apply to Supplementary Tables or Videos, but we advise authors to be selective in including supplemental data.

GUIDELINES FOR EXPERIMENTAL AND STATISTICAL REPORTING

REPORTING REQUIREMENTS – We are trying to improve the quality of methods and statistics reporting in our papers. To that end, we are now asking authors to complete a reporting summary that collects information on experimental design and reagents. The Reporting Summary can be found here <https://www.nature.com/documents/nr-reporting-summary.pdf> If you would like to reference the guidance text as you complete the template, please access these flattened versions at <http://www.nature.com/authors/policies/availability.html>.

STATISTICS – Wherever statistics have been derived the legend needs to provide the n number (i.e. the sample size used to derive statistics) as a precise value (not a range), and define what this value

represents. Error bars need to be defined in the legends (e.g. SD, SEM) together with a measure of centre (e.g. mean, median). Box plots need to be defined in terms of minima, maxima, centre, and percentiles. Ranges are more appropriate than standard errors for small data sets. Wherever statistical significance has been derived, precise p values need to be provided and the statistical test used needs to be stated in the legend. Statistics such as error bars must not be derived from $n < 3$. For sample sizes of $n < 5$ please plot the individual data points rather than providing bar graphs. Deriving statistics from technical replicate samples, rather than biological replicates is strongly discouraged. Wherever statistical significance has been derived, precise p values need to be provided and the statistical test stated in the legend.

Author Rebuttal to Initial comments

We would like to thank all reviewers for their insightful comments and suggestions. As we detail below, in addition to various additional analyses, we have performed major experiments to address the two key concerns:

- 1) We have added additional functional experiments that strengthen the claim that PRC2 shields naïve cells from differentiation. We have repeated and expanded our GATA3 staining experiments to include a second, orthogonal inhibitor of PRC2 (EED226, Fig 4b), performed the same experiment in another naïve hESC line (Extended Data Fig 10a), and we have performed an

18acute genetic targeting of EED using synthetic CRISPR/Cas9 complex (Fig 4e-g, Extended Data Fig 10b). All of these experiments agree on the spontaneous induction of GATA3⁺ cells within 7d of PRC2 inhibition/deletion.

- 2) We have performed single-cell RNA-sequencing of a time course of EZH2 inhibition in naïve hESC, including controls in primed hESC and detailed analysis (Figs 5-6), which provide substantial new insights in addition to validating all our previous conclusions.

In an effort to streamline the manuscript, we have combined original Figs 1 and 2 into new Fig 1 to accommodate an in-depth analysis of the new scRNA-seq data (Figs. 5-6). We have substantially rewritten the main text and also removed a number of sections that became peripheral to the overall scope of the manuscript. Due to the substantial editing, we note that the track-change version of the revised manuscript became quite fragmented, and we apologize for the inconvenience.

As detailed in the point-by-point responses below, we have adjusted the title of our manuscript to reflect the additional insight gained from scRNA-seq (but fully consistent with our prior bulk data), that mesoderm lineage cells are generated in addition to trophectoderm.

Reviewers' Comments:

Reviewer #1:

Remarks to the Author:

Kumar et al. investigate the role of Polycomb repressive complex 2 (PRC2) in human naïve and primed pluripotent stem cells. By performing MINUTE-ChIP of three histone modifications, the authors compare the epigenetic status between naïve and primed female H9 hESCs, in untreated and EZH2 inhibitor-treated conditions. They show that naïve hESCs have higher levels of H3K27me3 and H2Aub genome-wide compared to primed hESCs, and the majority of the H2Aub deposition is independent of H3K27me3. Unlike primed hESCs, naïve hESCs have elevated H3K27me3 on the X chromosomes, but this does not contribute to dosage compensation. The authors identify bivalent promoters which are specific to the naïve or primed state or present in both. Depletion of H3K27me3 resulted in changes in gene expression for a certain subset of bivalent genes including GATA3. Based on the derepressed genes in the naïve cells upon EZH2 inhibition, the authors suggest similarity to a rare population normally present in the naïve state and that PRC2 plays a role in counteracting gene

19expression of extraembryonic lineage markers although only a small fraction activates those in the treated naïve cells.

Overall, the data appear convincing and the presentation is robust (see below for a few visual improvements). Interestingly, the results presented in the current manuscript appear to correct and refine previous findings. Others showed lower global levels of H3K27me3 in naïve compared to primed, while Kumar et al. uses quantitative MINUTE-ChIP and identifies ~3-fold higher level, in line with recent mass-spectrometry-based data. However, some select loci, e.g., *Dusp6*, shows the same trend both in the current manuscript and Ref. 17. Furthermore, Ref. 14 states that “H3K27me3 peaks undergo genomic redistribution and become preferentially depleted from promoters and gene-body regions, rather than from intergenic regions”.

We thank the reviewer for this assessment and we agree that our study corrects and refines conclusions drawn by prior non-quantitative studies.

Secondly, PRC2 had been shown dispensable in naïve cells, as pluripotency markers did not change. In line with that, Kumar et al. also find that the cells grow the same way in the PRC2 inhibited state, but a fraction of cells lose pluripotency, referred to it as a stochastic event, and a subset of bivalent genes are derepressed. In their bulk assay, they see a loss of pluripotency and gain of new markers such as *GATA3*, though, in the discussion section, they state the majority of cells maintain pluripotency and do not gain *GATA3*. The latter point is worth addressing in more detail and possibly adjusting the title. If only a fraction turns trophectoderm genes on and transitions that way, then the title and some claims seem a bit misleading (more below).

This is an important question which must arise when bulk measurements are acquired from a cell population that is heterogeneous. We were also worried about this fact and we have thus carried out a comprehensive single-cell RNA-seq study (new Figs. 5 and 6). By integrating the profiles of naïve cultures with extensive embryo references we detect a small fraction (1.8%) of trophectoderm-like cells (TLC) and 1.0% mesoderm-like cells (MeLC) confirming a low grade spontaneous differentiation also within naïve culture conditions (Fig. 5). With EZH2i inhibition these two populations increased to 9.1% and 11.7% respectively after 7 days. This is in good agreement with our immunofluorescent stainings of *GATA3*, which yielded ~12% *GATA+* cells after 7d EZH2i treatment. The transcriptional analysis also resolved a distinct trajectory where cells first exit the naïve “groundstate” and become “activated” with increased

lineage markers and reduced expression of pluripotency markers such as NANOG and SOX2, but not POU5F1. Following this activation, the cell trajectory bifurcates in two paths towards TLC and MeLC with corresponding induction of lineage markers such as HAND1, CDX1, TBXT, GATA2&3. In the last step where the treated cells adopt TLC and MeLC states POU5F1 is finally reduced most probably resulting in irreversible lineage commitment. Within 7 days, the majority of the treated cells have transitioned from the ground state into or beyond the activated state, suggesting that this is not just a stochastic event in a few cells but a robust response following the removal of an epigenetic barrier.

Major comments:

- The authors put a major focus on the trophectoderm differentiation potential (see, e.g., title, discussion, though Figure 1-4 cover other aspects); moreover, the data shows (Figure 5a) that markers of multiple extraembryonic lineages are upregulated upon EZH2i (pointed out in a paragraph heading by the authors). With the current data presented, it is not possible to tell if lineage restriction by PRC2 in the naïve state is most prominent for the trophectoderm lineage; thus, the title is not perfectly fitting.

We agree that this was an important question that remained inconclusive from our bulk RNA-seq data. Indeed, we have also found an alternative differentiation trajectory towards mesoderm, consistent with the upregulation of mesoderm markers we also observed in our prior Figure 5a (now expanded with additional lineages as Figure 3d). We have therefore followed the reviewer's suggestion and adjusted the title accordingly.

- The authors state that the overall morphology of EZH2i naïve cells does not change, and only a fraction expresses GATA3. Quantification of that fraction would be important to assess the extent of trophectoderm potential (Figure 5c, d).

The added scRNA-seq experiments address this question: between 2 and 4 days EZH2i treatment, the epiblast-like naïve hESCs (ELC) population is largely maintained, still accounting for ~95% of all cells at these time points (Fig 5b). At 7 days EZH2i, the fraction of ELC is reduced but still remains the largest population (77%). 211 out of 1613 cells assume a TE-like transcriptome whereas 219 cells assume a Mesoderm-like transcriptome. As described in resonance above, the scRNAseq analysis further

reveals that EZH2 inhibition changes the majority of the cells within the naïve cultures as they initiate lineage differentiation towards trophectoderm and mesoderm-like states. However this may not immediately result in a distinctive growth phenotype. Together, this explains why the naïve colonies still retain a normal morphology at 7 days EZH2i.

- Figure 5e shows that H2Aub is still present over some of the now expressed loci, which seems to be in line with the heterogeneity in response. This should be quantified better.

Indeed, we have in Fig 2d (previously Fig 3d) assessed the loss of H2Aub across two sets of bivalent genes, those that gain expression upon depletion of H3K27me3 and those that remain off: Only on the derepressed promoters we observed a 15% of H2Aub. Hence our data suggests a weak anticorrelation of transcriptional activity and H2Aub at bivalent promoters but does not resolve if the modest reduction in H2Aub is prerequisite or a consequence of gene activation.

We have now also included H2Aub heatmaps in Figs. 2e, 4a, EFig 12a

- Staining for other extraembryonic lineage markers (e.g., Sox17 and Gata4 for PrE) and quantifying the fraction of cells gaining differentiation potential is recommended to determine the gain of differentiation potential of another extraembryonic lineage. Similar suggestion for amnion and yolk sac mesoderm markers.

This is addressed by the added time-course scRNA-seq experiment already discussed in detail above. Mapping against a comprehensive reference annotation, we identify mesoderm and trophectoderm as the dominating lineages arising from EZH2i inhibition, with a very small percentage of amnion (Fig 5b).

- The authors compare the EZH2i condition to the intermediate population in naïve cells identified by Messmer et al. There, in single cells, GATA3 and other markers (Figure 4d, e) are co-expressed. If the similarity of intermediate cells to EZH2i naïve cells is present, as suggested by the authors, the marker genes would be co-expressed in single cells. Staining of these markers (ideally different extraembryonic lineages) or single-cell analysis is recommended to address that.

This is addressed by the time-course scRNA-seq experiment already discussed in detail above. Consequently, the analysis of Messmer et al. presented in the original 4d,e

became redundant and we have removed it in the revised manuscript. However, we have validated that the 'intermediate population' described there expresses similar markers as our TLC and MeLC populations, hence adding further support that these lineages are generated from naïve hESC cultures spontaneously at a low percentage (Extended Data Fig. 11d).

- The authors identify naïve-specific and primed-specific sets of bivalent promoters. When analyzing the corresponding gene expression changes (Figure 3c), they conclude that EZH2i treatment in naïve cells derepresses naïve-only genes but not primed-only genes. However, they conclude that the same level of change for primed-only genes in EZH2i-treated primed cells is derepression. This conclusion is not supported by the data. A violin plot and heatmap of RNAseq are suggested to complement Figure 2b.

These conclusions are based on comparing the behavior of naïve-specific and primed-specific bivalent genes sets to all genes using statistical analysis and both adjusted p values from Wilcoxon test and Cohen's d effect sizes are given in the text.

We have now also included these values in the figure. The statistical test determines if the naïve and primed-specific bivalent gene sets are up- or down-regulated significantly *as a group*, hence individual fold-changes. E.g. the primed-specific bivalent gene set is significantly upregulated as a group in EZH2i-treated primed cells, but the same group of genes shows a non-significant response in EZH2i-treated naïve cells (some genes are derepressed but others are downregulated). We have clarified this in the result section to Fig. 2c (previously Fig 3c). Furthermore, we have added differential expression volcano plots for the EZH2i treatments of naïve (Fig 3a) and primed (EFig 7c) and for clarity the same groups as in Fig 2c are shown in the same colors in these plots.

Presenting the fraction of genes changing expression upon EZH2i treatment for both the naïve-specific and the primed-specific sets is suggested.

We have now expanded on this in Fig 3b.

- For TFAP2C, a naïve-specific gene, Kumar et al. show that it is bivalent in the primed state, and depleting H3K27me3 leads to basal activation therein. It would be useful to include this gene in Figure 3e, as the authors base that analysis on this finding.

We have now included TFAP2C in Fig 2e (former 3e), genome track is now EFig 7a

They start by “To generalize this observation,” however, as the data does not support it would work like this in general for naïve-specific pluripotency genes, it is important to highlight that in the text and avoid generalization.

We have substantially rewritten and streamlined the results section to provide a clearer flow and have also rephrased this.

- XIST expression seems lower in Figure 2c, and the H3K4me3 level is also lower in EZH2 treated cells, however in Figure 2g, it is indicated with black color that XIST does not change. Clarification would be useful.

Indeed XIST is not significantly up- or downregulated even though it appears so in the genome track. There are several reasons for this:

First, per-track normalization as done for genome browser visualization is not a precise way of normalizing. DESeq2 provides a statistical framework for properly normalizing datasets before calculating fold-changes. As a result, genome track and DESeq2 output may show small systematic differences. Second, the RNA-seq track is generated from one out of three biological replicates (always the same one to be consistent). We note that we did not explain this sufficiently in the figure legends since the ChIP-seq tracks show the combined triplicates. We have now clarified in the figure legends if individual replicates or a combined/average track is shown. In this particular case the replicates are quite variable in XIST expression and the tracks shown are thus not necessarily representative of the triplicate comparison (also the reason why DESeq2 returns no significant change).

Since we have shortened the discussion of X chromosome, we have also removed panel 2c from the figures, XIST is now only shown in chromosome overview EFig 4c.

- The EZH2i is a key component of the manuscript and while some analysis is provided (reads and IF), it would be important to show a western blot time course for the treatment. H3K27me3 would be most relevant but others could also be included.

We have now included a time course (0, 2, 4, 7 days EZH2i) as EFig 2e,f. The immunofluorescent staining demonstrates that H3K27me3 is consistently lost in all

cells with similar kinetics.

- In Figure 1c the effect on DNA methylation is briefly assessed. Overall the chromHMM shows limited differences per features as the authors note. Some features lose indeed H3K27 methylation and gain DNA methylation, but others seem to maintain or even gain (especially the bivalent promoters) H3K27 and still gain DNA methylation. This needs to be broken down in more detail.

We agree that this was only briefly touched upon in the text and did not provide new insight necessary to understand the further course of the manuscript. We decided to remove the DNA methylation panel and associated discussion entirely in the revised manuscript to streamline the text.

- The ChIPs are quantitative and it may be good to discuss some of the changes in enrichment further in the discussion (which is currently more of a recap). How does one have to think about the few-fold changes (up and down) mechanistically...more/less modified histones per locus or more cells that have the enrichment since it's a bulk assessment.

The scRNA-seq data helps addressing this question. In untreated cells, more than 96% of cells are homogeneous naïve and primed ESC according to the UMAP clustering. Hence the bulk ChIP and RNA-seq data comparing naïve and primed states quantitatively is valid in that it represents the average of a homogeneous population. After 7 days EZH2i, only 77% of cells remain in the ELC cluster, ~10% are MeLC and TLC. Hence it is clear that the bulk epigenome and transcriptome profiles represent this mixed population. However scRNA-seq, bulk RNA-seq and epigenome data agree very well on key trends, e.g. the downregulation of core pluripotency factors in the ELC population is mirrored in a reduction in bulk RNA-seq and a loss of H3K4me3 at the respective promoters (EFig 7b).

Minor comments:

- In the text, there are instances where it is not entirely clear which condition is described (naïve versus primed or untreated versus treated).

We have carefully checked that the comparisons are clearly mentioned

- The wording H3K27me3 hypermethylation is a bit confusing. The lysine can't

25be further methylated, so its of course more about the local enrichment.

The term is commonly used for DNA methylation levels, for which the same caveat exists (DNA hypermethylation does not imply more methyl on a C). Technically speaking, de Clerk et. al, Scientific Reports (2019) also demonstrate by mass spectrometry that the increase in H3K27me3 in naive cells comes with a decrease in me1, hence the term may be more appropriate to use for H3K27me3 then for DNA methylation. In any case we have removed the wording in many instances because it implies a higher-than-normal level but H3K27me3 levels in naïve cells are just normal for this cell type. We have used it in one instance to describe the higher-than-normal H3K27me3 accumulation on the X chromosomes.

- Figure 2b: scale

missing fixed

-Figure 2c is very small and has many details, while Figure 2d is almost the same size. This can be balanced better between panels.

fixed

- Figure 2d: color scale

missing fixed

- Figure 3a: data visualization in the alluvial plot and the corresponding main text does not help the reader to easily understand the presented results. The three groups described in the text could be shown in a more effective way. Also the panel is huge given its limited information.

We have condensed the panel

- Figure 3c: instead of a violin plot for all genes, a violin plot for the class-specific genes is suggested.

We believe it is important to interpret the class-specific changes against the background trend and hence we use the violin plot for all genes. Statistics on this comparison are now also included in the figure.

- Figure 4c: The labels on the Venn diagram are unclear, and most numbers described in the corresponding main text do not fit the numbers on the plot, thus making it difficult to follow.

We have corrected this, ensuring that the numbers in the venn diagram are correctly referred to in the text. Supplementary Table 2 also contains annotations to reproduce the numbers and associated gene sets.

- Figure 4e: color scale missing

We have replaced this with our own scRNA-seq data in Figure 6

- Figure 5a: indicate Gata3 on the plot (importantly for TE) and provide clearer annotation of gene names to corresponding dots.

fixed (now Figure 3d)

Reviewer #2:

Remarks to the Author:

In their manuscript, Kumar et al. report that H3K27me3 is hypermethylated in naïve human PSCs and also a new naïve-specific set of bivalent promoters. By inhibiting EZH1/2, the enzymatic subunits of PRC2, H3K27me3 was depleted without changing the H3k4me3 status. Especially, PRC2 inhibition depleted naïve PSC-specific H3K27me3. Interestingly, trophoctoderm genes were upregulated after the depletion of H3K27me3 in naïve human PSCs but not in primed human PSCs.

The findings provide interesting insight on why naïve human PSCs can differentiate into trophoctoderm. However, I have several concerns before publication.

One issue in that the manuscript confuses naïve-derived trophoctoderm-like cells and primed-derived BMP-induced cells. The authors often refer to previous reports regarding primed PSC-derived BMP-induced cells but then perform experiments using naïve PSCs(p16-17). Naïve and primed PSCs are different cell types. In addition, recent reports suggest that BMP-induced cells are a counterpart of the amnion-like state, while naïve PSC-derived cells are trophoblast lineage cells (Guo et al., Cell Stem Cell 2021; Io et al., Cell Stem Cell 2021; Zhou et al Nature 219).

We fully agree that naïve and primed stem cells are distinct in their potential to make trophoctoderm cells and did not want to suggest that they are comparable.

27Our single-cell data now clarifies that our naïve cell-derived GATA3+ cells cluster with true trophoblast lineage (a very small fraction with amnion) and also clearly shows that naïve cells do not transit through a post-implantation-like primed state (only a very small number of naïve cells maps to the primed cluster after EZH2i treatment) to give rise to trophoblast.

The authors mentioned the similarity of H3K27me3 hypermethylation in naïve human and mouse PSCs. If true, can they provide some explanation for why mouse ES cells are difficult to differentiate into trophoblast-like cell type without transgene-expression? If exposed to PRC2 inhibitor, do naïve mouse PSCs differentiate into trophoblast?

PRC2 has been studied quite well and it is a general agreement that mESC are strictly pluripotent, hence neither naïve nor primed mESC can form trophoblast. E.g. a study on naïve (2i ground state) mouse ESC with EED KO assessed transcriptional changes in the absence of PRC2 (van Mierlo et al. , 2019). The reported changes were relatively minor, in line with the fact that ground state and naïve mouse ESC in general (either LIF+serum or LIF+2i) are viable and maintain pluripotency. We have reanalyzed the published transcriptome dataset for TE markers and did not find a significant upregulation of these (with the exception of some induction of Krt7, Igf2, Gata2).

Together, these observations suggest that the naïve mouse ESC culture does not fully resemble human naïve conditions and/or the mechanism of TE induction is divergent between species. However, this is a common unknown in the field and not the scope of our study. Hence we have not further pursued this question.

-Fig. 3: Text and Figs do not match. Please revise.

We have revised this

-The authors observed that EZH2i treatment caused naïve PSCs to express trophoblast marker genes, but in Fig. 5a, amnion marker genes also seem to be upregulated. Since both amnion and trophoblast express many of the same genes, distinguishing the two by gene sets is difficult. The authors should consider PCA or UMAP using naïve (EZH2i +/-) and primed (EZH2i+/-) cells and human embryo data including trophoblast and amnion.

We fully agree that the transcriptional overlap between trophoblast and amnion has caused confusion in the literature. For this reason we have now followed the cell fates using a time course scRNA-seq, now presented in Figures 5-6. Within this figure we contrast the signatures with amnion and trophoblast references to distinguish between these two lineages. This analysis clearly confirms the induction of trophoblast-like cells over amnion cells.

-Fig. 5c, 5d: The GATA3 expression in Fig. 5d is very heterogeneous. The authors should co-stain pluripotency genes and GATA3.

□ done, include in Fig 4b

- Other ES/iPS cell lines should be used to confirm their observations.

We have now included EZH2 inhibition in an additional pluripotent stem cell line, HS975 (Extended Data Fig. 10a), resulting in emergence of GATA3+ cells only in the EZH2i-treated but not the untreated control.

Minor points

-KLF2 and SOX11 are not shown in Fig. 1e.

We apologize for this oversight, we had removed them from the figure since there was not enough space. They are now removed also from the text.

-Fig. 5a Please show each gene set and the expression level in a table. Color codes look the opposite of what is described in the legend.

29We apologize for this mistake, the colors were indeed swapped. The gene sets can already be found annotated in Supplementary Table 2.

-Extended Data Fig.3 c, d, e, Fig. 4 a, c. What are the color codes?

Thanks for pointing this out, we have now adjusted this.

Reviewer #3:

Remarks to the Author:

In this manuscript, Kumar and collaborators aimed to clarify the molecular mechanisms that limit the conversion of human embryonic stem cells (hESCs) into extra-embryonic cell lineages. For that, they performed a quantitative epigenome profiling (MINUTE-ChIP), on H9 female hESCs, for three specific histone posttranslational modifications (H3K4me3, H3K27me3, and H2AK119ub), together with transcriptome analysis, in four different experimental conditions (naïve/primed-state plus/minus EZH2 inhibitor; EPZ-6438). By performing complex computational analysis, the authors provide an exhaustive characterization of the distribution of these histone modifications on naïve and primed hESCs. They use this information to categorize different genomic loci, according to the abundance of each histone mark, with a potential impact hESCs lineage restriction. Upon EPZ-6438 treatment, the authors analysed (1) the expression profile of lineage-specific genes and, (2) performed immunofluorescent of pluripotency factors (e.g. NANOG, SOX2, and OCT3/4) and GATA3, the master TF for trophectoderm development. These analyses enable the authors to postulate a role of the Polycomb Repressive Complex 2 restricting the lineage potential of hESCs.

How the cell types are established and maintained during early human development, while other alternative lineages are restricted, is an open and interesting question for both fundamental and biomedical research. In this sense, the characterization of the epigenome dynamics in hESCs at different stages could provide relevant insights into the molecular mechanism controlling cell type specification in vivo. For that, we consider that this is a very useful resource manuscript, with intensive computational analysis, and with some relevant functional information included.

However, (1) we encourage the authors to provide a revision of the narrative of the text, to increase and facilitate understanding; and, (2) authors must provide additional information to strengthen their conclusions regarding the functional role of PRC2 in lineage restriction in hESCs.

30Main points:

(1) The amount of interesting epigenomic and transcriptomic data generating or this manuscript is clear. However, we are afraid that, despite being a topic very much related to the research area in our lab, it has been extremely difficult to follow the narrative of the manuscript, which needs to be 'dramatically' rewritten!

We have rewritten large parts of the manuscript, in some cases shortening or removing discussions that became peripheral to the main narrative. We have condensed Fig 1 and 2 into one Figure on the background of the substantial new data added (new Fig 5 and 6). We hope this will provide a much more readable manuscript.

In addition, using quantitative epigenome profiling, the authors provide a comprehensive categorization of different genomic features. These categories are characterized by gain/loss of histone marks and/or gain/loss of expression in different experimental conditions. To increase the readability of this resource, we would suggest including an informative table with all genomic features categorized in this study (active promoters, bivalent promoters, bivalent promoters de novo, primed-only, naïve-only, common bivalent, H3K4me-only promoters, etc), thus including their abundance and the criteria used for their categorization.

This is a very useful suggestion, we have now included this as Supplementary Table 3.

(2) As acknowledged by the authors in their discussion, the suggested functionality of PRC2 on restricting the trophectodermal lineage in naïve hESCs is certainly surprising, as two previous studies indicate a dispensable function of PRC2 for the maintenance of the naïve pluripotent state in hESCs (PMID: 28864533; PMID: 28939884). The authors provided two potential explanations for this inconsistency: (1) the stochasticity differentiation towards trophectoderm of a fraction of hESCs, which might be overlooked in the previous studies; or, (2) a more pronounced phenotype observed in this study by using an EZH2/1 inhibitor, in contrast to a previous study (PMID: 28939884) in which EZH1 expression was reduced in an EZH2-KO background.

However, in the first case, the potential stochasticity is not evaluated in the current study. And, in the second case, the current study associates the functional impact of EPZ-6438 treatment to the loss of H3K27me3, although non-specific or secondary effects (e.g. the observed global lost H3K4me3) are not experimentally ruled out. Thus, considering that the role of PRC2 in sustaining naïve hESCs identity is an

important conclusion raised by the authors, it needs to be further supported at the experimental level.

a) by single-cell RNA sequencing (naïve hESCs +/- EZH2 inh.) to evaluate the proportion of hESCs in naïve state that spontaneously transition to trophectoderm upon EZH2 inhibitor.

We thank the reviewer for this suggestion and have now performed scRNA-seq over a 2, 4, 7 day time course EZH2i treatment, and also used this data to quantify the fraction of cells transitioning to TE and Mesoderm. See also comments to Reviewer 1.

b) Evaluating the functional impact of full KO of PRC2 core components SUZ12 or EED in naïve hESCs (loss of pluripotency markers, gain of trophectoderm lineage).

We thank the reviewer for this suggestion and have addressed this using an acute CRISPR/Cas9 knockout strategy that efficiently depleted EED across clonal naïve hESC colonies. The results presented in Fig. 4e-g, EFig 10b demonstrate GATA3 induction in 16.7% of cells in EED-targeted cultures, compared to 1.4% GATA3+ in untargeted, wildtype cultures. Hence, genetic targeting mirrors the pharmacologic inhibition with EZH2i.

Check figure references:

- Figure 5a, “up” and “down” in the legend are

switched fixed

- Extended Data Figure 10a is not a bar chart, as it says in the figure description, and it does not show what the manuscript refers to.

fixed

- In the results paragraph: “whereas primed-only bivalent genes were predominantly higher expressed (..) in naïve (Fig.3d -> 3c...)”

fixed

- In the discussion paragraph: “on the other hand, we observed an overall reduction

32of core pluripotency marker expression on the population level (Extended Data Fig. 7d -> 8c)

fixed

Extra comments on the comparison of the two co-submissions:

At the functional level, although both studies point towards an implication of PRC2 in pluripotent-to-trophoblasts transition, there are relevant discrepancies among both studies. Zijlmans et al. findings, well-supported at multiple experimental levels, nominate a function of PRC2 as a barrier during trophoblast induction. This is because, in the absence of instructive signals, PRC2 inhibition does not cause spontaneous differentiation or loss of pluripotency marks in naïve hPSCs. This dispensable role of PRC2 in naïve culture conditions is an observation that seems in line with previously published studies (PMID: 28864533; PMID: 28939884). Only when cultured under trophoblast differentiation media, hPSCs transient more efficiently towards trophoblast cells in the presence of the EZH2 inhibitor. In contrast, Kumar et al. study shows that the treatment with EPZ-6438 for 7 days results in the spontaneous differentiation of a fraction of naïve hESCs (yet to be quantified) towards trophoblast cells, in the absence of inductive signals. This would suggest that PRC2 functions as the active blocker of trophoblast differentiation. We agree that the discrepancies might result from technical differences between both studies.

Thanks for pointing this out. We have now addressed this point through a time-course experiment where we assay the emergence GATA3+ cells following EZH2 inhibition with IF and the transcriptional changes on a global level using scRNAseq. This analysis revealed that even in untreated naïve cultures there is a low level of trophectoderm differentiation. After 7d EZH2 inhibition we can detect a robust increase in the trophectoderm-like population after. Our analysis also revealed that we also have mesoderm-like cells in the unperturbed naïve state which increases to 11.7% with 7d EZH2 inhibition. These populations emerged through a shared trajectory where most cells in the inhibited naïve cultures actually initiate progressive differentiation towards these two populations. Of note, we see early gene expression changes within the Epiblast-like population (ELC) of naïve hESC treated with EZH2i already after 2-4 days (Fig 5b,f), but less than 6% of cells have fully differentiated at this point (Fig 5b), hence most significant population changes happen between day 4 and day 7 in our EZH2i time course.

33To add complementary support, we have performed additional experiments using a second inhibitor of PRC2 (EED226 targeting the EED regulatory subunit), and we have performed acute knockout of EED using CRISPR/Cas9. In both cases, we have observed the appearance of GATA3+ colonies comparable to our initial observations with EZH2i. Together, our data clearly supports the functional importance of PRC2 as an epigenetic barrier in naive cells preventing trophectoderm and mesoderm differentiation.

Decision Letter, first revision:

Subject: Your manuscript, NCB-E46376A
Message: Our ref: NCB-E46376A

11th March 2022

Dear Dr. Elsässer,

Thank you for submitting your revised manuscript "Polycomb Repressive Complex 2 shields naïve human pluripotent cells from trophectoderm and mesoderm differentiation" (NCB-E46376A). It has now been seen by the original referees and their comments are below. The reviewers find that the paper has improved in revision, and therefore we'll be happy in principle to publish it in Nature Cell Biology, pending minor revisions to satisfy the referees' final requests and to comply with our editorial and formatting guidelines.

Thank you again for your interest in Nature Cell Biology. Please do not hesitate to contact me if you have any questions.

Sincerely,

Jie Wang, PhD
Senior Editor
Nature Cell Biology

34Tel: +44 (0) 207 843 4924
email: jie.wang@nature.com

Reviewer #1 (Remarks to the Author):

The authors have addressed all the major issues and the more detailed investigation now reflected in the updated title is a nice improvement.

Overall the figure presentation could still be improved, but that is mostly cosmetic and just a suggestion.

Reviewer #2 (Remarks to the Author):

The authors responded to all the issues raised by my comments and adequately addressed my concerns. I believe that the series of studies they have presented is a valuable report on the role of PRC2 and trophoblast differentiation in naive human pluripotent stem cells and represents a significant novelty. However, I have not convinced the authors' interpretation of MeLC on new scRNA-seq data. Do naive PSCs differentiate into MeLC under EZH2i on day 2 (Figure 5b)? Or did a subpopulation of untreated naive PSCs proliferate? 97.7% of Naive 2d EZH2i are ELC and there are no primitive streak-like cells at day 2. Since mesoderm emerges after gastrulation, I guess it takes more time for naive PSCs to differentiate into epiblast of the post-implantation stage. Indeed, the authors wrote primitive streak and mesodermal lineage marker, TBXT, was expressed on day 7 (Figure 5e).

In figure 5b, the authors label ExE Mes near or in MeLC. I wondered if MeLC of Naive 2d EZH2i (Figure 5b) could be ExE Mes rather than MeLC. Are they really mesoderm cells that gastrulating cells give rise to? If the authors claim they are mesoderm or ExE Mes, they should more thoroughly and carefully compare cell types. If the authors want to claim EZH2i induces trophoblast cells from naive PSCs, the authors may not necessarily specify the cell type as mesoderm. If they are ExE Mes, it will be an additional interesting finding but undetermined cells may be enough because I don't think 100% of cells differentiate into specific lineages in vitro.

Reviewer #3 (Remarks to the Author):

We would like to congratulate the authors from both teams for the thorough revision of their manuscript. We are glad that they appreciated our constructive revision and helpful suggestions for the

35improvement of their studies. The new data provided is of high quality and further supports the initial conclusions raised in the first version of their manuscripts. We believe that both studies offer an important source of data for the scientific community, and provide relevant functional insights into the mechanism controlling the lineage specification of human pluripotent cells. Although the discrepancy on whether PRC2 inhibition is sufficient or not to induced trophoblast fate remains unresolved between the co-submitted studies, we acknowledge that both teams have dedicated intense efforts to resolving it. As initially pointed out, the new data indicates that some technical variations are likely responsible for the observed differences. Likewise, both studies agree on the existence of an epigenetic restriction in human naïve pluripotent stem cells and, that PRC2 activity results in chromatin barrier for alternative cell fates. These major conclusions provide a relevant framework to understand cell-type specification during human early embryonic development. We are very pleased to support the publication of both studies in Nature Cell Biology.

Decision Letter, final requests:

Subject: NCB: Your manuscript, NCB-E46376A
Message: Our ref: NCB-E46376A

22nd March 2022

Dear Dr. Elsässer,

Thank you for your patience as we've prepared the guidelines for final submission of your Nature Cell Biology manuscript, "Polycomb Repressive Complex 2 shields naïve human pluripotent cells from trophoblast and mesoderm differentiation" (NCB-E46376A). Please carefully follow the step-by-step instructions provided in the attached file, and add a response in each row of the table to indicate the changes that you have made. Ensuring that each point is addressed will help to ensure that your revised manuscript can be swiftly handed over to our production team.

We would like to start working on your revised paper, with all of the requested files and forms, as soon as possible (preferably within one week). Please get in contact with us if you anticipate delays.

36If you have not done so already, please alert us to any related manuscripts from your group that are under consideration or in press at other journals, or are being written up for submission to other journals (see: <https://www.nature.com/nature-research/editorial-policies/plagiarism#policy-on-duplicate-publication> for details).

In recognition of the time and expertise our reviewers provide to Nature Cell Biology's editorial process, we would like to formally acknowledge their contribution to the external peer review of your manuscript entitled "Polycomb Repressive Complex 2 shields naïve human pluripotent cells from trophoblast and mesoderm differentiation". For those reviewers who give their assent, we will be publishing their names alongside the published article.

Nature Cell Biology offers a Transparent Peer Review option for new original research manuscripts submitted after December 1st, 2019. As part of this initiative, we encourage our authors to support increased transparency into the peer review process by agreeing to have the reviewer comments, author rebuttal letters, and editorial decision letters published as a Supplementary item. When you submit your final files please clearly state in your cover letter whether or not you would like to participate in this initiative. Please note that failure to state your preference will result in delays in accepting your manuscript for publication.

Cover suggestions

As you prepare your final files we encourage you to consider whether you have any images or illustrations that may be appropriate for use on the cover of Nature Cell Biology.

Nature Cell Biology has now transitioned to a unified Rights Collection system which will allow our Author Services team to quickly and easily collect the rights and permissions required to publish your work. Approximately 10 days after your paper is formally accepted, you will receive an email in providing you with a link to complete the grant of rights. If your paper is eligible for Open Access, our Author Services team will also be in touch regarding any additional information that may be required to arrange payment for your article.

Please note that Nature Cell Biology is a Transformative Journal (TJ). Authors may publish their research with us through the traditional subscription access route or make their paper immediately open access through payment of an article-processing charge (APC). Authors will not be required to make a final decision about access to their article until it has been accepted. Find out more about Transformative Journals

Authors may need to take specific actions to achieve compliance with funder and institutional open access mandates. If your research is supported by a funder that requires immediate open access (e.g. according to Plan S principles) then you should select the gold OA route, and we will direct you to the compliant route where possible. For authors selecting the subscription publication route, the journal's standard licensing terms will need to be accepted, including self-archiving policies. Those licensing terms will supersede any other terms that the author or any third party may assert apply to any version of the manuscript.

For information regarding our different publishing models please see our Transformative Journals page. If you have any questions about costs, Open Access requirements, or our legal forms, please contact ASJournals@springernature.com.

[REDACTED]

Best regards,

Ziqian Li
Editorial Assistant
Nature Cell Biology

On behalf of

Jie Wang, PhD
Senior Editor
Nature Cell Biology

Tel: +44 (0) 207 843 4924
email: jie.wang@nature.com

Reviewer #1:

Remarks to the Author:

The authors have addressed all the major issues and the more detailed investigation now reflected in the updated title is a nice improvement.

Overall the figure presentation could still be improved, but that is mostly cosmetic and just a suggestion.

Reviewer #2:

Remarks to the Author:

The authors responded to all the issues raised by my comments and adequately addressed my concerns. I believe that the series of studies they have presented is a valuable report on the role of PRC2 and trophoblast differentiation in naive human pluripotent stem cells and represents a significant novelty. However, I have not convinced the authors' interpretation of MeLC on new scRNA-seq data. Do naive PSCs differentiate into MeLC under EZH2i on day 2 (Figure 5b)? Or did a subpopulation of untreated naive PSCs proliferate? 97.7% of Naive 2d EZH2i are ELC and there are no primitive streak-like cells at day 2. Since mesoderm emerges after gastrulation, I guess it takes more time for naive PSCs to

39differentiate into epiblast of the post-implantation stage. Indeed, the authors wrote primitive streak and mesodermal lineage marker, TBXT, was expressed on day 7 (Figure 5e).

In figure 5b, the authors label ExE Mes near or in MeLC. I wondered if MeLC of Naive 2d EZH2i (Figure 5b) could be ExE Mes rather than MeLC. Are they really mesoderm cells that gastrulating cells give rise to? If the authors claim they are mesoderm or ExE Mes, they should more thoroughly and carefully compare cell types. If the authors want to claim EZH2i induces trophoblast cells from naive PSCs, the authors may not necessarily specify the cell type as mesoderm. If they are ExE Mes, it will be an additional interesting finding but undetermined cells may be enough because I don't think 100% of cells differentiate into specific lineages in vitro.

Reviewer #3:

Remarks to the Author:

We would like to congratulate the authors from both teams for the thorough revision of their manuscript. We are glad that they appreciated our constructive revision and helpful suggestions for the improvement of their studies. The new data provided is of high quality and further supports the initial conclusions raised in the first version of their manuscripts. We believe that both studies offer an important source of data for the scientific community, and provide relevant functional insights into the mechanism controlling the lineage specification of human pluripotent cells. Although the discrepancy on whether PRC2 inhibition is sufficient or not to induced trophoblast fate remains unresolved between the co-submitted studies, we acknowledge that both teams have dedicated intense efforts to resolving it. As initially pointed out, the new data indicates that some technical variations are likely responsible for the observed differences. Likewise, both studies agree on the existence of an epigenetic restriction in human naïve pluripotent stem cells and, that PRC2 activity results in chromatin barrier for alternative cell fates. These major conclusions provide a relevant framework to understand cell-type specification during human early embryonic development. We are very pleased to support the publication of both studies in Nature Cell Biology.

Author Rebuttal, first revision:

Reviewer #1:

Remarks to the Author:

The authors have addressed all the major issues and the more detailed investigation now reflected in the updated title is a nice improvement.

40Overall the figure presentation could still be improved, but that is mostly cosmetic and just a suggestion.

We thank the reviewer and hope that figure rearrangements and various formatting updates will improve the presentation

Reviewer #2:

Remarks to the Author:

The authors responded to all the issues raised by my comments and adequately addressed my concerns. I believe that the series of studies they have presented is a valuable report on the role of PRC2 and trophoblast differentiation in naive human pluripotent stem cells and represents a significant novelty.

However, I have not convinced the authors' interpretation of MeLC on new scRNA-seq data. Do naive PSCs differentiate into MeLC under EZH2i on day 2 (Figure 5b)? Or did a subpopulation of untreated naive PSCs proliferate? 97.7% of Naive 2d EZH2i are ELC and there are no primitive streak-like cells at day 2. Since mesoderm emerges after gastrulation, I guess it takes more time for naive PSCs to differentiate into epiblast of the post-implantation stage. Indeed, the authors wrote primitive streak and mesodermal lineage marker, TBXT, was expressed on day 7 (Figure 5e).

In figure 5b, the authors label ExE Mes near or in MeLC. I wondered if MeLC of Naive 2d EZH2i (Figure 5b) could be ExE Mes rather than MeLC. Are they really mesoderm cells that gastrulating cells give rise to? If the authors claim they are mesoderm or ExE Mes, they should more thoroughly and carefully compare cell types. If the authors want to claim EZH2i induces trophoblast cells from naive PSCs, the authors may not necessarily specify the cell type as mesoderm. If they are ExE Mes, it will be an additional interesting finding but undetermined cells may be enough because I don't think 100% of cells differentiate into specific lineages in vitro.

We thank the reviewer for these insightful comments. We have also invested significant efforts in elucidating the nature of the MeLC. Indeed, we also thought it would be more likely that the MeLC could be ExE Mes rather than definitive mesoderm as ExE would be "closer" in developmental time to the naive epiblast. Indeed, a fraction of the MeLC in the starting naive cultures resemble the ExE mesoderm of the gastrulation stage embryo (Fig 5b), but with EZH2 inhibition we see primarily an increase of MeLC which more closely aligns with the definitive mesoderm reference. This was indeed somewhat unexpected to us, but we subsequently saw transcripts indicating streak formation and gastrulation, including TBXT. The trajectory analysis in Figure 6 further supports a true differentiation towards mesoderm through a streak/gastrulation process. We would also like to point out that the MeLC, just as the TLC are present at low fractions already prior to EZH2i indicating a low grade spontaneous differentiation in naive stem cell cultures. However, we are also cautious about

41the fact that the developmental origin of ExE mesoderm is still uncertain in the human (including whether it is formed through a streak process or not) and genes that conclusively would distinguish it from definitive mesoderm is also an open question. For this reason we would like to not make a strong distinction between ExE and definitive mesoderm in this analysis, although most data point towards a dominating fraction of definitive mesoderm. This is why we would like to keep the more generic term of MeLC and include both types. The distinction between ExE and definitive mesoderm is a fascinating and still largely uncharted question in human developmental biology which we hope will receive more attention in the coming years.

We are willing to tone down the claims concerning MeLC but we do not want to annotate the cells as “undetermined cells” as they truly align with the mesoderm reference of gastrulation stage embryos which we think is a relevant observation to report. For these reasons we agree to tone down our focus on the mesoderm and

- 1) omit the reference to mesoderm in the title “*Polycomb 1 Repressive Complex 2 shields naive human pluripotent cells from trophoblast differentiation*”.
- 2) In the results section we now further highlight the presence of both ExE and definitive mesoderm in our reference as well in the cells within the naive cultures with and without EZH inhibition and that we merge the two in our MeLC annotation.

Reviewer #3:

Remarks to the Author:

We would like to congratulate the authors from both teams for the thorough revision of their manuscript. We are glad that they appreciated our constructive revision and helpful suggestions for the improvement of their studies. The new data provided is of high quality and further supports the initial conclusions raised in the first version of their manuscripts. We believe that both studies offer an important source of data for the scientific community, and provide relevant functional insights into the mechanism controlling the lineage specification of human pluripotent cells. Although the discrepancy on whether PRC2 inhibition is sufficient or not to induced trophoblast fate remains unresolved between the co-submitted studies, we acknowledge that both teams have dedicated intense efforts to resolving it. As initially pointed out, the new data indicates that some technical variations are likely responsible for the observed differences.

Likewise, both studies agree on the existence of an epigenetic restriction in human naïve pluripotent stem cells and, that PRC2 activity results in chromatin barrier for alternative cell fates. These major conclusions provide a relevant framework to understand cell-type specification during human early embryonic development. We are very pleased to support the publication of both studies in Nature Cell Biology.

We thank the reviewer for these comments. We have of course also been curious to understand the different outcome between the two studies. Since the same H9 cell line is

42used, and we have confirmed our observation in a second cell line, genetic background is not a variable to consider. However, the different timing (7d vs 4d), inhibitor used (EPZ vs UNC) and medium (t2iLGö vs PXGL) leave a number of variables that could explain the discrepancy. Because of the prominent induction of Wnt5a (Fig 5f, 6e) and the fact that a major distinction between t2iLGö and PXGL is the inclusion of Tankyrase/WNT pathway inhibitor XAV939 in the latter, we considered that culture media contributed to the difference. XAV939 has already been shown to reduce basal expression of GATA3 and TBXT in naïve hESC (ref 70). Hence, to support a more direct comparative discussion of the two studies, we performed an experiment analogous to Fig 4b comparing t2iLGö and PXGL media (now added as Extended Data Fig 7c). Indeed we found that our H9 cells treated for 7d with EZH2i (EPZ) in PXGL did not generate GATA3+ colonies.

We cannot conclusively derive from this experiment if differentiation is blocked entirely or delayed, but speculate that WNT signaling is required for cells to enter the 'activated' ELC state. In the revised manuscript we have now included a discussion of the work by Pasque and colleagues and we also refer to Extended Data Fig 7c in the discussion.

Final Decision Letter:

Subject: Decision on Nature Cell Biology submission NCB-E46376B

Message:

Dear Dr Elsässer,

I am pleased to inform you that your manuscript, "Polycomb Repressive Complex 2 shields naïve human pluripotent cells from trophectoderm differentiation", has now been accepted for publication in Nature Cell Biology.

43After the grant of rights is completed, you will receive a link to your electronic proof via email with a request to make any corrections within 48 hours. If, when you receive your proof, you cannot meet this deadline, please inform us at rjsproduction@springernature.com immediately.

Please note that Nature Cell Biology is a Transformative Journal (TJ). Authors may publish their research with us through the traditional subscription access route or make their paper immediately open access through payment of an article-processing charge (APC). Authors will not be required to make a final decision about access to their article until it has been accepted. Find out more about Transformative Journals

Authors may need to take specific actions to achieve compliance with funder and institutional open access mandates. If your research is supported by a funder that requires immediate open access (e.g. according to Plan S principles) then you should select the gold OA route, and we will direct you to the compliant route where possible. For authors selecting the subscription publication route, the journal's standard licensing terms will need to be accepted, including self-archiving policies. Those licensing terms will supersede any other terms that the author or any third party may assert apply to any version of the manuscript.

If you have not already done so, we strongly recommend that you upload the step-by-step protocols used in this manuscript to the Protocol Exchange (www.nature.com/protocolexchange), an open online resource established by Nature Protocols that allows researchers to share their detailed experimental know-how. All uploaded protocols are made freely available, assigned DOIs for ease of citation and are fully searchable through nature.com. Protocols and Nature Portfolio journal papers in which they are used can be linked to one another, and this link is clearly and prominently visible in the online versions of both papers. Authors who performed the specific experiments can act as primary authors for the Protocol as they will be best placed to share the methodology details, but the Corresponding Author of the present research paper should be included as one of the authors. By uploading your Protocols to Protocol Exchange, you are enabling researchers to more readily reproduce or adapt the methodology you use, as well as increasing the visibility of your protocols and papers. You can also establish a dedicated page to collect your lab Protocols. Further information can be found at www.nature.com/protocolexchange/about

With kind regards,

Jie Wang, PhD
Senior Editor
Nature Cell Biology

Tel: +44 (0) 207 843 4924
email: jie.wang@nature.com